# DISTRIBUTIONALLY ROBUST POLICY LEARNING UNDER CONCEPT DRIFTS

## ABSTRACT

Distributionally robust policy learning aims to find a policy that performs well under the worst-case distributional shift, and yet most existing methods for robust policy learning consider the worst-case *joint* distribution of the covariate and the outcome. The joint-modeling strategy can be unnecessarily conservative when we have more information on the source of distributional shifts. This paper studies a more nuanced problem — robust policy learning under the *concept drift*, when only the conditional relationship between the outcome and the covariate changes. To this end, we first provide a doubly-robust estimator for evaluating the worst-case average reward of a given policy under a set of perturbed conditional distributions. We show that the policy value estimator enjoys asymptotic normality even if the nuisance parameters are estimated with a slower-than-root-$n$ rate. We then propose a learning algorithm that outputs the policy maximizing the estimated policy value within a given policy class $\Pi$, and show that the sub-optimality gap of the proposed algorithm is of the order $\kappa(\Pi)n^{-1/2}$, with $\kappa(\Pi)$ is the entropy integral of $\Pi$ under the Hamming distance and $n$ is the sample size. A matching lower bound is provided to show the optimality of the rate. The proposed methods are implemented and evaluated in numerical studies, demonstrating substantial improvement compared with existing benchmarks.

## 1 INTRODUCTION

In a wide range of fields, the abundance of user-specific historical data provides opportunities for learning efficient individualized policies. Examples include learning the optimal personalized treatment from electronic health record data (Murphy, 2003; Kim et al., 2011; Chan et al., 2012), or obtaining an individualized advertising strategy using past customer behavior data (Bottou et al., 2013; Kallus & Udell, 2016). Driven by such a practical need, a line of works have been devoted to developing efficient policy learning algorithms using historical data — a task often known as *offline policy learning* (Dudík et al., 2011; Zhang et al., 2012; Swaminathan & Joachims, 2015a;b;c; Kitagawa & Tetenov, 2018; Athey & Wager, 2021; Zhou et al., 2023; Zhan et al., 2023; Bibaut et al., 2021; Jin et al., 2021; 2022a).

Most existing methods for offline policy learning deliver performance guarantees under the premise that the target environment remains the same as that from which the historical data is collected. It has been widely observed, however, that such a condition is hardly met in practice (see e.g., Recht et al. (2019); Namkoong et al. (2023); Liu et al. (2023); Jin et al. (2023) and the references therein). Under distribution shift, a policy learned in one environment often shows degraded performance when deployed in another environment. To address this issue, there is an emerging body of research on *robust policy learning*, which aims at finding a policy that still performs well when the target distribution is perturbed. Pioneering works in this area consider the case where the *joint distribution* of the covariates and the outcome is shifted from the training distribution, and devise algorithms that output a policy achieving reliable worst-case performance under the aforementioned shifts Si et al. (2023); Kallus et al. (2022). The joint modeling approach, however, ignores the *type* of distributional shifts, and the resulting worst-case value can be unnecessarily conservative in practice.

Indeed, distributional shifts can be categorized into two classes by their sources: (1) the shift in the covariate $X$, and/or (2) the shift in the conditional relationship between the outcome $Y$ and the covariate $X$. The two types of distributional shifts are different in nature, have different implications

on the objectives, and call for distinct treatment (Namkoong et al., 2023; Liu et al., 2023; Jin et al., 2023; Ai & Ren, 2024). To be concrete, imagine that the distribution of covariates changes while that of $Y \mid X$ remains invariant — in this case, the distribution shift is identifiable/estimable since the covariates are often accessible in the target environment. As a result, it is often uncessary to account for the worst-case covariate shift rather than directly correcting for it. Alternatively, when the $Y \mid X$ distribution changes but the $X$ distribution remains invariant, the distribution shift is no longer identifiable, and we need to account for the worst-case situation. This latter setting, known as *concept drift*, occurs when the distribution of the unobserved confounder changes over time, or due to sudden external shocks (Widmer & Kubat, 1996; Lu et al., 2018; Gama et al., 2014). For example, in advertising, the customer behavior can evolve over time as the environment changes, while the population remains largely the same. In personalized medicine, treatment may be affecting patients' outcomes through some unmeasured confounders that have different distributions in the training and target cohort, thereby inducing a concept drift. In these applications — with the one extra bit of information that the shift is only in the conditional reward distribution — can we obtain a more efficient policy learning algorithm?

Motivated by the above situations, we study robust policy learning under concept drift in this work. Most existing methods for robust policy learning (Si et al., 2023; Kallus et al., 2022) model the distributional shift jointly without distinguishing the sources, and the corresponding algorithms turn out to be suboptimal. The reason behind their suboptimality is that the worst-case distributions under the two models — the joint-shift model and the concept-drift model — can be substantially different, so it would be a "waste" of our budget to consider adversarial distributions that are not feasible under concept drift. It is worth mentioning that a recent paper by Mu et al. (2022) accounts for the sources of distributional shifts in policy learning; their approach, however, applies only when the covariates take *a finite number of values*, and therefore is limited in its applicability. When the covariate space is infinite, it remains unclear how to efficiently learn a robust policy under concept drift. The current work aims to fill in the gap by answering the following question:

*How can we efficiently learn a policy with optimal worst-case average performance under concept drift with minimal assumptions?*

We provide a rigorous answer to the above question. Specifically, we assume the covariate distribution remains the same in the training and target environments, while the $Y \mid X$ distribution shift is bounded in KL-divergence by a pre-specified constant $\delta$. Our goal is to find a policy that maximizes the worst-case averaged outcome over all possible target distributions satisfying the previous condition.

## 1.1 OUR CONTRIBUTIONS

Towards robust policy learning under concept drift, we make the following contributions.

1. *Policy evaluation:* Given a policy, we present a doubly-robust estimator for the worst-case policy value under concept drift. We prove that the estimator is asymptotic normal under mild conditions on the estimation rate of the nuisance parameter. Our approach involves solving the dual form of a distributionally robust optimization problem and taking a de-biased step to deal with the slow convergence of the optimizer, thereby obtaining an estimator with root-$n$ convergence rate.

2. *Policy learning:* We propose a robust policy learning algorithm that outputs a policy maximizing the estimated policy value over a policy class $\Pi$. Compared with the oracle optimal policy, the policy provided by our algorithm with high probability has a regret/suboptimality gap of the order $\kappa(\Pi)/\sqrt{n}$, where $\kappa(\Pi)$ is a measure quantifying the policy class complexity (to be formalized shortly) and $n$ is the number of samples. Compared with Mu et al. (2022), our algorithm and theory apply to general covariate spaces and potentially infinite policy classes, while their method is restricted to finite covariate space and policy class. We complement the upper bound with a matching lower, thus establishing the minimax optimality of our proposed algorithm. We summarize the comparison between our result and prior work in Table 1 for better demonstration.

3. *Implementation and empirics:* We provide efficient implementation of our robust policy learning algorithm, and compare its empirical performance with existing benchmarks in numerical studies. Our proposed method exhibits substantial improvement.

| | Distribution shift | Unknown $\pi_0$ | General $\mathcal{X}$ | Upper bound | Lower bound |
|---|---|---|---|---|---|
| Athey & Wager (2021) | ✗ | ✓ | ✓ | — | — |
| Zhou et al. (2023) | ✗ | ✓ | ✓ | — | — |
| Si et al. (2023) | Joint | ✗ | ✓ | — | — |
| Kallus et al. (2022) | Joint | ✓ | ✓ | — | — |
| Mu et al. (2022) | Separate | ✓ | ✗ | $O\left(\sqrt{\frac{\log n \log(|\mathcal{X}||\mathcal{A}|)}{n}}\right)$ | ✗ |
| This work | Separate | ✓ | ✓ | $O\left(\frac{\kappa(\Pi)}{\sqrt{n}}\right)$ | $\Omega\left(\sqrt{\frac{\mathrm{Ndim}(\Pi)}{n}}\right)$ |

Table 1: Comparison of results in the offline policy learning literature. "Unknown $\pi_0$" refers to whether an algorithm assumes knowledge of the behavior policy $\pi_0$. "General $\mathcal{X}$" refers to whether an algorithm allows for general types of covariates. Athey & Wager (2021); Zhou et al. (2023); Si et al. (2023); Kallus et al. (2022) have the regret upper and lower bounds for the specific problems they consider that are not directly comparable to ours, so we do not include them in the table. $|\mathcal{X}|$ refers to the cardinality of the covariate support (if finite) and $|\mathcal{A}|$ to that of the action set. $\kappa(\Pi)$ and $\mathrm{Ndim}(\Pi)$ are the entropy integral under Hamming distance and the Natarajan dimension of a policy class $\Pi$, with the relation $\kappa(\Pi) = O(\log(d)\mathrm{Ndim}(\Pi))$, where $d$ is the covariate space dimension.

## 1.2 RELATED WORKS

**Offline policy learning.** There is a long list of works devoted to offline policy learning. Most of them assume no distributional shifts (e.g., Dudík et al. (2011); Zhang et al. (2012); Swaminathan & Joachims (2015a;b;c); Kitagawa & Tetenov (2018); Athey & Wager (2021); Zhou et al. (2023)). Zhan et al. (2023); Jin et al. (2021; 2022a) allow the data to be adaptively collected, but the distribution over the covariate and the (potential) outcomes remain invariant in the training and target environment.

As mentioned earlier, the work of Si et al. (2023); Kallus et al. (2022) study robust policy learning when the joint distribution of $(X, Y)$ ranges in the neighborhood of the training distribution; Mu et al. (2022) consider the case when the covariate shift and $Y \mid X$ shift are specified separately; their method, however, is restricted to finite covariate space, and their sub-optimality gap is logarithmic factors slower than parametric rates. The work of Kallus & Zhou (2021) concerns robust policy learning when the distribution shift is caused by hidden confounders — this is in fact a special type of concept drift — and the corresponding $Y \mid X$ shift is assumed to be bounded uniformly, which is quite different from our $f$-divergence bound. More recently, Guo et al. (2024) considers a pure covariate shift with a focus on policy evaluation, where the setup and the goal are different from ours.

**Distributionally robust optimization.** More broadly, our work is also closely related to DRO, where the goal is to learn a model that has good performance under the worst-case distribution (e.g., Bertsimas & Sim (2004); Delage & Ye (2010); Hu & Hong (2013); Duchi et al. (2019); Dudík et al. (2011); Zhang et al. (2023)). The major focus of the aforementioned works involves parameter estimation and prediction in supervised settings; we however take a decision-making perspective and aim at learning a individualized policy with optimal worst-case performance guarantees.

## 1.3 NOTATION

We use $[n]$ to denote the discrete set $\{1, 2, \cdots, n\}$ for any $n \in \mathbb{Z}$. We use $\mathrm{argmin}$ and $\mathrm{argmax}$ to denote the minimizers and maximizers; if the minimzer or the maximizer cannot be attained, we project it back to the feasible set. We denote the usual $p$-norm as $\|\cdot\|_p$. For any probability measure $P$ defined on the probability space $(\Omega, \sigma(\Omega), P)$. For any function $f$, we denote the $L_2(P)$-norm of $f$ conventionally as $\|f\|_{L_2(P)} = (\int |f(x)|^2 \, dP(x))^{1/2}$ and $\|f\|_{L_\infty} = \sup_{x \in \mathcal{X}} |f(x)|$. We use $\widehat{P}$ to denote the empirical distribution of $P$. For any random variables $X, Y$, we use $X \perp\!\!\!\perp Y$ to denote that $X$ is independent of $Y$. For a random variable/vector $X$, we use $\mathbb{E}_X[\cdot]$ to indicate the expectation taken over the distribution of $X$.

## 2 PRELIMINARIES

Consider a set of $M$ actions denoted by $[M]$ and let $\mathcal{X} \subseteq \mathbb{R}^d$. Throughout the paper, we follow the potential outcome framework (Imbens & Rubin, 2015), where $Y(a) \in \mathcal{Y}_a \subseteq \mathbb{R}$ denotes the

potential outcome had action $a$ been taken for any $a \in [M]$. We posit the underlying data-generating distribution $P$ on the joint covariate-outcome random vector $(X, Y(1), \cdots, Y(M)) \in \mathcal{X} \times \prod_{a=1}^{M} \mathcal{Y}_a$. Consider a data set $\mathcal{D} = \{(X_i, A_i, Y_i)\}_{i \in [n]}$ consisting of $n$ i.i.d. draws of $(X, A, Y)$, where $X_i \in \mathcal{X}$ is the observed contextual vector, $A_i \in [M]$ the action, and $Y_i = Y(A_i)$ the realized reward. The actions are selected by the *behavior policy* $\pi_0$, where $\pi_0(a \,|\, x) := \mathbb{P}(A_i = a \,|\, X = x)$ is the *propensity score*, for any $a \in [M], x \in \mathcal{X}$. We make the following assumptions for $\pi_0$ and $P$.

**Assumption 2.1.** The behavior policy $\pi_0$ and the joint distribution $P$ satisfy the following.

(1) *Unconfoundedness:* $(Y(1), \cdots, Y(M)) \perp\!\!\!\perp A \,|\, X$.

(2) *Overlap:* for some $\varepsilon > 0$, $\pi_0(a \,|\, x) \geq \varepsilon$, for all $(a, x) \in [M] \times \mathcal{X}$.

(3) *Bounded reward support:* there exists $\bar{y} > 0$, such that $0 \leq Y(a) \leq \bar{y}$ for all $a \in [M]$.

The above assumptions are standard in the literature (see e.g., Athey & Wager, 2021; Zhou et al., 2023; Si et al., 2023; Kallus et al., 2022). In particular, the unconfoundedness assumption guarantees identifiability, and the overlap assumption ensures sufficient exploration when collecting the training dataset. The bounded reward support is assumed for the ease of exposition, and can be relaxed to the sub-Gaussian reward straightforwardly.

### 2.1 THE KL-DISTRIBUTIONALLY ROBUST FORMULATION

Given the training set $\mathcal{D} = \{(X_i, A_i, Y_i)\}_{i \in [n]}$ and a policy class $\Pi$, we aim to learn a policy $\pi \in \Pi$ that achieves high expected reward in a target environment that may deviate from the data-collection environment where $\mathcal{D}$ is collected. While distribution shift can take place in various forms, we focus primarily on the concept drift, where only the conditional reward distribution $Y(a) \,|\, X$ differs in the training and target environment. The distribution shift is quantified by the KL divergence.

**Definition 2.2** (KL divergence). The KL divergence between two distributions $Q$ and $P$ is defined as $D_{\mathrm{KL}}(Q \,\|\, P) = \mathbb{E}_Q[\log \frac{dQ}{dP}]$, where $\frac{dQ}{dP}$ is the Radon-Nikodym derivative of $Q$ with respect to $P$.

We define an uncertainty set of neighboring distributions around $P$, whose conditional outcome distribution is bounded in KL divergence from $P$. Given a radius $\delta > 0$, the uncertainty set of the conditional distribution is defined as $\mathcal{P}(P_{Y \,|\, X}, \delta) := \{Q_{Y \,|\, X} : D_{\mathrm{KL}}(Q_{Y \,|\, X} \,\|\, P_{Y \,|\, X}) \leq \delta\}$, where $P_{Y \,|\, X}$ and $Q_{Y \,|\, X}$ refers to the distribution of $(Y(1), \ldots, Y(d)) \,|\, X$ under $P$ and $Q$ respectively. The distributionally robust policy value for any policy $\pi$ at level $\delta$ is defined as

$$\mathcal{V}_\delta(\pi) := \mathbb{E}_{P_X}\left[ \inf_{Q_{Y \,|\, X} \in \mathcal{P}(P_{Y \,|\, X}, \delta)} \mathbb{E}_{Q_{Y \,|\, X}}\left[ Y\big(\pi(X)\big) \,\Big|\, X \right] \right]. \tag{1}$$

The optimal policy in $\Pi$ is the one that maximizes $\mathcal{V}_\delta(\pi)$, i.e. $\pi_\delta^* := \mathrm{argmax}_{\pi \in \Pi} \ \mathcal{V}_\delta(\pi)$.[1]

Under this formulation, our goal is to learn a "robust" policy with a high value of $\mathcal{V}_\delta(\pi)$ using a dataset drawn from $P$. The task here is two-fold: we need to (i) estimate the policy value $\mathcal{V}_\delta(\pi)$ for a given policy $\pi$, and (ii) find a near-optimal robust policy $\widehat{\pi} \in \Pi$ whose policy value is close to the optimal policy $\pi_\delta^*$. Here, the performance of a learned policy $\widehat{\pi}$ is measured by the sub-optimality gap (regret), defined as $\mathcal{R}_\delta(\widehat{\pi}) := \mathcal{V}_\delta(\pi_\delta^*) - \mathcal{V}_\delta(\widehat{\pi})$.

In the following sections, we tackle each task sequentially.

### 2.2 STRONG DUALITY

In order to estimate $\mathcal{V}_\delta(\pi)$, we first rewrite the inner optimization problem in Equation (1) in its dual form using standard results in convex optimization (see e.g., Luenberger (1997)). The transformation is formalized in the following lemma, with its proof provided in Appendix B.1.

**Lemma 2.3** (Strong Duality). *Given any $\pi \in \Pi$ and any $x \in \mathcal{X}$, the optimal value of inner optimization problem in Equation (1) equals to*

$$- \min_{\alpha \geq 0, \eta \in \mathbb{R}} \mathbb{E}_P\left[ \alpha \exp\Big( -\frac{Y(\pi(X)) + \eta}{\alpha} - 1 \Big) + \eta + \alpha\delta \,\Big|\, X = x \right]. \tag{2}$$

---

[1]When the supremum cannot be attained, we can always construct a sequence of policies whose policy values converge to the supremum, and all the arguments go through with a limiting argument.

We note that the optimization problem in (2) depends on $x$ and $\pi$ — to manifest this dependence, we use $(\boldsymbol{\alpha}_\pi^*(x), \boldsymbol{\eta}_\pi^*(x))$ to denote its optimizer, i.e., $\boldsymbol{\alpha}_\pi^*$ and $\boldsymbol{\eta}_\pi^*$ are functions of $x$ and

$$\left(\boldsymbol{\alpha}_\pi^*(x), \boldsymbol{\eta}_\pi^*(x)\right) \in \underset{\alpha \geq 0, \eta \in \mathbb{R}}{\arg\min} \, \mathbb{E}_P\left[\alpha \exp\left(-\frac{Y(\pi(X)) + \eta}{\alpha} - 1\right) + \eta + \alpha\delta \,\Big|\, X = x\right].$$

With this notation and Lemma 2.3, the robust policy value becomes

$$\mathcal{V}_\delta(\pi) = -\mathbb{E}_P\left[\boldsymbol{\alpha}_\pi^*(X) \exp\left(-\frac{Y(\pi(X)) + \boldsymbol{\eta}_\pi^*(X)}{\boldsymbol{\alpha}_\pi^*(X)} - 1\right) + \boldsymbol{\eta}_\pi^*(X) + \boldsymbol{\alpha}_\pi^*(X)\delta\right]. \tag{3}$$

The above formulation has thus translated the original distributionally robust optimization problem into an *empirical risk minimization (ERM)* problem. We note that, unlike the well-studied joint distributional shift formulation, the above representation admits an optimizer pair $(\boldsymbol{\alpha}_\pi^*(x), \boldsymbol{\eta}_\pi^*(x))$ that is *dependent* on the context $x$ (i.e. $\boldsymbol{\alpha}_\pi^*, \boldsymbol{\eta}_\pi^*$ are functions of $x$) and the policy $\pi$. As we shall see shortly, our proposed policy value estimation procedure employs ERM tools to estimate $(\boldsymbol{\alpha}_\pi^*, \boldsymbol{\eta}_\pi^*)$, and then compute an estimate of $\mathcal{V}_\delta(\pi)$ by plugging $(\boldsymbol{\alpha}_\pi^*, \boldsymbol{\eta}_\pi^*)$ into Equation (3).

The remaining challenge in this proposal is the slow estimation rate of the optimizers — if we naïvely plug in the optimizers, the resulting policy value estimator typically has a convergence rate slower than root-$n$. To overcome this, we incorporate a novel adjustment method to debias the estimator, which allows us to obtain a doubly-robust estimator that achieves root-$n$ rate of convergence even when then nuisance parameters (e.g., $(\boldsymbol{\alpha}_\pi^*, \boldsymbol{\eta}_\pi^*)$) are converging slower than the root-$n$ rate.

We end this section by discussing when $\boldsymbol{\alpha}_\pi^*(x) > 0$. Throughout, we shall make the following mild assumption on the conditional outcome distribution.

**Assumption 2.4.** For $a \in [M]$ and $x \in \mathcal{X}$, define $\underline{y}(x; a) = \sup\{t : \mathbb{P}(Y(a) < t \mid X = x, A = a) = 0\}$ and $\tilde{p}(x; a) = \mathbb{P}(Y(a) = \underline{y}(x; a) \mid X = x, A = a)$. It holds that $\log(1/\tilde{p}(x; a)) > \delta$ for $P_{X|A=a}$-almost all $x$.

The above assumption requires that $P_{Y \mid X, A}$ does not posit a large point mass at its essential infimum, which can be satisfied by many commonly used distributions, e.g., all the continuous distributions. The following result from Jin et al. (2022b, Proposition 4), shows that $\alpha^* > 0$ when Assumption 2.4 holds, which ensures that the gradient of the risk function in ERM has a zero mean.

**Proposition 2.5** (Jin et al. (2022b))**.** *Under Assumption 2.4, the optimizer $\alpha^*$ of (2) satisfies $\alpha^* > 0$.*

## 3 POLICY VALUE ESTIMATION UNDER CONCEPT DRIFT

### 3.1 THE ESTIMATION PROCEDURE

Fixing a policy $\pi$, we aim to estimate the policy value $\mathcal{V}_\delta(\pi)$ using the training dataset $\mathcal{D}$. We first split $\mathcal{D}$ into $K$ equally sized disjoint folds, $\mathcal{D}^{(k)}$ for $k \in [K]$,[2] where we slightly abuse the notation to use $\mathcal{D}^{(k)}$ for denoting the data points or the corresponding indices interchangeably. For each $k \in [K]$, we use data points in $\mathcal{D}^{(k+1)}$ to obtain the propensity score estimator $\widehat{\pi}_0^{(k)}$ and the optimizers $(\widehat{\boldsymbol{\alpha}}_\pi^{(k)}, \widehat{\boldsymbol{\eta}}_\pi^{(k)})$.[3] Next, we define

$$\widehat{G}_\pi^{(k)}(x, y) := \widehat{\boldsymbol{\alpha}}_\pi^{(k)}(x) \cdot \exp\left(-\frac{y + \widehat{\boldsymbol{\eta}}_\pi^{(k)}(x)}{\widehat{\boldsymbol{\alpha}}_\pi^{(k)}(x)} - 1\right) + \widehat{\boldsymbol{\eta}}_\pi^{(k)}(x) + \widehat{\boldsymbol{\alpha}}_\pi^{(k)}(x) \cdot \delta,$$

and its conditional expectation $\bar{g}_\pi^{(k)}(x) := \mathbb{E}_P\left[\widehat{G}_\pi^{(k)}\left(X, Y(\pi(X))\right) \mid X = x\right]$. We then use $\mathcal{D}^{(k+2)}$ to obtain $\widehat{g}_\pi^{(k)}$ as an estimator of $g_\pi$. The policy value estimator $\widehat{\mathcal{V}}_\delta^{(k)}(\pi)$ for the $k$-th fold is

$$\widehat{\mathcal{V}}_\delta^{(k)}(\pi) = \frac{1}{|\mathcal{D}^{(k)}|} \sum_{i \in \mathcal{D}^{(k)}} \frac{\mathbb{1}\{\pi(X_i) = A_i\}}{\widehat{\pi}_0^{(k)}(A_i \mid X_i)} \cdot \left(\widehat{G}_\pi^{(k)}(X_i, Y_i) - \widehat{g}_\pi^{(k)}(X_i)\right) + \widehat{g}_\pi^{(k)}(X_i). \tag{4}$$

The final policy value estimator is given by $\widehat{\mathcal{V}}_\delta(\pi) := -\frac{1}{K} \sum_{k=1}^K \widehat{\mathcal{V}}_\delta^{(k)}(\pi)$. The complete procedure is summarized in Algorithm 1. A few remarks are in order.

---

[2] We assume without loss of generality that $n$ is divisible by $K$. In practice, we only need a minimum of $K = 3$ folds.

[3] We use the convention that $\mathcal{D}^{(k+j)} = \mathcal{D}^{(k+j \bmod K)}$ for any $j, k$.

---

**Algorithm 1** Policy estimation under concept drift

---

**Input:** Dataset $\mathcal{D}$; policy $\pi$; uncertainty set parameter $\delta$; propensity score estimation algorithm $\mathcal{C}$; ERM algorithm $\mathcal{E}$ for obtaining $(\boldsymbol{\alpha}_\pi^*, \boldsymbol{\eta}_\pi^*)$; regression algorithm $\mathcal{R}$ for estimating $\bar{g}_\pi$.

Randomly split $\mathcal{D}$ into $K$ non-overlapping equally-sized folds $\mathcal{D}^{(k)}$, $k \in [K]$;
**for** $k = 1, \cdots, K$ **do**
  On $\mathcal{D}^{(k+1)}$: $\widehat{\pi}_0^{(k)} \leftarrow \mathcal{C}(\mathcal{D}^{(k+1)})$, $(\widehat{\boldsymbol{\alpha}}_\pi^{(k)}, \widehat{\boldsymbol{\eta}}_\pi^{(k)}) \leftarrow \mathcal{E}(\mathcal{D}^{(k+1)})$;
  On $\mathcal{D}^{(k+2)}$: $\widehat{g}_\pi^{(k)} \leftarrow \mathcal{R}(\{X_i, A_i, \widehat{G}_\pi^{(k)}(X_i, Y_i); i \in \mathcal{D}^{(k+2)}\})$;
  On $\mathcal{D}^{(k)}$: compute $\widehat{\mathcal{V}}_\delta^{(k)}(\pi)$ according to Equation (4);
**end for**

**Return:** $\widehat{\mathcal{V}}_\delta(\pi) \leftarrow -\frac{1}{K} \sum_{k=1}^K \widehat{\mathcal{V}}_\delta^{(k)}(\pi)$.

---

*Remark* 3.1. The estimation procedure involves three model-fitting steps corresponding to $\pi_0$, $(\boldsymbol{\alpha}_\pi^*, \boldsymbol{\eta}_\pi^*)$, and $\bar{g}_\pi$, respectively. The propensity score function $\pi_0$ can be estimated with off-the-shelf algorithms (e.g., logistic regression, random forest); the conditional mean $\bar{g}_\pi^{(k)}$ can be obtained by regressing $\widehat{G}_\pi^{(k)}(X_i, Y_i)$ onto $X_i$ for the points such that $A_i = \pi(X_i)$ with standard regression algorithms, e.g., kernel regression (Nadaraya, 1964; Watson, 1964), local polynomial regression (Cleveland, 1979; Cleveland & Devlin, 1988), smoothing spline (Green & Silverman, 1993), regression trees (Loh, 2011) and random forests (Ho et al., 1995). The ERM step is more complex, and will be discussed in detail shortly.

*Remark* 3.2. The construction of the estimator $\widehat{\mathcal{V}}_\delta(\pi)$ employs two major techniques: cross-fitting and de-biasing. The cross-fitting technique crucially provides the convenient property of independence and the de-biasing technique overcomes the slow rate of estimating the nuisance parameter $\alpha_\pi, \eta_\pi$, leading to the doubly-robust property of the proposed estimator.

**The ERM step.** For notational simplicity, we denote $\theta = (\alpha, \eta)$ and write the loss function as

$$\ell(x, y; \theta) = \alpha \exp\left(-\frac{y + \eta}{\alpha} - 1\right) + \eta + \alpha\delta. \tag{5}$$

By the notation, $\boldsymbol{\theta}_\pi^*(x) = (\boldsymbol{\alpha}_\pi^*(x), \boldsymbol{\eta}_\pi^*(x))$ is the optimizer of $\mathbb{E}_P[\ell(x, Y(\pi(x)); \theta) \mid X = x]$ with respect to $\theta$. Throughout, we make the following assumption on $\theta_\pi^*$.

**Assumption 3.3.** For any policy $\pi$, there exist constants $\underline{\alpha}, \bar{\alpha}, \bar{\eta}$ such that $0 < \underline{\alpha} \le \boldsymbol{\alpha}_\pi^*(x) \le \bar{\alpha}$, and $\left|\boldsymbol{\eta}_\pi^*(x)\right| \le \bar{\eta}$, for all $x \in \mathcal{X}$.

The above assumption is mild and can be achieved, for example, when $\boldsymbol{\theta}_\pi^*(x)$ is continuous in $x$ and when $\mathcal{X}$ is compact. We refer the readers to Jin et al. (2022b) for a more detailed discussion.

Under the unconfoundedness assumption, it can be seen that $\boldsymbol{\theta}_\pi^*$ is also a minimizer of $\mathbb{E}_P\big[\ell(X, Y; \boldsymbol{\theta}(X)) \mathbb{1}\{A = \pi(X)\}\big]$. We can thus estimate $\boldsymbol{\theta}_\pi^*$ by minimizing the empirical risk:

$$\widehat{\boldsymbol{\theta}}_\pi^{(k)} \in \underset{\boldsymbol{\theta} \in \Theta}{\operatorname{argmin}} \left\{ \frac{1}{|\mathcal{D}^{(k+1)}|} \sum_{i \in D^{(k+1)}} \mathbb{1}\{A_i = \pi(X_i)\} \cdot \ell\big(X_i, Y_i; \boldsymbol{\theta}(X_i)\big) \right\}, \tag{6}$$

where $\Theta \subseteq \{(\boldsymbol{\alpha}, \boldsymbol{\eta}) \mid \boldsymbol{\alpha} : \mathcal{X} \mapsto \mathbb{R}_{\ge 0}, \boldsymbol{\eta} : \mathcal{X} \mapsto \mathbb{R}\}$ is to be determined. In our implementation, we follow Yadlowsky et al. (2022); Jin et al. (2022b); Sahoo et al. (2022), and adopt the method of sieves (Geman & Hwang, 1982) to solve (6). Specifically, we consider an increasing sequence $\Theta_1 \subset \Theta_2 \subset \cdots$ of spaces of smooth functions, and let $\Theta = \Theta_n$ in Equation (6). For example, $\Theta_n$ can be a class of polynomials, splines, or wavelets. It has been shown in Jin et al. (2022b, Section 3.4) that under mild regularity conditions, $\widehat{\boldsymbol{\theta}}_\pi^{(k)}$ converges to $\boldsymbol{\theta}_\pi^*$ at a reasonably fast rate. For example, if $\mathcal{X} = \prod_{j=1}^d \mathcal{X}_j \subseteq \mathbb{R}^d$ for some compact intervals $\mathcal{X}_j$ and that $\boldsymbol{\theta}_\pi^*$ belongs to the Hölder class of $p$-smooth functions — with some other mild regularity conditions — then $\|\widehat{\boldsymbol{\theta}}_\pi^{(k)} - \boldsymbol{\theta}_\pi^*\|_{L_2(P_{X \mid A = \pi(X)})} = O_P((\frac{\log n}{n})^{-p/(2p+d)})$ and $\|\widehat{\boldsymbol{\theta}}_\pi^{(k)} - \boldsymbol{\theta}_\pi^*\|_{L_\infty} = O_P((\frac{\log n}{n})^{-2p^2/(2p+d)^2})$. We refer the readers to Yadlowsky et al. (2018) and Jin et al. (2022b) for more details.

## 3.2 Theoretical guarantees

We are now ready to present the theoretical guarantees for the policy value estimator $\widehat{\mathcal{V}}_\delta(\pi)$. To start, we assume the following for the convergence rates of the nuisance parameter estimators.

**Assumption 3.4** (Asymptotic estimation rate). For any policy $\pi$, assume that for each $k \in [K]$, (1) the estimators $\widehat{\pi}_0^{(k)}$ and $\widehat{g}_\pi^{(k)}$ satisfy the following for some $\gamma_1, \gamma_2 \geq 0$ and $\gamma_1 + \gamma_2 \geq \frac{1}{2}$: $\|\widehat{\pi}_0^{(k)} - \pi_0\|_{L_2(P_{X \mid A=\pi(X)})} = o_P(n^{-\gamma_1})$, $\|\widehat{g}_\pi^{(k)} - \bar{g}_\pi^{(k)}\|_{L_2(P_{X \mid A=\pi(X)})} = o_P(n^{-\gamma_2})$; (2) the empirical risk optimizer $\widehat{\boldsymbol{\theta}}_\pi^{(k)}$ satisfies $\|\widehat{\boldsymbol{\theta}}_\pi^{(k)} - \boldsymbol{\theta}_\pi^*\|_{L_2(P_{X \mid A=\pi(X)})} = o_P(n^{-\frac{1}{4}})$, $\|\widehat{\boldsymbol{\theta}}_\pi^{(k)} - \boldsymbol{\theta}_\pi^*\|_{L_\infty} = o_P(1)$.

Assumption 3.4 (1) requires either the propensity score $\pi_0$ or the conditional mean of $\widehat{G}_\pi^{(k)}(X, Y)$ is well estimated. This is a standard assumption in the double machine learning literature (Chernozhukov et al., 2018; Athey & Wager, 2021; Zhou et al., 2023; Kallus et al., 2019; 2022; Jin et al., 2022b) and can be achieved by various commonly-used machine learning methods discussed in Section 3.1. Assumption 3.4 (2) requires the optimizer $\widehat{\boldsymbol{\theta}}_\pi^{(k)}$ to be estimated at a rate faster than $n^{-1/4}$, and can be achieved by, for example, the estimators discussed in Section 3.1 under mild conditions.

The following theorem states that our estimated policy value $\widehat{\mathcal{V}}_\delta(\pi)$ is consistent for estimating $\mathcal{V}_\delta$ and is asymptotically normal. Its proof is provided in Appendix B.2.

**Theorem 3.5** (Asymptotic normality). *Suppose Assumptions 2.1, 2.4, 3.3, and 3.4 hold. For any policy $\pi : \mathcal{X} \mapsto [M]$, we have $\sqrt{n} \cdot \big(\widehat{\mathcal{V}}_\delta(\pi) - \mathcal{V}_\delta(\pi)\big) \xrightarrow{\mathrm{d}} N(0, \sigma_\pi^2)$, where*

$$\sigma_\pi^2 = \mathrm{Var}\left( \frac{\mathbb{1}\{A = \pi(X)\}}{\pi_0(A \mid X)} \cdot \big(G(X, Y) - g(X)\big) + g(X) \right);$$

$$G_\pi(x, y) = \ell(x, y; \theta_\pi^*) \text{ and } g_\pi(x) := \mathbb{E}\big[G_\pi(X, Y(\pi(X))) \mid X = x\big].$$

## 4 Policy learning under concept drift

Building on the results and methodology in Section 3, we turn to the problem of policy learning under concept drift. Given a policy class $\Pi$ and an estimated policy value $\widehat{\mathcal{V}}_\delta(\pi)$ for each $\pi \in \Pi$, it is natural to consider optimizing the estimated policy value over $\Pi$ to find the best policy. The biggest challenge here is that the quantity $\widehat{\boldsymbol{\theta}}_\pi^{(k)}$ in defining $\widehat{\mathcal{V}}_\delta(\pi)$ is not only a function of $x \in \mathcal{X}$, but also a function of $\pi \in \Pi$. The above strategy requires carrying out the ERM step in Section 3.1, for all possible policies $\pi \in \Pi$, posing major computational difficulties.

Instead of solving $\widehat{\boldsymbol{\theta}}_\pi^{(k)}$ for each $\pi \in \Pi$, we propose a computational shortcut that solves a similar ERM problem for each action $a \in [M]$. To see why this is sufficient, note that for any $\pi \in \Pi$,

$$\mathbb{E}\big[\ell(X, Y(\pi(X)); \theta) \mid X = x\big] = \sum_{a=1}^{M} \mathbb{1}\{\pi(X) = a\} \cdot \mathbb{E}[\ell(x, Y(a); \theta) \mid X = x]. \tag{7}$$

Letting $\boldsymbol{\theta}_a^*(x) \in \underset{\theta}{\mathrm{argmin}} \big\{\mathbb{E}[\ell(x, Y(a); \theta) \mid X = x]\big\}$, we can see that $\boldsymbol{\theta}_{\pi(x)}^*(x)$ is a minimizer of (7). Then, the policy learning problem reduces to finding $\pi \in \Pi$ that maximizes

$$-\mathbb{E}\left[ \boldsymbol{\alpha}_{\pi(X)}^*(X) \cdot \exp\left( -\frac{Y(\pi(X)) + \boldsymbol{\eta}_{\pi(X)}^*(X)}{\boldsymbol{\alpha}_{\pi(X)}^*(X)} - 1 \right) + \boldsymbol{\eta}_{\pi(X)}^*(X) + \boldsymbol{\alpha}_{\pi(X)}^*(X)\delta \right].$$

The following section instantiates this idea and provides a detailed policy learning algorithm.

### 4.1 The learning algorithm

The policy learning algorithm consists of two main steps: (1) solving for $\boldsymbol{\theta}_a^*$ for each $a \in [M]$ and constructing the policy value estimator $\widehat{\mathcal{V}}_\delta(\pi)$; (2) learning the optimal policy $\pi_\delta^*$ by minimizing $\widehat{\mathcal{V}}_\delta(\pi)$. As before, we randomly split the original data set $\mathcal{D}$ into $K$ folds. For each fold $k \in [K]$,

---

**Algorithm 2** Policy learning under concept drift

---

**Input:** Dataset $\mathcal{D}$; policy class $\Pi$; uncertainty set parameter $\delta$; propensity score estimation algorithm $\mathcal{C}$; ERM algorithm $\mathcal{E}(\cdot)$ for obtaining $\boldsymbol{\theta}_a^*$; regression algorithm $\mathcal{R}$ for estimating $\bar{g}_a$.

Randomly split $\mathcal{D}$ into $K$ equal-sized folds;
**for** $k = 1, \ldots, K$ **do**
    $\widehat{\pi}_0^{(k)} \leftarrow \mathcal{C}(\mathcal{D}^{(k+1)})$,
    **for** $a = 1, \cdots, M$ **do**
        $\widehat{\boldsymbol{\theta}}_a^{(k)} \leftarrow \mathcal{E}(\mathcal{D}^{(k+1)})$, $\widehat{g}_a^{(k)} \leftarrow \mathcal{R}(X_i, A_i, \widehat{G}_a^{(k)}(X_i, Y_i); i \in \mathcal{D}^{(k+2)})$;
    **end for**
**end for**

**Return:** $\widehat{\pi}_{\mathrm{LN}}$ that maximizes $\widehat{\mathcal{V}}_\delta^{\mathrm{LN}}(\pi)$ as in Equation (8).

---

we use samples in the $(k+1)$-th data fold $\mathcal{D}^{(k+1)}$ to obtain the propensity estimator $\widehat{\pi}_0^{(k)}(a \mid \cdot)$ (by regression) and the optimizer $\boldsymbol{\theta}_a^*$ (by ERM) for each $a \in [M]$. Next, for each $a \in [M]$, define

$$G_a(x, y) = \ell(x, y; \boldsymbol{\theta}_a^*(x)), \ \widehat{G}_a^{(k)}(x, y) = \ell(x, y; \widehat{\boldsymbol{\theta}}_a^{(k)}(x)), \ \text{and} \ \bar{g}_a^{(k)}(x) = \mathbb{E}\big[\widehat{G}_a^{(k)}(X, Y(a)) \mid X = x\big].$$

We then obtain an estimator $\widehat{g}_a^{(k)}$ for $\bar{g}_a^{(k)}$ by regressing $\widehat{G}_a^{(k)}(X_i, Y_i)$ onto $X_i$ with $i \in \mathcal{D}^{(k+2)}$. Finally, we obtain the learned policy by maximizing the estimated policy value: $\widehat{\pi}_{\mathrm{LN}} = \underset{\pi \in \Pi}{\operatorname{argmax}} \ \widehat{\mathcal{V}}_\delta^{\mathrm{LN}}(\pi)$,

where $\widehat{\mathcal{V}}_\delta^{\mathrm{LN}}(\pi) := -\frac{1}{K} \sum_{k=1}^K \widehat{\mathcal{V}}_\delta^{\mathrm{LN},(k)}(\pi)$, and

$$\widehat{\mathcal{V}}_\delta^{\mathrm{LN},(k)}(\pi) = \frac{1}{|\mathcal{D}^{(k)}|} \sum_{i \in \mathcal{D}^{(k)}} \frac{\mathbb{1}\{A_i = \pi(X_i)\}}{\widehat{\pi}_0^{(k)}(A_i \mid X_i)} \cdot \big(\widehat{G}_{\pi(X_i)}^{(k)}(X_i, Y_i) - \widehat{g}_{\pi(X_i)}^{(k)}(X_i)\big) + \widehat{g}_{\pi(X_i)}^{(k)}(X_i).$$

$$(8)$$

The above optimization problem can be solved efficiently by first-order optimization methods or policy tree search as in Zhou et al. (2023); we shall elaborate on the implementation in Section 5. The complete policy learning procedure is summarized in Algorithm 2.

## 4.2 REGRET UPPER BOUND

In this section, we present the regret analysis of $\widehat{\pi}_{\mathrm{LN}}$ obtained by Algorithm 2 (recall the definition of regret) Before we embark on the formal analysis, we introduce the Hamming entropy integral $\kappa(\Pi)$, which measures the complexity of $\Pi$.

**Definition 4.1.** Given a policy class $\Pi$ and $n$ data points $\{x_1, \ldots, x_n\} \subseteq \mathcal{X}$, the *Hamming distance* between $\pi, \pi' \in \Pi$ is $d_H(\pi, \pi') := \frac{1}{n} \sum_{i=1}^n \mathbb{1}\{\pi(x_i) \neq \pi'(x_i)\}$. The $\varepsilon$-*covering number* of $\{x_1, \ldots, x_n\}$, denoted as $\mathcal{C}(\epsilon, \Pi; \{x_1, \ldots, x_n\})$, is the smallest number $L$ of policies $\{\pi_1, \ldots, \pi_L\}$ in $\Pi$, such that $\forall \ \pi \in \Pi$, $\exists \ \pi'_\ell$ such that $d_H(\pi, \pi_\ell) \leq \epsilon$. Denote $N_H(\epsilon, \Pi) := \sup_{n \geq 1} \sup_{x_1, \ldots, x_n} \mathcal{C}(\epsilon, \Pi; \{x_1, \ldots, x_n\})$. The *Hamming entropy integral* of $\Pi$ is defined as $\kappa(\Pi) := \int_0^1 \sqrt{\log N_H(\epsilon^2, \Pi)} \, d\epsilon$.

The following theorem provides a regret upper bound for the policy learned by Algorithm 2.

**Theorem 4.2.** *Suppose Assumptions 2.1, 2.4, 3.3, 3.4 hold. For any $\beta \in (0, 1)$, there exists $N \in \mathbb{N}_+$ such that when $n \geq N$, we have with probability at least $1 - \beta$ that*

$$\mathcal{R}_\delta(\widehat{\pi}_{\mathrm{LN}}) \leq \frac{C_0(\bar{\alpha}, \underline{\alpha}, \bar{\eta}, \delta, \varepsilon)}{\sqrt{n}}\big(65 + 8\kappa(\Pi) + \sqrt{\log(1/\beta)}\big),$$

*where $C_0(\bar{\alpha}, \underline{\alpha}, \bar{\eta}, \delta, \varepsilon) := 6(\bar{\alpha} \cdot \exp(\bar{\eta}/\underline{\alpha} - 1) + \bar{\eta} + \bar{\alpha}\delta)/\varepsilon$.*

The proof of Theorem 4.2 is deferred to Appendix B.3. At a high level, we decompose the regret and upper bound it with the supremum of the estimation error of policy values. which can be upper bounded by establishing uniform convergence results for the policy value estimators. Through a careful chaining argument, we show that the dependence of $\mathcal{R}(\widehat{\pi}_{\mathrm{LN}})$ on $n$ is of the order $O(n^{-\frac{1}{2}})$, which is sharper than the $O(n^{-\frac{1}{2}} \log n)$ dependence for that of Mu et al. (2022) by a logarithmic factor. We also note that both regrets are asymptotic in $n$ and hold for sufficiently large $n$.

## 4.3 REGRET LOWER BOUND

In this section, we complement the regret upper bound in Theorem 4.2 with a minimax lower bound that characterizes the fundamental difficulty of policy learning under concept drift. Our lower bound is stated in terms of the Natarajan dimension (Natarajan, 1989), defined as follows.

**Definition 4.3** (Natarajan dimension). Given an $M$-action policy class $\Pi$, we say a set of $m$ points $\{x_1, \ldots, x_m\}$ is shattered by $\Pi$ if there exist two functions $f_{-1}, f_1 : \{x_1, \ldots, x_m\} \mapsto [M]$ such that (i) $f_{-1}(x_j) \neq f_1(x_j)$ for all $j \in [m]$; (ii) for any $\sigma \in \{\pm 1\}^m$, there exists a policy $\pi \in \Pi$ such that for any $j \in [m]$, $\pi(x_j) = f_{\sigma_j}(x_j)$. The Natarajan dimension of $\Pi$, denoted by $\mathrm{Ndim}(\Pi)$, is defined to be the size of the largest set shattered by $\Pi$.

*Remark* 4.4 (Connection to other complexity measures). The Natarajan dimension generalizes the Vapnik-Chervonenkis (VC) dimension (Vapnik & Chervonenkis, 2015) to the multi-class classification setting. The Natarajan dimension is also closely related to the Hamming entropy integral $\kappa(\Pi)$ in our upper bound, as $\kappa(\Pi) = O(\sqrt{\log(d)\mathrm{Ndim}(\Pi)})$ (Cai et al., 2020).

**Theorem 4.5.** *Let $\mathcal{P}$ denote the set of all distributions of $(X, A, Y(1), \ldots, Y(M))$ that satisfy Assumption 2.1, 2.4, 3.3, and 3.4.[4] Suppose that $\delta \leq 0.2$, $n \geq \mathrm{Ndim}(\Pi)^2$, and $\mathrm{Ndim}(\Pi) \geq 4/(9\varepsilon)$. For any policy leaning algorithm that outputs $\widehat{\pi}$ as a function of $\{(X_i, A_i, Y_i)\}_{i=1}^n$, there is*

$$\sup_{P \in \mathcal{P}} \mathbb{E}_{P^n}[\mathcal{R}(\widehat{\pi})] \geq \frac{1}{120}\sqrt{\frac{\mathrm{Ndim}(\Pi)}{n\varepsilon}}.$$

The proof of Theorem 4.5 is provided in Appendix B.4. Theorem 4.5 implies that for any learning algorithm, there exists a problem instance such that the regret scales as $\Omega(\sqrt{\mathrm{Ndim}(\Pi)/n})$. Recalling the relationship between the Natarajan dimension and the Hamming entropy integral in the remark above, we see that our proposed algorithm achieves the minimax rate in the sample size and the policy class complexity up to logarithmic factors.

## 5 NUMERICAL RESULTS

We evaluated Algorithm 2 in a simulated setting against the benchmark SNLN in Si et al. (2023).

**Data generating process.** Our data generating process follows that of the linear boundary example in Si et al. (2023). We let the context set $\mathcal{X} = \{x \in \mathbb{R}^5 : \|x\|_2 \leq 1\}$ to be the closed unit ball of $\mathbb{R}^5$ and let the action set to be $\mathcal{A} = \{1, 2, 3\}$; the rewards $Y(a)$'s are mutually independent conditioned on $X$ with $Y(a) \mid X \sim N(\beta_a^\top X, \sigma_a^2)$, for $a \in [3]$. The training dataset $\mathcal{D}_{\text{train}} = \{(X_i, Y_i(\pi_0(X_i)))\}_{i=1}^n$ are generated with a given behavior policy $\pi_0$ (unknown to policy learning algorithms). The testing datasets $\mathcal{D}_{\text{test}}$ are i.i.d. draws of data tuple $\{(X_i, Y_i(1), Y_i(2), Y_i(3))\}_{i=1}^n$. The specific values of $\{\beta_1, \beta_2, \beta_3\} \in \mathbb{R}^5$ and $\{\sigma_1^2, \sigma_2^2, \sigma_3^2\} \in \mathbb{R}_+$ as well as the behavior policy $\pi_0$ are given in Appendix A.

**Implementation.** In our implementation, the number of splits is taken to be $K = 3$. We use the Random Forest regressor from the `scikit-learn` Python library to estimate $\widehat{\pi}_0$ and $\widehat{g}$. For estimating $\boldsymbol{\theta}^*$, we adopt the cubic spline method and employ the Nelder-Mead optimization method in `SciPy` Python library (Virtanen et al., 2020) to optimize the coefficients in the spline approximation, where the obtained estimator has threshold at 0.001 to guarantee Proposition 2.5. Finally, we optimize and find $\widehat{\pi}_{\text{LN}}$ with `policytree` (Athey & Wager, 2021).

The benchmark algorithm SNLN is adapted from Si et al. (2023, Algorithm 2). Since Si et al. (2023, Algorithm 2) is designed for joint distribution shift formulation, we revised the original algorithm to fit our concept drift setting. It is well-known that the chain rule of KL-divergence (Cover, 1999) gives

$$D_{\text{KL}}(Q_{X,Y} \| P_{X,Y}) = D_{\text{KL}}(Q_X \| P_X) + D_{\text{KL}}(Q_{Y \mid X} \| P_{Y \mid X}). \tag{9}$$

Therefore, given any uncertainty set radius $\delta$ and known covariate shift (in this experiment, we assume no covariate shift), Si et al. (2023, Algorithm 2) can be used to implement policy learning under concept drift. Note that SNLN admits known propensity scores. As we only consider the case where the propensity scores are unknown, we complement Si et al. (2023, Algorithm 2) with

---

[4]When we say a distribution $P$ satisfies Assumption 3.4, we mean that under $P$ there exist $\widehat{\boldsymbol{\theta}}$, $\widehat{\pi}_0$, and $\widehat{g}$ that satisfy the convergence rates in Assumption 3.4.

estimated propensity scores from Random Forest Regressor in `scikit-learn`, the same way as in the implementation of Algorithm 2. The additional setup details are in Appendix A.

**Evaluation.** For a learnt policy $\widehat{\pi}$, we evaluate its performance by the following two performance metrics. (i) Using the testing dataset according to $\widehat{\pi}$: $\mathcal{D}_{\text{test},\widehat{\pi}} = \{(X_i, Y_i(\widehat{\pi}(X_i)))\}$, we estimate the policy value $\mathcal{V}^*_\delta(\widehat{\pi})$ by the empirical policy value $\bar{\mathcal{V}}_\delta(\widehat{\pi})$ where the nuisance parameters $\boldsymbol{\alpha}_{\widehat{\pi}}(X_i), \boldsymbol{\eta}_{\widehat{\pi}}(X_i)$ are found via the cubic spline and Nelder-Mead optimization method. (ii) For 100 testing dataset each containing 10000 data points $\left\{\{(X_i^{(j)}, Y_i^{(j)}(1), Y_i^{(j)}(2), Y_i^{(j)}(3))\}_{i=1}^{10000}\right\}_{j=1}^{100}$, we randomly sample a new testing dataset $\left\{\{(X_i^{(j)}, \tilde{Y}_i^{(j)}(1), \tilde{Y}_i^{(j)}(2), \tilde{Y}_i^{(j)}(3))\}_{i=1}^{10000}\right\}_{j=1}^{100}$ on the KL-sphere centered at $(Y_i^{(j)}(1), Y_i^{(j)}(2), Y_i^{(j)}(3))$ with radius $\delta$. Then we evaluate $\widehat{\pi}$ using

$$\tilde{\mathcal{V}}_\delta^{\min}(\widehat{\pi}) := \min_{1 \le j \le 100} \left\{ \frac{1}{10000} \sum_{i=1}^{10000} \tilde{Y}_i^{(j)}\big(\widehat{\pi}(X_i^{(j)})\big) \right\}.$$

This simulates a more realistic scenario by mimicking real-world concept drifts.

**Results.** Table 2 and 3 report the values $\bar{\mathcal{V}}_\delta, \tilde{\mathcal{V}}_\delta^{\min}$ (with 95% confidence intervals) of the learnt policies $\widehat{\pi}_{\text{LN}}$ and $\widehat{\pi}_{\text{SNLN}}$, by Algorithm 2 and Si et al. (2023, Algorithm 2) respectively. Table 2 shows that $\widehat{\pi}_{\text{LN}}$ outperforms the benchmark $\widehat{\pi}_{\text{SNLN}}$ consistently, with higher policy values and similar 95% confidence intervals. With a higher $\delta$, the policy values of $\widehat{\pi}_{\text{LN}}, \widehat{\pi}_{\text{SNLN}}$ are smaller, due to a bigger uncertainty set. Table 3 shows that $\widehat{\pi}_{\text{LN}}$ achieves higher worst-case rewards than $\widehat{\pi}_{\text{SNLN}}$ does, in a more realistic setting with concept drift testing datasets. Together, we see that $\widehat{\pi}_{\text{LN}}$ succeeds in finding a better policy under concept drift; while the performance of $\widehat{\pi}_{\text{SNLN}}$ is comprised by its conservative policy learning process, in which it considers joint distributional shifts even though it is given the information that no covariate shifts took place.

The results align with the intuition that Algorithm 2 admits a subset of the uncertainty set that the benchmark algorithm SNLN considers, as explained in Equation (9). Consequently, $\mathcal{V}_\delta(\widehat{\pi}_{\text{SNLN}})$ is a lower bound of $\mathcal{V}_\delta(\widehat{\pi}_{\text{LN}})$ in theory, and by the results in Table 2, in practice. In real-world applications, knowing the source of the distribution shift effectively shrinks the uncertainty set, thereby yielding less conservative results. Since it is fairly easy to identify covariate shifts (comparing to detecting concept drift), when the decision maker observes none or little covariate shifts and would like to hedge against the risk of concept drift, it is suitable to apply our method which outperforms existing method designed for learning under the joint distributional shift.

In Appendix A, we also provide simulation results of Algorithm 1 for a fixed target policy, which show that Algorithm 1 can estimate the distributionally robust policy value under concept drift efficiently.

|  |  | $n = 7500$ | $n = 13500$ | $n = 16500$ | $n = 19500$ |
|---|---|---|---|---|---|
| $\bar{\mathcal{V}}_{0.05}$ | $\widehat{\pi}_{\text{LN}}$ | $0.2272 \pm 0.002$ | $0.2299 \pm 0.001$ | $0.2303 \pm 0.001$ | $0.2310 \pm 0.001$ |
|  | $\widehat{\pi}_{\text{SNLN}}$ | $0.0554 \pm 0.005$ | $0.0589 \pm 0.004$ | $0.0617 \pm 0.004$ | $0.0664 \pm 0.003$ |
| $\bar{\mathcal{V}}_{0.1}$ | $\widehat{\pi}_{\text{LN}}$ | $0.1579 \pm 0.007$ | $0.1662 \pm 0.002$ | $0.1663 \pm 0.002$ | $0.1678 \pm 0.002$ |
|  | $\widehat{\pi}_{\text{SNLN}}$ | $0.0548 \pm 0.004$ | $0.0580 \pm 0.004$ | $0.0583 \pm 0.003$ | $0.0616 \pm 0.004$ |
| $\bar{\mathcal{V}}_{0.2}$ | $\widehat{\pi}_{\text{LN}}$ | $0.0781 \pm 0.003$ | $0.0802 \pm 0.002$ | $0.0804 \pm 0.002$ | $0.0831 \pm 0.002$ |
|  | $\widehat{\pi}_{\text{SNLN}}$ | $0.0182 \pm 0.003$ | $0.0183 \pm 0.003$ | $0.0200 \pm 0.003$ | $0.0219 \pm 0.003$ |

Table 2: Empirical robust policy value $\bar{\mathcal{V}}_\delta$ of policies $\widehat{\pi}_{\text{LN}}, \widehat{\pi}_{\text{SNLN}}$ learned by Algorithm 2 and SNLN respectively, under $\delta = 0.05, 0.1, 0.2$, each with 50 trials.

|  |  | $n = 7500$ | $n = 13500$ | $n = 16500$ | $n = 19500$ |
|---|---|---|---|---|---|
| $\tilde{\mathcal{V}}_{0.1}^{\min}$ | $\widehat{\pi}_{\text{LN}}$ | $0.2075 \pm 0.015$ | $0.2139 \pm 0.005$ | $0.2149 \pm 0.007$ | $0.2167 \pm 0.003$ |
|  | $\widehat{\pi}_{\text{SNLN}}$ | $0.1884 \pm 0.007$ | $0.2009 \pm 0.008$ | $0.2017 \pm 0.006$ | $0.2020 \pm 0.004$ |

Table 3: Empirical worst case policy reward on the KL-sphere $\tilde{\mathcal{V}}_\delta^{\min}$ of policies $\widehat{\pi}_{\text{LN}}, \widehat{\pi}_{\text{SNLN}}$ learned by Algorithm 2 and SNLN respectively, under $\delta = 0.1$, each with 20 trials.

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

## A    EXPERIMENT DETAILS

In the data generating process, we choose $\beta$'s and $\sigma$'s to be

$$\beta_1 = (1, 0, 0, 0, 0), \quad \beta_2 = (-1/2, \sqrt{3}/2, 0, 0, 0), \quad \beta_3 = (-1/2, -\sqrt{3}/2, 0, 0, 0); \quad \sigma = (0.2, 0.5, 0.8).$$

The underlying policy $\pi_0$ chooses actions with context $x$ according to the following rules:

$$(\pi_0(1 \mid x), \pi_0(2 \mid x), \pi_0(3 \mid x)) = \begin{cases} (0.5, 0.25, 0.25), & \text{if } \underset{i=1,2,3}{\arg\max}\{\beta_i^\top x\} = 1, \\ (0.25, 0.5, 0.25), & \text{if } \underset{i=1,2,3}{\arg\max}\{\beta_i^\top x\} = 2, \\ (0.25, 0.25, 0.5), & \text{if } \underset{i=1,2,3}{\arg\max}\{\beta_i^\top x\} = 3. \end{cases}$$

We generate $\mathcal{D}_{\text{train}}$ according to the procedure described above as the training dataset. We also generate 10,000 samples as our testing dataset $\mathcal{D}_{\text{test}} = \{i \in [10,000] : (X_i, Y_i(1), Y_i(2), Y_i(3))\}$, which we use to estimate the true policy value.

We present the result of the policy estimation experiments in Figure 1, using Algorithm 1 with inputs of the training datasets and the target policy $\pi$

$$\pi(x) = \begin{cases} 1, & \text{if } \|x\|_2 \in [0, 1/3], \\ 2, & \text{if } \|x\|_2 \in [1/3, 2/3], \\ 3, & \text{if } \|x\|_2 \in [2/3, 1]. \end{cases}$$

The underlying true policy value is obtained by the testing dataset $\mathcal{D}_{\text{test}}$. Similar to the learning experiment, we repeat the estimation experiment over 50 seeds. Figure 1 shows that as the sample size increases, the estimated policy value by Algorithm 1 is more accurate and stable.

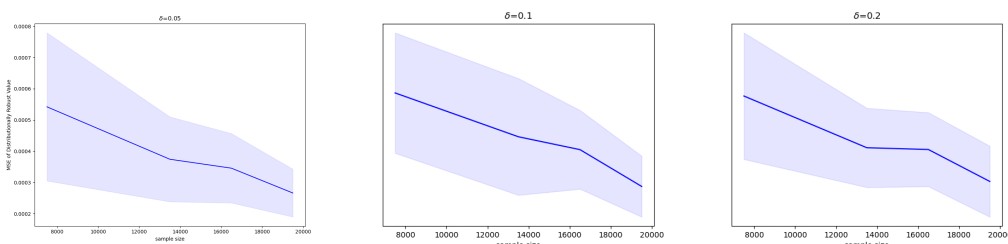

Figure 1: The Mean Square Error (MSE) of the estimated policy value by Algorithm 1. The $x$-axis is the number of samples used by Algorithm 1, and the $y$-axis is the mean squared error (MSE) of the policy value estimator.

**Computation details.**    The experiments were run on the following cloud servers: (i) an Intel Xeon Platinum 8160 @ 2.1 GHz with 766GB RAM and 96 CPU x 2.1 GHz; (ii) an Intel Xeon Platinum 8160 @ 2.1 GHz with 1.5TB RAM and 96 CPU x 2.1 GHz; (iii) an Intel Xeon Gold 6132 @ 2.59 GHz with 768GB RAM and 56 CPU x 2.59 GHz and (iv) an Intel Xeon GPU E5-2697A v4 @ 2.59 GHz with 384GB RAM and 64 CPU x 2.59 GHz.

## B    DEFERRED PROOFS OF THE MAIN RESULTS

### B.1    PROOF OF LEMMA 2.3

Fix $\pi \in \Pi$ and $x \in \mathcal{X}$. Letting $L = \frac{dQ_{Y \mid X=x}}{dP_{Y \mid X=x}}$, we can rewrite the inner minimization in Equation (1) as

$$\inf_{L \text{ measurable}} \mathbb{E}_{P_{Y \mid X}}[Y(\pi(x))L \mid X = x]$$
$$\text{s.t. } \mathbb{E}_{P_{Y \mid X}}[L \mid X = x] = 1, \tag{10}$$
$$\mathbb{E}_{P_{Y \mid X}}[f_{\text{KL}}(L) \mid X = x] \le \delta,$$

where the function $f_{\text{KL}}(x) = x \log x$ represents the KL divergence function. In (10), the first constraint reflects that $L$ is an likelihood ratio, and the second constraint corresponds to the KL divergence bound.

For notational simplicity, let $\mathbb{E}_x$ be the shorthand of $\mathbb{E}_{P_{Y \mid X}}[\cdot \mid X = x]$. By Theorem 8.6.1 of Luenberger (1997), the Slater's condition is satisfied and strong duality holds:

$$\inf_{\substack{\mathbb{E}_x[L]=1, \\ \mathbb{E}_x[f_{\text{KL}}(L)] \le \delta}} \mathbb{E}_x\big[Y(\pi(x))L\big] = \max_{\alpha \ge 0, \eta \in \mathbb{R}} \varphi(\alpha, \eta, x), \tag{11}$$

where

$$\varphi(\alpha, \eta, x) = \inf_{L \ge 0} \mathcal{L}(\alpha, \eta, L, x),$$

$$\mathcal{L}(\alpha, \eta, L, x) = \mathbb{E}_x[Y(\pi(x))L] + \eta \cdot \big(\mathbb{E}_x[L] - 1\big) + \alpha \cdot \big(\mathbb{E}_x[f_{\text{KL}}(L)] - \delta\big)$$
$$= \mathbb{E}_x\big[Y(\pi(x))L + \eta(L-1) + \alpha(f_{\text{KL}}(L) - \delta)\big].$$

We can explicitly work out the minimum of $\mathcal{L}(\alpha, \eta, L, x)$, and we have

$$\varphi(\alpha, \eta, x) = \mathbb{E}_x\bigg[-\alpha f_{\text{KL}}^*\bigg(-\frac{Y(\pi(x)) + \eta}{\alpha}\bigg) - \eta - \alpha\delta\bigg],$$

where $f_{\text{KL}}^*(y) = \exp(y-1)$ is the conjugate function of $f_{\text{KL}}$. Using Equation (11), we arrive at

$$\inf_{\substack{\mathbb{E}_x[L]=1, \\ \mathbb{E}_x[f_{\text{KL}}(L)] \le \delta}} \mathbb{E}_x\big[Y(\pi(x))L\big] = -\min_{\alpha \ge 0, \eta \in \mathbb{R}} \mathbb{E}_x\bigg[\alpha \exp\bigg(-\frac{Y(\pi(x)) + \eta}{\alpha} - 1\bigg) + \eta + \alpha\delta\bigg].$$

The proof is thus completed.

## B.2 Proof of Theorem 3.5

For notational simplicity, we drop the dependence on $P$ in $\mathbb{E}_P$ when the context is clear. The proof of Theorem 3.5 makes use of the following lemma, which establishes some useful properties of the optimizer $\boldsymbol{\theta}_\pi^*$. The proof of Lemma B.1 can be found in Appendix C.1.

**Lemma B.1.** *For any policy $\pi$, assume that Assumption 3.3 holds. We have the following properties of the optimizer $\boldsymbol{\theta}_\pi^*$.*

*(1)* $\mathbb{E}\big[\nabla_\theta \ell(x, Y(\pi(x)); \theta) \mid X = x\big] = 0$ *at $\theta = \boldsymbol{\theta}_\pi^*(x)$ for any $x \in \mathcal{X}$.*

*(2) There exists a constant $\xi > 0$ such that for any $x$ and $\theta$ satisfying $\|\theta - \boldsymbol{\theta}_\pi^*(x)\|_2 \le \xi$,*

$$\big|\ell(x, y; \theta) - \ell(x, y; \boldsymbol{\theta}_\pi^*(x)) - \nabla_\theta \ell(x, y; \boldsymbol{\theta}_\pi^*(x))^\top (\theta - \boldsymbol{\theta}_\pi^*(x))\big| \le \bar{\ell}(x, y) \cdot \big\|\theta - \boldsymbol{\theta}_\pi^*(x)\big\|_2^2,$$

*for some function $\bar{\ell}(x, y)$ such that $\sup_{x \in \mathcal{X}} \mathbb{E}[\bar{\ell}(x, Y(\pi(x))) \mid X = x] < L$ for some $L > 0$.*

*(3) There exists a constant $\xi_1 > 0$ such that for any $\boldsymbol{\theta}$ satisfying $\|\boldsymbol{\theta} - \boldsymbol{\theta}_\pi^*\|_{L_\infty} \le \xi_1$.*

$$\big\|\ell(X, Y(\pi(X)); \boldsymbol{\theta}(X)) - \ell(X, Y(\pi(X)); \boldsymbol{\theta}_\pi^*(X))\big\|_{L_2(P_{X,Y(\pi(X)) \mid A=\pi(X)})} \le C_\ell \|\boldsymbol{\theta} - \boldsymbol{\theta}_\pi^*\|_{L_2(P_{X \mid A=\pi(X)})},$$

*for some constant $C_\ell > 0$.*

We proceed to show the asymptotic normality of $\widehat{\mathcal{V}}_\delta(\pi)$. For each $k \in [K]$, we first define the following oracle quantity:

$$\mathcal{V}_\delta^{*(k)}(\pi) = \frac{1}{|\mathcal{D}^{(k)}|} \sum_{i \in \mathcal{D}^{(k)}} \frac{\mathbb{1}\{\pi(X_i) = A_i\}}{\pi_0(A_i \mid X_i)} \cdot \big(G_\pi(X_i, Y_i) - g_\pi(X_i)\big) + g_\pi(X_i).$$

In the sequel, we shall show that $\widehat{\mathcal{V}}_\delta^{(k)}(\pi) = \mathcal{V}_\delta^{*(k)}(\pi) + o_p(n^{-\frac{1}{2}})$. We begin by decomposing the difference between $\widehat{\mathcal{V}}_\delta^{(k)}(\pi)$ and $\mathcal{V}_\delta^{*(k)}$:

$$\widehat{\mathcal{V}}_\delta^{(k)}(\pi) - \mathcal{V}_\delta^{*(k)}(\pi)$$

$$= \frac{1}{|\mathcal{D}^{(k)}|} \sum_{i \in \mathcal{D}^{(k)}} \left[ \frac{\mathbb{1}\{\pi(X_i) = A_i\}}{\widehat{\pi}_0^{(k)}(A_i \mid X_i)} \cdot \left( \widehat{G}_\pi^{(k)}(X_i, Y_i) - \widehat{g}_\pi^{(k)}(X_i) \right) - \frac{\mathbb{1}\{\pi(X_i) = A_i\}}{\pi_0(A_i \mid X_i)} \cdot \left( G_\pi(X_i, Y_i) - g_\pi(X_i) \right) \right]$$

$$+ \frac{1}{|\mathcal{D}^{(k)}|} \sum_{i \in \mathcal{D}^{(k)}} \left( \widehat{g}_\pi^{(k)}(X_i) - g_\pi(X_i) \right)$$

$$= \underbrace{\frac{1}{|\mathcal{D}^{(k)}|} \sum_{i \in \mathcal{D}^{(k)}} \frac{\mathbb{1}\{A_i = \pi(X_i)\}}{\pi_0(A_i \mid X_i)} \cdot \left( \widehat{G}_\pi^{(k)}(X_i, Y_i) - G_\pi(X_i, Y_i) \right)}_{\text{(I)}}$$

$$- \underbrace{\frac{1}{|\mathcal{D}^{(k)}|} \sum_{i \in \mathcal{D}^{(k)}} \left( \frac{\mathbb{1}\{A_i = \pi(X_i)\}}{\widehat{\pi}_0^{(k)}(A_i \mid X_i)} - \frac{\mathbb{1}\{A_i = \pi(X_i)\}}{\pi_0(A_i \mid X_i)} \right) \cdot \left( \widehat{g}_\pi^{(k)}(X_i) - \bar{g}_\pi^{(k)}(X_i) \right)}_{\text{(II)}}$$

$$+ \underbrace{\frac{1}{|\mathcal{D}^{(k)}|} \sum_{i \in \mathcal{D}^{(k)}} \left( \frac{\mathbb{1}\{A_i = \pi(X_i)\}}{\widehat{\pi}_0^{(k)}(A_i \mid X_i)} - \frac{\mathbb{1}\{A_i = \pi(X_i)\}}{\pi_0(A_i \mid X_i)} \right) \cdot \left( \widehat{G}_\pi^{(k)}(X_i, Y_i) - \bar{g}_\pi^{(k)}(X_i) \right)}_{\text{(III)}}$$

$$- \underbrace{\frac{1}{|\mathcal{D}^{(k)}|} \sum_{i \in \mathcal{D}^{(k)}} \frac{\mathbb{1}\{A_i = \pi(X_i)\}}{\pi_0(A_i \mid X_i)} \cdot \left( \widehat{g}_\pi^{(k)}(X_i) - g_\pi(X_i) \right) + \frac{1}{|\mathcal{D}^{(k)}|} \sum_{i \in \mathcal{D}^{(k)}} \left( \widehat{g}_\pi^{(k)}(X_i) - g_\pi(X_i) \right)}_{\text{(IV)}}.$$

**Bounding term (I).** Recall that $\boldsymbol{\theta}_\pi^*(x)$ is the minimizer of

$$\mathbb{E}\Big[ \ell\big(x, Y(\pi(x)); \theta\big) \,\big|\, X = x \Big].$$

By the first-order condition established in part (1) of Lemma B.1, we have

$$\mathbb{E}\Big[ \nabla_\theta \ell\big(x, Y(\pi(x)); \boldsymbol{\theta}(x)\big) \,\big|\, X = x \Big] = 0. \tag{12}$$

For any $i \in \mathcal{D}^{(k)}$, by the unconfoundedness condition in Assumption 2.1, we have

$$\mathbb{E}\left[ \frac{\mathbb{1}\{A_i = \pi(X_i)\}}{\pi_0(A_i \mid X_i)} \cdot \left( \widehat{G}_\pi^{(k)}(X_i, Y_i) - G_\pi(X_i, Y_i) \right) \,\bigg|\, \mathcal{D}^{(-k)} \right]$$

$$= \mathbb{E}\left[ \frac{\mathbb{1}\{A_i = \pi(X_i)\}}{\pi_0(A_i \mid X_i)} \cdot \left( \widehat{G}_\pi^{(k)}(X_i, Y_i(\pi(X_i))) - G_\pi(X_i, Y_i(\pi(X_i))) \right) \,\bigg|\, \mathcal{D}^{(-k)} \right]$$

$$= \mathbb{E}\left[ \widehat{G}_\pi^{(k)}(X_i, Y_i(\pi(X_i))) - G_\pi(X_i, Y_i(\pi(X_i))) \,\big|\, \mathcal{D}^{(-k)} \right]$$

$$= \mathbb{E}\left[ \ell\big(X_i, Y_i(\pi(X_i)); \widehat{\boldsymbol{\theta}}_\pi^{(k)}(X_i)\big) - \ell\big(X_i, Y_i(\pi(X_i)); \boldsymbol{\theta}_\pi^*(X_i)\big) - \nabla_\theta \ell\big(X_i, Y(\pi(X_i)); \boldsymbol{\theta}_\pi^*(X_i)\big) \,\big|\, \mathcal{D}^{(-k)} \right],$$

where the last step is due to Equation (12). By Assumption 3.4, $\|\widehat{\boldsymbol{\theta}}_\pi^{(k)} - \boldsymbol{\theta}_\pi^*\|_{L_\infty} = o_P(1)$. Therefore, for any $\beta \in (0, 1)$, there exists $N \in \mathbb{N}_+$ such that for $n \geq N$, $\|\widehat{\boldsymbol{\theta}}_\pi^{(k)} - \boldsymbol{\theta}_\pi^*\|_{L_\infty} \leq \min(\xi, \xi_1)$. On the event that $\|\widehat{\boldsymbol{\theta}}_\pi^{(k)}(x) - \boldsymbol{\theta}_\pi^*(x)\|_{L_\infty} \leq \min(\xi, \xi_1)$ by part (2) of Lemma B.1 and Jensen's inequality, we have

$$\left| \mathbb{E}\left[ \frac{\mathbb{1}\{A_i = \pi(X_i)\}}{\pi_0(A_i \mid X_i)} \cdot \left( \widehat{G}_\pi^{(k)}(X_i, Y_i) - G_\pi(X_i, Y_i) \right) \right] \,\bigg|\, \mathcal{D}^{(-k)} \right|$$

$$\leq \mathbb{E}\left[ \left| \ell(X_i, Y_i(\pi(X_i)); \widehat{\boldsymbol{\theta}}_\pi^{(k)}(X_i)) - \ell(X_i, Y_i(\pi(X_i)); \boldsymbol{\theta}_\pi^*(X_i)) - \nabla_\theta \ell\big(X_i, Y(\pi(X_i)); \boldsymbol{\theta}_\pi^*(X_i)\big) \right| \,\bigg|\, \mathcal{D}^{(-k)} \right]$$

$$\leq \mathbb{E}\left[ \bar{\ell}(X_i, Y_i) \cdot \left\| \widehat{\boldsymbol{\theta}}_\pi^{(k)}(X_i) - \boldsymbol{\theta}_\pi^*(X_i) \right\|_2^2 \right] \leq L\mathbb{E}\left[ \left\| \widehat{\boldsymbol{\theta}}_\pi^{(k)}(X_i) - \boldsymbol{\theta}_\pi^*(X_i) \right\|_2^2 \,\big|\, \mathcal{D}^{(-k)} \right] = L\left\| \widehat{\boldsymbol{\theta}}_\pi^{(k)} - \boldsymbol{\theta}_\pi^* \right\|_{L_2(P_X)}^2.$$

By Chebyshev's inequality, we have for any $t > 0$ that

$$\mathbb{P}\Bigg(\Bigg|\frac{1}{|\mathcal{D}^{(k)}|}\sum_{i\in\mathcal{D}^{(k)}}\frac{\mathbb{1}\{A_i = \pi(X_i)\}}{\pi_0(A_i\,|\,X_i)}\cdot\big(\widehat{G}_\pi^{(k)}(X_i, Y_i) - G_\pi(X_i, Y_i)\big)$$

$$-\,\mathbb{E}\left[\frac{\mathbb{1}\{A = \pi(X)\}}{\pi_0(A\,|\,X)}\cdot\big(\widehat{G}_\pi^{(k)}(X, Y) - G_\pi(X, Y)\big)\,\bigg|\,\mathcal{D}^{(-k)}\right]\Bigg| \geq t\,\bigg|\,\mathcal{D}^{(-k)}\Bigg)$$

$$\leq\frac{1}{|\mathcal{D}^{(k)}|t^2}\mathrm{Var}\left(\frac{\mathbb{1}\{A = \pi(X)\}}{\pi_0(A\,|\,X)}\cdot\Big[\widehat{G}_\pi^{(k)}(X, Y) - G_\pi(X, Y)\Big]\right)$$

$$\leq\frac{\big\|\widehat{G}_\pi^{(k)} - G_\pi\big\|^2_{L_2(P_{X,Y\,|\,A=\pi(X)})}}{\varepsilon^2|\mathcal{D}^{(k)}|t^2}$$

$$\leq\frac{C_\ell\Big(\big\|\widehat{\boldsymbol{\theta}}_\pi^{(k)} - \boldsymbol{\theta}_\pi^*\big\|^2_{L_2(P_{X\,|\,A=\pi(X)})}\Big)}{\varepsilon^2|\mathcal{D}^{(k)}|t^2},$$

where the last step is due to part (3) of Lemma B.1. Combining the above results, we have that

$$\text{term (I)} = O_P\big(n^{-1/2}\cdot\|\widehat{\boldsymbol{\theta}}_\pi^{(k)} - \boldsymbol{\theta}_\pi^*\|_{L_2(P_X)} + \|\widehat{\boldsymbol{\theta}}_\pi^{(k)} - \boldsymbol{\theta}_\pi^*\|^2_{L_2(P_X)}\big) = o_P(n^{-1/2}),$$

where the last step is due to Assumption 3.4.

**Bounding term (II).** Applying the Cauchy-Schwarz inequality to term (II), we have

$$\left|\frac{1}{|\mathcal{D}^{(k)}|}\sum_{i\in\mathcal{D}^{(k)}}\left(\frac{\mathbb{1}\{A_i = \pi(X_i)\}}{\widehat{\pi}_0^{(k)}(A_i\,|\,X_i)} - \frac{\mathbb{1}\{A_i = \pi(X_i)\}}{\pi_0(A_i\,|\,X_i)}\right)\cdot\big(\widehat{g}_\pi^{(k)}(X_i) - \bar{g}_\pi^{(k)}(X_i)\big)\right|$$

$$\leq\sqrt{\frac{1}{|\mathcal{D}^{(k)}|}\sum_{i\in\mathcal{D}^{(k)}}\mathbb{1}\{A_i = \pi(X_i)\}\cdot\left(\frac{1}{\widehat{\pi}_0^{(k)}(A_i\,|\,X_i)} - \frac{1}{\pi_0(A_i\,|\,X_i)}\right)^2}$$

$$\times\sqrt{\frac{1}{|\mathcal{D}^{(k)}|}\sum_{i\in\mathcal{D}^{(k)}}\mathbb{1}\{A_i = \pi(X_i)\}\cdot\big(\widehat{g}_\pi^{(k)}(X_i) - \bar{g}_\pi^{(k)}(X_i)\big)^2}$$

$$= O_P\left(\epsilon^{-2}\big\|\widehat{\pi}_0^{(k)} - \pi_0\big\|_{L_2(P_{X\,|\,A=\pi(X)})}\cdot\big\|\widehat{g}_\pi^{(k)} - \bar{g}_\pi^{(k)}\big\|_{L_2(P_{X\,|\,A=\pi(X)})}\right) = o_P(n^{-1/2}),$$

where the next-to-last inequality is due to the lower bound on $\pi_0$ and $\widehat{\pi}^{(k)}$; the last equality is due to the given convergence rate of the product estimation error in Assumption 3.4.

**Bounding term (III).** By Assumption 3.4, for any $\beta \in (0, 1)$, there exists $N_1 \in \mathbb{N}_+$ such that for $n \geq N_1$,

$$\mathbb{P}\big(\|\widehat{\boldsymbol{\theta}}_\pi^{(k)} - \boldsymbol{\theta}^*\|_{L_\infty} \leq \min(\underline{\alpha}, \bar{\eta})/2\big) \geq 1 - \beta.$$

On the event $\|\widehat{\boldsymbol{\theta}}_\pi^{(k)} - \boldsymbol{\theta}^*\|_{L_\infty} \leq \min(\underline{\alpha}, \bar{\eta})/2$,

$$\big|\widehat{G}_\pi^{(k)}(x, y)\big| = \big|\ell(x, y; \widehat{\boldsymbol{\theta}}_\pi^{(k)})\big| \leq \bar{\alpha}\exp\left(\frac{\bar{y} + \bar{\eta}}{\underline{\alpha}} - 1\right) + \bar{\eta} + \bar{\alpha}\delta =: L_g.$$

Next, for any $i \in \mathcal{D}^{(k)}$,

$$\mathbb{E}\left[\left(\frac{\mathbb{1}\{A_i = \pi(X_i)\}}{\widehat{\pi}_0^{(k)}(A_i\,|\,X_i)} - \frac{\mathbb{1}\{A_i = \pi(X_i)\}}{\pi_0(A_i\,|\,X_i)}\right)\cdot\big(\widehat{G}_\pi^{(k)}(X_i, Y_i) - \bar{g}_\pi^{(k)}(X_i)\big)\,\bigg|\,\mathcal{D}^{(-k)}\right]$$

$$= \mathbb{E}\left[\mathbb{E}\left[\frac{\mathbb{1}\{A_i = \pi(X_i)\}}{\widehat{\pi}_0^{(k)}(A_i\,|\,X_i)} - \frac{\mathbb{1}\{A_i = \pi(X_i)\}}{\pi_0(A_i\,|\,X_i)}\,\bigg|\,X_i, \mathcal{D}^{(-k)}\right]\right.$$

$$\left.\times\,\mathbb{E}\left[\widehat{G}_\pi^{(k)}(X_i, Y(\pi(X_i))) - \bar{g}_\pi^{(k)}(X_i)\,\big|\,X_i, \mathcal{D}^{(-k)}\right]\,\bigg|\,\mathcal{D}^{(-k)}\right] = 0,$$

where the first step is by the unconfoundedness assumption and the second step is due to the fact that $\bar{g}_\pi^{(k)}$ is the conditional expectation of $\widehat{G}_\pi^{(k)}$.

On the event $\{\|\widehat{\boldsymbol{\theta}}_\pi^{(k)} - \boldsymbol{\theta}_\pi^*\|_{L_\infty} \le \min(\underline{\alpha}, \bar{\eta})\}$. By Chebyshev's inequality, for any $t > 0$,

$$\mathbb{P}\left(\left|\frac{1}{|\mathcal{D}^{(k)}|} \sum_{i \in \mathcal{D}^{(k)}} \left(\frac{\mathbb{1}\{A_i = \pi(X_i)\}}{\widehat{\pi}_0^{(k)}(A_i \mid X_i)} - \frac{\mathbb{1}\{A_i = \pi(X_i)\}}{\pi_0(A_i \mid X_i)}\right) \cdot \left(\widehat{G}_\pi^{(k)}(X_i, Y_i) - \bar{g}_\pi^{(k)}(X_i)\right)\right| \ge t \,\middle|\, \mathcal{D}^{(-k)}\right)$$

$$\le \frac{1}{|\mathcal{D}^{(k)}|t^2} \text{Var}\left(\left[\frac{\mathbb{1}\{A_i = \pi(X_i)\}}{\widehat{\pi}_0^{(k)}(A_i \mid X_i)} - \frac{\mathbb{1}\{A_i = \pi(X_i)\}}{\pi_0(A_i \mid X_i)}\right] \cdot \left(\widehat{G}_\pi^{(k)}(X_i, Y_i) - \bar{g}_\pi^{(k)}(X_i)\right) \,\middle|\, \mathcal{D}^{(-k)}\right)$$

$$\le \frac{1}{|\mathcal{D}^{(k)}|t^2} \mathbb{E}\left[\left[\frac{\mathbb{1}\{A_i = \pi(X_i)\}}{\widehat{\pi}_0^{(k)}(A_i \mid X_i)} - \frac{\mathbb{1}\{A_i = \pi(X_i)\}}{\pi_0(A_i \mid X_i)}\right]^2 \cdot \left(\widehat{G}_\pi^{(k)}(X_i, Y_i) - \bar{g}_\pi^{(k)}(X_i)\right)^2 \,\middle|\, \mathcal{D}^{(-k)}\right]$$

$$\le \frac{4L_g^2}{|\mathcal{D}^{(k)}|\varepsilon^4 t^2} \|\widehat{\pi}_0^{(k)} - \pi_0\|_{L_2(P_{X \mid T = \pi(X)})}^2.$$

The above inequality along with a union bound implies that

$$\text{term (III)} = O_P\left(\|\widehat{\pi}_0^{(k)} - \pi_0\|_{L_2(P_{X \mid A = \pi(X)})} / \sqrt{|\mathcal{D}^{(k)}|}\right) = o_P(n^{-1/2}),$$

where the last step is by the consistency of $\widehat{\pi}_0^{(k)}$ assumed in Assumption 3.4.

**Bounding term (IV).** We first show that term (IV) is of zero-mean:

$$\mathbb{E}\left[-\frac{1}{|\mathcal{D}^{(k)}|} \sum_{i \in \mathcal{D}^{(k)}} \frac{\mathbb{1}\{A_i = \pi(X_i)\}}{\pi_0(A_i \mid X_i)} \cdot \left(\widehat{g}_\pi^{(k)}(X_i) - g_\pi(X_i)\right) + \frac{1}{|\mathcal{D}^{(k)}|} \sum_{i \in \mathcal{D}^{(k)}} \left(\widehat{g}_\pi^{(k)}(X_i) - g_\pi(X_i)\right) \,\middle|\, \mathcal{D}^{(-k)}\right]$$

$$= -\mathbb{E}\left[\frac{\mathbb{1}\{A_i = \pi(X_i)\}}{\pi_0(A_i \mid X_i)} \cdot \left(\widehat{g}_\pi^{(k)}(X_i) - g_\pi(X_i)\right) \,\middle|\, \mathcal{D}^{(-k)}\right] + \mathbb{E}\left[\widehat{g}_\pi^{(k)}(X_i) - g_\pi(X_i) \,\middle|\, \mathcal{D}^{(-k)}\right] = 0.$$

By Chebyshev's inequality, for any $t > 0$,

$$\mathbb{P}\left(\left|\frac{1}{|\mathcal{D}^{(k)}|} \sum_{i \in \mathcal{D}^{(k)}} \frac{\mathbb{1}\{\pi(X_i) = A_i\}}{\pi_0(A_i \mid X_i)} \cdot \left(\widehat{g}_\pi^{(k)}(X_i) - g_\pi(X_i)\right) - \frac{1}{|\mathcal{D}^{(k)}|} \sum_{i \in \mathcal{D}^{(k)}} \left(\widehat{g}_\pi^{(k)}(X_i) - g_\pi(X_i)\right)\right| \ge t \,\middle|\, \mathcal{D}^{(-k)}\right)$$

$$\le \frac{1}{|\mathcal{D}^{(k)}|t^2} \text{Var}\left(\frac{\mathbb{1}\{A_i = \pi(X_i)\}}{\pi_0(A_i \mid X_i)} \cdot \left(\widehat{g}_\pi^{(k)}(X_i) - g_\pi(X_i)\right) - \left(\widehat{g}_\pi^{(k)}(X_i) - g_\pi(X_i)\right) \,\middle|\, \mathcal{D}^{(-k)}\right)$$

$$= \frac{1}{|\mathcal{D}^{(k)}|t^2} \mathbb{E}\left[\frac{1 - \pi_0(\pi(X_i) \mid X_i)}{\pi_0(\pi(X_i) \mid X_i)} \cdot \left(\widehat{g}_\pi^{(k)}(X_i) - g_\pi(X_i)\right)^2 \,\middle|\, \mathcal{D}^{(-k)}\right].$$

As a result, term (IV) $= O_P\left(\|\widehat{g}_\pi^{(k)} - g_\pi\|_{L_2(P_X)} / \sqrt{n}\right)$. Note that

$$\|\widehat{g}_\pi^{(k)} - g_\pi\|_{L_2(P_X)} = O(\|\widehat{g}_\pi^{(k)} - g_\pi\|_{L_2(P_{X \mid A = \pi(X)})})$$

$$\le O\left(\|\widehat{g}_\pi^{(k)} - \bar{g}_\pi\|_{L_2(P_{X \mid A = \pi(X)})} + \|\bar{g}_\pi^{(k)} - g_\pi\|_{L_2(P_{X \mid A = \pi(X)})}\right),$$

where the first inequality follows from the overlap condition. By Assumption 3.4, $\|\widehat{g}_\pi^{(k)} - \bar{g}_\pi\|_{L_\infty} = o_P(1)$. Meanwhile,

$$\|\bar{g}_\pi^{(k)} - g_\pi\|_{L_2(P_{X \mid A = \pi(X)})}^2$$

$$= \mathbb{E}\left[(\bar{g}(X) - g(X))^2 \mid A = \pi(X), \mathcal{D}^{-k}\right]$$

$$= \mathbb{E}\left[\left(\mathbb{E}\left[\ell(X, Y(\pi(X)); \widehat{\boldsymbol{\theta}}_\pi^{(k)}(X)) - \ell(X, Y(\pi(X)); \boldsymbol{\theta}_\pi^*(X)) \mid X\right]\right)^2 \,\middle|\, A = \pi(X), \mathcal{D}^{(-k)}\right]$$

$$\overset{(i)}{\le} \mathbb{E}\left[\left(\ell(X, Y(\pi(X)); \widehat{\boldsymbol{\theta}}_\pi^{(k)}) - \ell(X, Y(\pi(X)); \boldsymbol{\theta}_\pi^*)\right)^2 \,\middle|\, A = \pi(X), \mathcal{D}^{(-k)}\right]$$

$$\overset{(ii)}{=} O\left(\|\widehat{\boldsymbol{\theta}}_\pi^{(k)} - \boldsymbol{\theta}_\pi^*\|_{L_2(P_{X \mid A = \pi(X)})}^2\right) = o_P(1).$$

Above, step (i) follows from Jensen's inequality and step (ii) from part (3) of Lemma B.1. Combining everything, we have that term (IV) is of rate $o_P(n^{-1/2})$.

**Putting everything together.** So far we have shown that for each fold $k \in [K]$, there is

$$\widehat{\mathcal{V}}_\delta^{(k)}(\pi) - \mathcal{V}_\delta^{*(k)}(\pi) = o_P(n^{-1/2}).$$

Averaging over all $k$ folds, we have

$$\sqrt{n} \cdot \left( \widehat{\mathcal{V}}_\delta(\pi) - \mathcal{V}_\delta(\pi) \right)$$

$$= \frac{1}{\sqrt{n}} \sum_{i=1}^{n} \left\{ -\frac{\mathbb{1}\{A_i = \pi(X_i)\}}{\pi_0(A_i \mid X_i)} \cdot \left( G_\pi(X_i, Y_i) - g_\pi(X_i) \right) - g_\pi(X_i) - \mathcal{V}_\delta(\pi) \right\} + o_P(1),$$

By the central limit theorem and Slutsky's theorem.

$$\sqrt{n} \cdot \left( \widehat{\mathcal{V}}_\delta(\pi) - \mathcal{V}_\delta(\pi) \right) \xrightarrow{\mathrm{d}} \mathcal{N}(0, \sigma^2),$$

where

$$\sigma^2 = \mathrm{Var}\left( \frac{\mathbb{1}\{A = \pi(X)\}}{\pi_0(A \mid X)} \cdot \left( G_\pi(X, Y) - g_\pi(X) \right) + g_\pi(X) \right).$$

## B.3 PROOF OF THEOREM 4.2

By Assumption 3.3, taking $\pi(x) \equiv a$ for any $a \in [M]$, there exist constants $\bar{\alpha}_a, \underline{\alpha}_a, \bar{\eta}_a$ such that

$$0 < \underline{\alpha}_a \leq \boldsymbol{\alpha}_a^*(x) \leq \bar{\alpha}_a, \quad |\boldsymbol{\eta}_a(x)| \leq \bar{\eta}_a, \quad \forall x \in \mathcal{X}.$$

Letting $\underline{\alpha} = \min_{a \in [M]} \underline{\alpha}_a$, $\bar{\alpha} = \max_{a \in [M]} \bar{\alpha}_a$, $\bar{\eta} = \max_{a \in [M]} \bar{\eta}_a$, it follows that

$$0 < \underline{\alpha} \leq \boldsymbol{\alpha}_a^*(x) \leq \bar{\alpha}, \quad |\boldsymbol{\eta}_a(x)| \leq \bar{\eta}, \quad \forall x \in \mathcal{X}, \forall a \in [M]. \tag{13}$$

For any $a \in [M]$, if we take $\pi(x) \equiv a$, then by (1) of Lemma B.1,

$$\mathbb{E}\left[ \nabla_\theta \ell(x, Y(a); \boldsymbol{\theta}_a^*(x)) \mid X = x \right] = 0.$$

By (2) of Lemma B.1, for any $a \in [M]$, there exists a constant $\xi_a > 0$ such that for any $\|\theta - \boldsymbol{\theta}_a^*(x)\|_2 \leq \xi_a$

$$|\ell(x, y; \theta) - \ell(x, y; \boldsymbol{\theta}_a^*(x)) - \nabla_\theta \ell(x, y; \boldsymbol{\theta}_a^*(x))^\top (\theta - \boldsymbol{\theta}_a^*(x))| \leq \bar{\ell}_a(x, y)\|\theta - \boldsymbol{\theta}_a^*(x)\|_2^2,$$

for some function $\bar{\ell}_a(x, y) \leq L_a$ for some constant $L_a$. Similarly, we shall take $\xi = \min_{a \in [M]} \xi_a$, $\bar{\ell}(x, y) = \max_a \bar{\ell}_a(x, y)$, and $L = \sum_{a \in [M]} L_a$.

By (3) of Lemma B.1, for any $a \in [M]$, there exists a constant $\xi_{1,a} > 0$ such that for any $\|\boldsymbol{\theta} - \boldsymbol{\theta}_a^*\|_{L_\infty} \leq \xi_{1,a}$,

$$\left\| \ell(X, Y(a); \boldsymbol{\theta}(X)) - \ell(X, Y(a); \boldsymbol{\theta}_a^*(X)) \right\|_{L_2(P_{X,Y(a) \mid A=a})} \leq C_{\ell,a}\|\boldsymbol{\theta} - \boldsymbol{\theta}_a^*\|_{L_2(P_{X \mid A=a})}.$$

Taking $\xi_1 = \min_{a \in [M]} \xi_{1,a}$ and $C_\ell = \sum_{a \in [M]} C_{\ell,a}$, the above inequality holds for any $a \in [M]$ and any $\|\boldsymbol{\theta} - \boldsymbol{\theta}_a^*\|_{L_\infty} \leq \xi_1$.

### B.3.1 REGRET DECOMPOSITION

The regret bound of Algorithm 2 builds on the following regret decomposition:

$$\begin{aligned}
\mathcal{R}_\delta(\widehat{\pi}_{\mathrm{LN}}) &= \mathcal{V}_\delta(\pi^*) - \mathcal{V}_\delta(\widehat{\pi}_{\mathrm{LN}}) \\
&= \mathcal{V}_\delta(\pi^*) - \widehat{\mathcal{V}}_\delta^{\mathrm{LN}}(\pi^*) + \widehat{\mathcal{V}}_\delta^{\mathrm{LN}}(\pi^*) - \widehat{\mathcal{V}}_\delta^{\mathrm{LN}}(\widehat{\pi}_{\mathrm{LN}}) + \widehat{\mathcal{V}}_\delta^{\mathrm{LN}}(\widehat{\pi}_{\mathrm{LN}}) - \mathcal{V}_\delta(\widehat{\pi}_{\mathrm{LN}}) \\
&\leq \mathcal{V}_\delta(\pi^*) - \widehat{\mathcal{V}}_\delta^{\mathrm{LN}}(\pi^*) + \widehat{\mathcal{V}}_\delta^{\mathrm{LN}}(\widehat{\pi}_{\mathrm{LN}}) - \mathcal{V}_\delta(\widehat{\pi}_{\mathrm{LN}}) \\
&\leq 2 \sup_{\pi \in \Pi} \left| \widehat{\mathcal{V}}_\delta^{\mathrm{LN}}(\pi) - \mathcal{V}_\delta(\pi) \right|, \tag{14}
\end{aligned}$$

where the second-to-last step is by the choice of $\widehat{\pi}_{\mathrm{LN}}$. For any $\pi \in \Pi$ and any fold $k \in [K]$, we define an intermediate quantity

$$\tilde{\mathcal{V}}_\delta^{(k)} := \frac{1}{|\mathcal{D}^{(k)}|} \sum_{i \in \mathcal{D}^{(k)}} \frac{\mathbb{1}\{A_i = \pi(X_i)\}}{\pi_0(A_i \mid X_i)} \cdot \left( G_{\pi(X_i)}(X_i, Y_i) - g_{\pi(X_i)}(X_i) \right) + g_{\pi(X_i)}(X_i).$$

Letting $\tilde{\mathcal{V}}_\delta = -\frac{1}{K} \sum_{k=1}^K \tilde{\mathcal{V}}_\delta^{(k)}$, we have

$$\left| \widehat{\mathcal{V}}_\delta^{\text{LN}}(\pi) - \mathcal{V}_\delta(\pi) \right| = \left| -\frac{1}{K} \sum_{k=1}^K \widehat{\mathcal{V}}_\delta^{\text{LN},(k)}(\pi) - \mathcal{V}_\delta(\pi) \right|$$

$$\leq \left| \frac{1}{K} \sum_{k=1}^K \widehat{\mathcal{V}}_\delta^{\text{LN},(k)}(\pi) - \tilde{\mathcal{V}}_\delta(\pi) \right| + \left| \tilde{\mathcal{V}}_\delta(\pi) - \mathcal{V}_\delta(\pi) \right|$$

$$\leq \sup_{\pi \in \Pi} \frac{1}{K} \sum_{k=1}^K \left| \widehat{\mathcal{V}}_\delta^{\text{LN},(k)}(\pi) - \tilde{\mathcal{V}}_\delta^{(k)}(\pi) \right| + \sup_{\pi \in \Pi} \left| -\tilde{\mathcal{V}}_\delta(\pi) - \mathcal{V}_\delta(\pi) \right|.$$

Taking the supremum over all $\pi \in \Pi$, we have that

$$\sup_{\pi \in \Pi} \left| \widehat{\mathcal{V}}_\delta^{\text{LN}}(\pi) - \mathcal{V}_\delta(\pi) \right| \leq \sup_{\pi \in \Pi} \left| -\tilde{\mathcal{V}}_\delta(\pi) - \mathcal{V}_\delta(\pi) \right| + \sup_{\pi \in \Pi} \frac{1}{K} \sum_{k=1}^K \left| \widehat{\mathcal{V}}_\delta^{\text{LN},(k)}(\pi) - \tilde{\mathcal{V}}_\delta^{(k)}(\pi) \right|.$$

We shall show that the first term above is $O_P(n^{-1/2})$ and the second term is $o_P(n^{-1/2})$. In the following, we refer to the two terms as the effective term and the negligible term, respectively. The following lemma is essential for establishing the uniform convergence results.

**Lemma B.2.** *Suppose $h$ is a function of $(x, a, y, \pi(x))$. Given a set of data $\{z_i = (x_i, a_i, y_i)\}_{i=1}^n$, suppose that $|h(z_i, \pi(x_i))| \leq c_i(z_i)$. Then the Rademacher complexity*

$$\mathbb{E}_\epsilon \left[ \sup_{\pi \in \Pi} \left| \frac{1}{n} \sum_{i=1}^n \epsilon_i h(x_i, a_i, y_i, \pi(x_i)) \right| \right] \leq \frac{\sqrt{\sum_{i=1}^n c_i(z_i)^2}}{n} \cdot (32 + 4\kappa(\Pi)),$$

*where $\epsilon_i \overset{\text{i.i.d.}}{\sim} \text{Unif}\{\pm 1\}$ are i.i.d. Rademacher random variables and $\mathbb{E}_\epsilon$ means the expectation over $\epsilon$.*

### B.3.2 THE EFFECTIVE TERM

Denote $Z_i = (X_i, A_i, Y_i)$ and take

$$h(Z_i, \pi(X_i)) = -\frac{\mathbb{1}\{A_i = \pi(X_i)\}}{\pi_0(A_i \mid X_i)} \cdot \left( G_{\pi(X_i)}(X_i, Y_i) - g_{\pi(X_i)}(X_i) \right) - g_{\pi(X_i)}(X_i) - \mathcal{V}_\delta(\pi).$$

Under the unconfoundedness assumption in Assumption 2.1, $\mathbb{E}[h(Z_i, \pi(X_i))] = 0$. By Equation (13), we have

$$|h(Z_i, \pi(X_i))| \leq \frac{6}{\varepsilon} \cdot \left( \bar{\alpha} \cdot \exp\left( \frac{\bar{\eta}}{\underline{\alpha}} - 1 \right) + \bar{\eta} + \bar{\alpha}\delta \right) =: C_0(\bar{\alpha}, \underline{\alpha}, \bar{\eta}, \delta, \varepsilon).$$

Meanwhile, we have write

$$\sup_{\pi \in \Pi} \left| \frac{1}{K} \sum_{k=1}^K -\tilde{\mathcal{V}}_\delta^{(k)}(\pi) - \mathcal{V}_\delta(\pi) \right| = \sup_{\pi \in \Pi} \left| \frac{1}{K} \sum_{k=1}^K \frac{1}{|\mathcal{D}^{(k)}|} \sum_{i \in \mathcal{D}^{(k)}} h(Z_i; \pi(X_i)) \right| = \sup_{\pi \in \Pi} \left| \frac{1}{n} \sum_{i=1}^n h(Z_i; \pi(X_i)) \right|.$$

Next, we define

$$f(z_1, \ldots, z_n; \pi) = \frac{1}{n} \sum_{i=1}^n h(z_i, \pi(x_i)).$$

Consider two arbitrary data sets $\{z_i\}_{i=1}^n$ and $\{z_i'\}_{i=1}^n$. We can check that for any $\pi \in \Pi$ and any $j \in [n]$,

$$\left| f(z_1, \ldots, z_j, \ldots, z_n; \pi) \right| - \sup_{\pi' \in \Pi} \left| f(z_1, \ldots, z_j', \ldots, z_n; \pi') \right|$$

$$\leq \left| f(z_1, \ldots, z_j, \ldots, z_n; \pi) \right| - \left| f(z_1, \ldots, z_j', \ldots, z_n; \pi) \right|$$

$$\leq \sup_{\pi \in \Pi} \left| f(z_1, \ldots, z_j, \ldots, z_n; \pi) - f(z_1, \ldots, z_j', \ldots, z_n; \pi) \right|$$

$$= \sup_{\pi \in \Pi} \frac{1}{n} \left| h(z_j; \pi) - h(z_j'; \pi) \right| \leq C_0(\bar{\alpha}, \underline{\alpha}, \bar{\eta}, \delta, \varepsilon)/n. \tag{15}$$

Above, the first inequality is because of the definition of sup and the second is due to the triangle inequality; the last step is due to the boundedness of $h$. Taking the supremum over all $\pi \in \Pi$ in (15), we have that

$$\sup_{\pi \in \Pi} \left| f(z_1, \ldots, z_j, \ldots, z_n; \pi) \right| - \sup_{\pi \in \Pi} \left| f(z_1, \ldots, z_j', \ldots, z_n; \pi) \right| \leq C_0(\bar{\alpha}, \underline{\alpha}, \bar{\eta}, \delta, \varepsilon)/n.$$

By the bounded difference inequality (Wainwright, 2019, Corollary 2.21), for any $t > 0$,

$$\mathbb{P}\left( \sup_{\pi \in \Pi} \left| \frac{1}{n} h(Z_i, \pi(X_i)) \right| - \mathbb{E}\left[ \sup_{\pi \in \Pi} \left| \frac{1}{n} h(Z_i, \pi(X_i)) \right| \right] \geq t \right)$$

$$= \mathbb{P}\left( \sup_{\pi \in \Pi} \left| f(\{Z_i\}_{i \in [n]}; \pi) \right| - \mathbb{E}\left[ \sup_{\pi \in \Pi} \left| f(\{Z_i\}_{i \in [n]}; \pi) \right| \right] \geq t \right) \leq e^{-\frac{2nt^2}{C_0(\bar{\alpha}, \underline{\alpha}, \bar{\eta}, \delta, \varepsilon)^2}}.$$

Take $t = C(\bar{\alpha}, \underline{\alpha}, \bar{\eta}, \underline{\eta}) \sqrt{\frac{1}{2n} \log\left(\frac{1}{\beta}\right)}$. Then with probability at least $1 - \beta$,

$$\sup_{\pi \in \Pi} \left| \frac{1}{n} h(Z_i, \pi(X_i)) \right| < \mathbb{E}\left[ \sup_{\pi \in \Pi} \left| \frac{1}{n} h(Z_i, \pi(X_i)) \right| \right] + C(\bar{\alpha}, \underline{\alpha}, \bar{\eta}, \underline{\eta}) \sqrt{\frac{1}{2n} \log\left(\frac{1}{\beta}\right)}.$$

It remains to bound the expectation term. Let $Z_1', \ldots, Z_n'$ be an i.i.d. copy of $Z_1, \ldots, Z_n$, and let $\epsilon_i \overset{\text{i.i.d.}}{\sim} \text{Unif}(\{\pm 1\})$. Then

$$\mathbb{E}\left[ \sup_{\pi \in \Pi} \left| \frac{1}{n} \sum_{i \in [n]} h(Z_i, \pi(X_i)) - \mathbb{E}\left[ h(Z_i, \pi(X_i)) \right] \right| \right]$$

$$= \mathbb{E}\left[ \sup_{\pi \in \Pi} \left| \frac{1}{n} \sum_{i \in [n]} h(Z_i, \pi(X_i)) - \mathbb{E}_{Z'}\left[ \frac{1}{n} \sum_{i \in [n]} h(Z_i', \pi(X_i')) \right] \right| \right]$$

$$\overset{(i)}{\leq} \mathbb{E}\left[ \sup_{\pi \in \Pi} \left| \frac{1}{n} \sum_{i \in [n]} h(Z_i, \pi(X_i)) - \frac{1}{n} \sum_{i \in [n]} h(Z_i', \pi(X_i')) \right| \right]$$

$$\overset{(ii)}{=} \mathbb{E}\left[ \sup_{\pi \in \Pi} \left| \frac{1}{n} \sum_{i \in [n]} \epsilon_i \big( h(Z_i, \pi(X_i)) - h(Z_i', \pi(X_i')) \big) \right| \right],$$

$$\leq 2\mathbb{E}\left[ \sup_{\pi \in \Pi} \left| \frac{1}{n} \sum_{i \in [n]} \epsilon_i h(Z_i, \pi(X_i)) \right| \right]$$

$$= 2\mathbb{E}\left[ \mathbb{E}_\epsilon\left[ \sup_{\pi \in \Pi} \left| \frac{1}{n} \sum_{i \in [n]} \epsilon_i h(Z_i, \pi(X_i)) \right| \right] \right], \tag{16}$$

step (i) is by Jensen's inequality and step (ii) is because of the symmetry of $(Z_i, Z_i')$.

Applying Lemma B.2,

$$\mathbb{E}_\epsilon\left[ \sup_{\pi \in \Pi} \left| \frac{1}{n} \sum_{i \in [n]} \epsilon_i h(Z_i, \pi(X_i)) \right| \right] \leq \frac{2C(\bar{\alpha}, \underline{\alpha}, \bar{\eta}, \underline{\eta})}{\sqrt{n}} (32 + 4\kappa(\Pi)).$$

Combining the above, for any $\beta \in (0, 1)$, we have with probability at least $1 - \beta$,

$$\sup_{\pi \in \Pi} \left| \tilde{\mathcal{V}}_\delta(\pi) - \mathcal{V}_\delta(\pi) \right| \leq \frac{C_0(\bar{\alpha}, \underline{\alpha}, \bar{\eta}, \delta, \varepsilon)}{\sqrt{n}} \big( 64 + 8\kappa(\Pi) + \sqrt{\log(1/\beta)} \big). \tag{17}$$

### B.3.3 BOUNDING THE NEGLIGIBLE TERM

We now proceed to the negligible term. For any $\pi \in \Pi$ and any $k \in [K]$, consider the following decomposition:

$$
\begin{aligned}
&\widehat{\mathcal{V}}_\delta^{\mathrm{LN},(k)}(\pi) - \tilde{\mathcal{V}}_\delta^{(k)}(\pi) \\
&= \frac{1}{|\mathcal{D}^{(k)}|} \sum_{i \in \mathcal{D}^{(k)}} \frac{\mathbb{1}\{A_i = \pi(X_i)\}}{\widehat{\pi}_0(A_i \mid X_i)} \big(\widehat{G}_{\pi(X_i)}^{(k)}(X_i, Y_i) - \widehat{g}_{\pi(X_i)}^{(k)}(X_i)\big) + \widehat{g}_{\pi(X_i)}^{(k)}(X_i) \\
&\quad - \frac{1}{|\mathcal{D}^{(k)}|} \sum_{i \in \mathcal{D}^{(k)}} \frac{\mathbb{1}\{A_i = \pi(X_i)\}}{\pi_0(A_i \mid X_i)} \big(G_{\pi(X_i)}(X_i, Y_i) - g_{\pi(X_i)}(X_i)\big) - g_{\pi(X_i)}(X_i) \\
&= \frac{1}{|\mathcal{D}^{(k)}|} \sum_{i \in \mathcal{D}^{(k)}} \left(\frac{\mathbb{1}\{A_i = \pi(X_i)\}}{\widehat{\pi}_0(A_i \mid X_i)} - \frac{\mathbb{1}\{A_i = \pi(X_i)\}}{\pi_0(A_i \mid X_i)}\right) \big(\widehat{G}_{\pi(X_i)}^{(k)}(X_i, Y_i) - \bar{g}_{\pi(X_i)}^{(k)}(X_i)\big) \\
&\quad + \frac{1}{|\mathcal{D}^{(k)}|} \sum_{i \in \mathcal{D}^{(k)}} \left(\frac{\mathbb{1}\{A_i = \pi(X_i)\}}{\widehat{\pi}_0(A_i \mid X_i)} - \frac{\mathbb{1}\{A_i = \pi(X_i)\}}{\pi_0(A_i \mid X_i)}\right) \big(\bar{g}_{\pi(X_i)}^{(k)}(X_i) - \widehat{g}_{\pi(X_i)}^{(k)}(X_i)\big) \\
&\quad + \frac{1}{|\mathcal{D}^{(k)}|} \sum_{i \in \mathcal{D}^{(k)}} \frac{\mathbb{1}\{A_i = \pi(X_i)\}}{\pi_0(A_i \mid X_i)} \big(\widehat{G}_{\pi(X_i)}^{(k)}(X_i, Y_i) - G_{\pi(X_i)}(X_i, Y_i)\big) \\
&\quad - \frac{1}{|\mathcal{D}^{(k)}|} \sum_{i \in \mathcal{D}^{(k)}} \frac{\mathbb{1}\{A_i = \pi(X_i)\}}{\pi_0(A_i \mid X_i)} \big(\widehat{g}_{\pi(X_i)}^{(k)}(X_i) - g_{\pi(X_i)}(X_i)\big) + \frac{1}{|\mathcal{D}^{(k)}|} \sum_{i \in \mathcal{D}^{(k)}} \big(\widehat{g}_{\pi(X_i)}^{(k)}(X_i) - g_{\pi(X_i)}(X_i)\big).
\end{aligned}
$$

For notational simplicity, we denote

$$
K_1(\pi) := \frac{1}{|\mathcal{D}^{(k)}|} \sum_{i \in \mathcal{D}^{(k)}} \left(\frac{\mathbb{1}\{A_i = \pi(X_i)\}}{\widehat{\pi}_0(A_i \mid X_i)} - \frac{\mathbb{1}\{A_i = \pi(X_i)\}}{\pi_0(A_i \mid X_i)}\right) \big(\widehat{G}_{\pi(X_i)}^{(k)}(X_i, Y_i) - \bar{g}_{\pi(X_i)}^{(k)}(X_i)\big),
$$

$$
K_2(\pi) := \frac{1}{|\mathcal{D}^{(k)}|} \sum_{i \in \mathcal{D}^{(k)}} \left(\frac{\mathbb{1}\{A_i = \pi(X_i)\}}{\widehat{\pi}_0(A_i \mid X_i)} - \frac{\mathbb{1}\{A_i = \pi(X_i)\}}{\pi_0(A_i \mid X_i)}\right) \big(\bar{g}_{\pi(X_i)}^{(k)}(X_i) - \widehat{g}_{\pi(X_i)}^{(k)}(X_i)\big),
$$

$$
K_3(\pi) := \frac{1}{|\mathcal{D}^{(k)}|} \sum_{i \in \mathcal{D}^{(k)}} \frac{\mathbb{1}\{A_i = \pi(X_i)\}}{\pi_0(A_i \mid X_i)} \big(\widehat{G}_{\pi(X_i)}^{(k)}(X_i, Y_i) - G_{\pi(X_i)}(X_i, Y_i)\big),
$$

$$
K_4(\pi) := -\frac{1}{|\mathcal{D}^{(k)}|} \sum_{i \in \mathcal{D}^{(k)}} \frac{\mathbb{1}\{A_i = \pi(X_i)\}}{\pi_0(A_i \mid X_i)} \big(\widehat{g}_{\pi(X_i)}^{(k)}(X_i) - g_{\pi(X_i)}(X_i)\big) + \frac{1}{|\mathcal{D}^{(k)}|} \sum_{i \in \mathcal{D}^{(k)}} \big(\widehat{g}_{\pi(X_i)}^{(k)}(X_i) - g_{\pi(X_i)}(X_i)\big).
$$

We proceed to bound each term separately. To ease the presentation, we shall write $\mathbb{E}_k$ and $\mathbb{P}_k$ as the expectation and probability conditioned on $\mathcal{D}^{(-k)}$, respectively.

**Bounding $K_1(\pi)$.** Here, we take

$$
h_1(Z_i; \pi(X_i)) := \left(\frac{\mathbb{1}\{A_i = \pi(X_i)\}}{\widehat{\pi}_0(A_i \mid X_i)} - \frac{\mathbb{1}\{A_i = \pi(X_i)\}}{\pi_0(A_i \mid X_i)}\right) \big(\widehat{G}_{\pi(X_i)}^{(k)}(X_i, Y_i) - \bar{g}_{\pi(X_i)}^{(k)}(X_i)\big).
$$

Since $\bar{g}_a^{(k)}(X)$ is the conditional expectation of $\widehat{G}_a^{(k)}(X, Y(a))$, we have

$$
\begin{aligned}
\mathbb{E}_k\big[h_1(Z_i, \pi(X_i)) \mid X_i\big] &= \mathbb{E}_k\left[\left(\frac{\mathbb{1}\{A_i = \pi(X_i)\}}{\widehat{\pi}_0(A_i \mid X_i)} - \frac{\mathbb{1}\{A_i = \pi(X_i)\}}{\pi_0(A_i \mid X_i)}\right) \big(\widehat{G}_{A_i}^{(k)}(X_i, Y_i) - \bar{g}_{A_i}^{(k)}(X_i)\big) \,\Big|\, X_i\right] \\
&= \left(\frac{\pi_0(\pi(X_i))}{\widehat{\pi}_0(\pi(X_i) \mid X_i)} - 1\right) \mathbb{E}_k\big[\widehat{G}_{\pi(X_i)}^{(k)}(X_i, Y_i) - \bar{g}_{\pi(X_i)}^{(k)}(X_i) \mid X_i\big] \\
&= 0.
\end{aligned}
$$

By Assumption 3.4, there exists $N_1 \in \mathbb{N}_+$, such that when $n \geq N_1$, w. p. at least $1 - \beta$,

$$
\max_{a \in [M]} \|\widehat{\boldsymbol{\theta}}_a^{(k)} - \boldsymbol{\theta}_a^*\|_{L_\infty} \leq \max(\bar{\alpha}, \underline{\alpha}, \bar{\eta})/2.
$$

On the event $\{\max_{a\in[M]} \|\widehat{\boldsymbol{\theta}}_a^{(k)} - \boldsymbol{\theta}_a^*\|_{L_\infty} \leq \max(\bar{\alpha}, \underline{\alpha}, \bar{\eta})/2\}$, we have for any $a \in [M]$

$$|\ell(x, y, ; \widehat{\boldsymbol{\theta}}_a(x))| \leq 2\bar{\alpha} \exp\left(\frac{2\bar{y} + 4\bar{\eta}}{\underline{\alpha}} - 1\right) + 2\bar{\eta} + 2\bar{\alpha}\delta.$$

Letting $C_1(\bar{\alpha}, \underline{\alpha}, \bar{\eta}, \delta, \varepsilon) = 4\bar{\alpha} \exp\left(\frac{2\bar{y}+4\bar{\eta}}{\underline{\alpha}} - 1\right) + 4\bar{\eta} + 4\bar{\alpha}\delta)/\varepsilon^2$, We can then check that

$$\begin{aligned}
|h_1(Z_i; \pi(X_i))| &\leq 2C_1(\bar{\alpha}, \underline{\alpha}, \bar{\eta}, \delta, \varepsilon) \cdot \left|\widehat{\pi}_0(\pi(X_i) \,|\, X_i) - \pi_0(\pi(X_i) \,|\, X_i)\right| \\
&\leq 2C_1(\bar{\alpha}, \underline{\alpha}, \bar{\eta}, \delta, \varepsilon) \cdot \max_{a\in[M]} \left|\widehat{\pi}_0(a \,|\, X_i) - \pi_0(a \,|\, X_i)\right| =: c_1(X_i).
\end{aligned}$$

The upper bound is a constant conditional on $X_i$'s and $\mathcal{D}^{(-k)}$. We now apply the bounded difference inequality conditional on $X = \{X_i\}_{i\in[n]}$:

$$\mathbb{P}_k\left(\sup_{\pi\in\Pi}\left|\frac{1}{|\mathcal{D}^{(-k)}|}\sum_{i\in\mathcal{D}^{(k)}} h_1(Z_i, \pi(X_i))\right| - \mathbb{E}_k\left[\sup_{\pi\in\Pi}\left|\frac{1}{|\mathcal{D}^{(k)}|}\sum_{i\in\mathcal{D}^{(k)}} h_1(Z_i, \pi(X_i))\right|\,\Big|\,X\right] \geq t \,\Big|\, X\right)$$

$$\leq \exp\left(-\frac{2|\mathcal{D}^{(k)}|^2 t^2}{\sum_{i\in\mathcal{D}^{(k)}} c_1(X_i)^2}\right).$$

Taking $t = \sqrt{\sum_{i\in\mathcal{D}^{(k)}} c_1(X_i)^2 \log(1/\beta)}/|\mathcal{D}^{(k)}|$, we have with probability at least $1 - \beta$,

$$\begin{aligned}
\sup_{\pi\in\Pi}\left|\frac{1}{|\mathcal{D}^{(-k)}|}\sum_{i\in\mathcal{D}^{(k)}} h_1(Z_i, \pi(X_i))\right| &\leq \mathbb{E}_k\left[\sup_{\pi\in\Pi}\left|\frac{1}{|\mathcal{D}^{(k)}|}\sum_{i\in\mathcal{D}^{(k)}} h_1(Z_i, \pi(X_i))\right|\,\Big|\,X\right] \\
&\quad + \frac{\sqrt{\sum_{i\in\mathcal{D}^{(k)}} c_1(X_i)^2}}{|\mathcal{D}^{(k)}|}\sqrt{\log(1/\beta)}.
\end{aligned}$$

For each $i \in \mathcal{D}^{(k)}$, we take $A_i'$ and $Y_i'$ as i.i.d. copies of $A_i$ and $Y_i$ conditional on $X_i$, respectively. By a similar symmetrization argument as in the proof for the effective term, we have

$$\begin{aligned}
&\mathbb{E}_k\left[\sup_{\pi\in\Pi}\left|\frac{1}{|\mathcal{D}^{(k)}|}\sum_{i\in\mathcal{D}^{(k)}} h_1(Z_i, \pi(X_i))\right|\,\Big|\,X\right] \\
&= \mathbb{E}_k\left[\sup_{\pi\in\Pi}\left|\frac{1}{|\mathcal{D}^{(k)}|}\sum_{i\in\mathcal{D}^{(k)}} h_1(X_i, A_i, Y_i, \pi(X_i)) - \mathbb{E}_{A',Z'}\left[\frac{1}{|\mathcal{D}^{(k)}|}\sum_{i\in\mathcal{D}^{(k)}} h(X_i, A_i', Y_i', \pi(X_i))\right]\right|\,\Big|\,X\right] \\
&\leq \mathbb{E}_k\left[\sup_{\pi\in\Pi}\left|\frac{1}{|\mathcal{D}^{(k)}|}\sum_{i\in\mathcal{D}^{(k)}} h_1(X_i, A_i, Y_i, \pi(X_i)) - \frac{1}{|\mathcal{D}^{(-k)}|}\sum_{i\in\mathcal{D}^{(k)}} h(X_i, A_i', Y_i', \pi(X_i))\right|\,\Big|\,X\right] \\
&\leq \mathbb{E}_k\left[\sup_{\pi\in\Pi}\left|\frac{1}{|\mathcal{D}^{(k)}|}\sum_{i\in\mathcal{D}^{(k)}} \epsilon_i\left(h_1(X_i, A_i, Y_i, \pi(X_i)) - h(X_i, A_i', Y_i', \pi(X_i))\right)\right|\,\Big|\,X\right] \\
&\leq 2\mathbb{E}_k\left[\sup_{\pi\in\Pi}\left|\frac{1}{|\mathcal{D}^{(k)}|}\sum_{i\in\mathcal{D}^{(k)}} \epsilon_i h_1(X_i, A_i, Y_i, \pi(X_i))\right|\,\Big|\,X\right].
\end{aligned}$$

Applying Lemma B.2 with $c_i = c_1(X_i)$, we have that

$$\mathbb{E}_\epsilon\left[\sup_{\pi\in\Pi}\left|\frac{1}{|\mathcal{D}^{(k)}|}\sum_{i\in\mathcal{D}^{(k)}} \epsilon_i h_1(X_i, A_i, Y_i, \pi(X_i))\right|\,\Big|\,X\right] \leq \frac{2\sqrt{\sum_{i\in\mathcal{D}^{(k)}} c_1(X_i)^2}}{|\mathcal{D}^{(k)}|}(32 + 4\kappa(\Pi)).$$

Combining the above, on the event $\{\max_{a\in[M]} \|\widehat{\boldsymbol{\theta}}_a^{(k)} - \boldsymbol{\theta}_a^*\|_{L_\infty} \leq \max(\bar{\alpha}, \underline{\alpha}, \bar{\eta})/2\}$,

$$\mathbb{P}_k\left(\sup_{\pi\in\Pi} |K_1(\pi)| \geq \frac{\sqrt{\sum_{i\in\mathcal{D}^{(k)}} c_1(X_i)^2}}{|\mathcal{D}^{(k)}|}\left(64 + 8\kappa(\Pi) + \sqrt{\log(1/\beta)}\right)\,\Big|\,X\right) \leq \beta.$$

Since $\left|\widehat{\pi}_0(a\,|\,X) - \pi_0(a\,|\,X)\right|^2 \leq 1$,

$$\mathbb{P}_k\left(\frac{1}{|\mathcal{D}^{(k)}|}\sum_{i\in\mathcal{D}^{(k)}}\max_{a\in[M]}\left(\widehat{\pi}_0(a\,|\,X) - \pi_0(a\,|\,X)\right)^2 - \sum_{a\in[M]}\mathbb{E}\left[(\widehat{\pi}_0(a\,|\,X) - \pi_0(a\,|\,X))^2\right] \geq t\right)$$

$$\leq \mathbb{P}_k\left(\frac{1}{|\mathcal{D}^{(k)}|}\sum_{i\in\mathcal{D}^{(k)}}\sum_{a\in[M]}\left(\widehat{\pi}_0(a\,|\,X) - \pi_0(a\,|\,X)\right)^2 - \sum_{a\in[M]}\mathbb{E}\left[(\widehat{\pi}_0(a\,|\,X) - \pi_0(a\,|\,X))^2\right] \geq t\right)$$

$$\leq \sum_{a\in[M]}\mathbb{P}_k\left(\frac{1}{|\mathcal{D}^{(k)}|}\sum_{i\in\mathcal{D}^{(k)}}\left(\widehat{\pi}_0(a\,|\,X) - \pi_0(a\,|\,X)\right)^2 - \mathbb{E}\left[(\widehat{\pi}_0(a\,|\,X) - \pi_0(a\,|\,X))^2\right] \geq t\right)$$

$$\leq M\exp\left(-2|\mathcal{D}^{(k)}|t^2\right).$$

Taking a union bound, with probability at least $1 - 3\beta$, we have that

$$\sup_{\pi\in\Pi}\left|K_1(\pi)\right| \leq \frac{2C_1(\bar{\alpha}, \underline{\alpha}, \bar{\eta}, \delta, \varepsilon)}{\sqrt{|\mathcal{D}^{(k)}|}}\left(20 + 4\kappa(\Pi) + \sqrt{2\log(1/\beta)}\right)$$

$$\times \left(\sum_{a\in[M]}\|\widehat{\pi}_0 - \pi_0\|_{L_2(P_{X\,|\,A=a})} + \left(\frac{1}{2n}\log(M/\beta)\right)^{1/4}\right).$$

Since $\sum_{a\in[M]}\|\widehat{\pi}_0 - \pi_0\|_{L_2(P_{X\,|\,A=a})} = o_P(1)$, there exists $N_1' \geq N_1$ such that when $n \geq N_1'$, with probability at least $1 - \beta/(4K)$,

$$\sup_{\pi\in\Pi}\left|K_1(\pi)\right| \leq \frac{C(\bar{\alpha}, \underline{\alpha}, \bar{\eta}, \underline{\eta})}{4\sqrt{n}}. \tag{18}$$

**Bounding $K_2(\pi)$.** We first note that by Cauchy-Schwarz inequality,

$$\left|\frac{1}{|\mathcal{D}^{(k)}|}\sum_{i\in\mathcal{D}^{(k)}}\left(\frac{\mathbb{1}\{A_i = \pi(X_i)\}}{\widehat{\pi}_0(A_i\,|\,X_i)} - \frac{\mathbb{1}\{A_i = \pi(X_i)\}}{\pi_0(A_i\,|\,X_i)}\right)\left(\bar{g}_{A_i}^{(k)}(X_i) - \widehat{g}_{A_i}^{(k)}(X_i)\right)\right|$$

$$\leq \frac{1}{|\mathcal{D}^{(k)}|\varepsilon^2}\sqrt{\sum_{i\in\mathcal{D}^{(k)}}\left(\widehat{\pi}_0^{(k)}(\pi(X_i)\,|\,X_i) - \pi_0(\pi(X_i)\,|\,X_i)\right)^2}\sqrt{\sum_{i\in\mathcal{D}^{(k)}}\left(\bar{g}_{\pi(X_i)}^{(k)}(X_i) - \widehat{g}_{\pi(X_i)}^{(k)}(X_i)\right)^2}$$

$$\leq \frac{1}{|\mathcal{D}^{(k)}|\varepsilon^2}\sqrt{\sum_{i\in\mathcal{D}^{(k)}}\sum_{a=1}^{M}\left(\widehat{\pi}_0^{(k)}(a\,|\,X_i) - \pi_0(a\,|\,X_i)\right)^2}\sqrt{\sum_{i\in\mathcal{D}^{(k)}}\sum_{a=1}^{M}\left(\bar{g}_a^{(k)}(X_i) - \widehat{g}_a^{(k)}(X_i)\right)^2}.$$

Then for any $t > 0$, let

$$s = \frac{M}{t\varepsilon^2}\max_{a\in[M]}\left\{\|\widehat{\pi}_a^{(k)} - \pi_{0,a}^{(k)}\|_{L_2(P_X)}\right\}\max_{a\in[M]}\left\{\|\bar{g}_a^{(k)} - \widehat{g}_a^{(k)}\|_{L_2(P_X)}\right\}.$$

Then

$$\mathbb{P}_k\left(\max_{\pi\in\Pi}\left|\frac{1}{|\mathcal{D}^{(k)}|}\sum_{i\in\mathcal{D}^{(k)}}\left(\frac{\mathbb{1}\{A_i = \pi(X_i)\}}{\widehat{\pi}_0(A_i\,|\,X_i)} - \frac{\mathbb{1}\{A_i = \pi(X_i)\}}{\pi_0(A_i\,|\,X_i)}\right)\left(\widehat{g}_{A_i}^{(k)}(X_i) - \bar{g}_{A_i}^{(k)}(X_i)\right)\right| \geq s\right)$$

$$\leq \mathbb{P}_k\left(\frac{1}{|\mathcal{D}^{(k)}|\varepsilon^2}\sqrt{\sum_{i\in\mathcal{D}^{(k)}}\sum_{a=1}^{M}\left(\widehat{\pi}_0^{(k)}(a\,|\,X_i) - \pi_0(a\,|\,X_i)\right)^2}\sqrt{\sum_{i\in\mathcal{D}^{(k)}}\sum_{a=1}^{M}\left(\widehat{g}_a^{(k)}(X_i) - \bar{g}_a^{(k)}(X_i)\right)^2} \geq s\right)$$

$$\leq \mathbb{P}_k\left(\frac{1}{\varepsilon}\sqrt{\frac{1}{|\mathcal{D}^{(k)}|}\sum_{i\in\mathcal{D}^{(k)}}\sum_{a=1}^{M}\left(\widehat{\pi}_0^{(k)}(a\,|\,X_i) - \pi_0(a\,|\,X_i)\right)^2} \geq \frac{\sqrt{M}}{\sqrt{t}\varepsilon}\max_{a\in[M]}\left\{\|\widehat{\pi}_a^{(k)} - \pi_{0,a}^{(k)}\|_{L_2(P_X)}\right\}\right)$$

$$+ \mathbb{P}\left(\frac{1}{\varepsilon}\sqrt{\frac{1}{|\mathcal{D}^{(k)}|}\sum_{i\in\mathcal{D}^{(k)}}\sum_{a=1}^{M}\left(\widehat{g}_a^{(k)}(X_i) - \bar{g}_a^{(k)}(X_i)\right)^2} \geq \frac{\sqrt{M}}{\sqrt{t}\varepsilon}\max_{a\in[M]}\left\{\|\widehat{g}_a^{(k)} - \widehat{g}_a^{(k)}\|_{L_2(P_X)}\right\}\right)$$

$$\leq 2t,$$

where the last inequality is due to Chebyshev's inequality. Marginalizing over the randomness of $\mathcal{D}^{(-k)}$, for any $\beta \in (0, 1)$, we have with probability at least $1 - \beta$ that

$$\max_{\pi \in \Pi} |K_2(\pi)| < \frac{2M}{\beta \varepsilon^2} \max_{a \in [M]} \left\{ \|\widehat{\pi}_a^{(k)} - \pi_{0,a}^{(k)}\|_{L_2(P_X)} \right\} \max_{a \in [M]} \left\{ \|\widehat{g}_a^{(k)} - \bar{g}_a^{(k)}\|_{L_2(P_X)} \right\}.$$

By Assumption 3.4, there exists $N_2' \in \mathbb{N}_+$ such that when $n \geq N_2'$, with probability at least $1 - \beta/(4K)$,

$$\sup_{\pi \in \Pi} |K_2(\pi)| \leq \frac{C(\bar{\alpha}, \underline{\alpha}, \bar{\eta}, \underline{\eta})}{4\sqrt{n}}. \tag{19}$$

**Bounding $K_3(\pi)$.** We start by taking

$$h_3(Z_i, \pi(X_i)) = \frac{\mathbb{1}\{A_i = \pi(X_i)\}}{\pi_0(A_i \mid X_i)} \cdot \left[ \widehat{G}_{\pi(X_i)}^{(k)}(X_i, Y_i(\pi(X_i))) - G_{\pi(X_i)}(X_i, Y_i(\pi(X_i))) \right].$$

For any $\pi \in \Pi$,

$$\mathbb{E}_k \left[ h_3(Z_i, \pi(X_i)) \mid X_i \right]$$

$$= \mathbb{E}_k \left[ \frac{\mathbb{1}\{A_i = \pi(X_i)\}}{\pi_0(A_i \mid X_i)} \cdot \left( \widehat{G}_{A_i}^{(k)}(X_i, Y_i(\pi(X_i))) - G_{A_i}(X_i, Y_i(\pi(X_i))) \right) \mid X_i \right]$$

$$= \mathbb{E}_k \left[ \widehat{G}_{\pi(X_i)}^{(k)}(X_i, Y_i(\pi(X_i))) - G_{\pi(X_i)}(X_i, Y_i(\pi(X_i))) \mid X_i \right]$$

$$= \mathbb{E}_k \left[ \ell\left(X_i, Y_i(\pi(X_i)); \boldsymbol{\theta}_{\pi(X_i)}^{(k)}(X_i)\right) - \ell\left(X_i, Y_i; \boldsymbol{\theta}_{\pi(X_i)}^*(X_i)\right) \right.$$

$$\left. - \nabla \ell\left(X_i, Y_i(\pi(X_i)); \boldsymbol{\theta}_{\pi(X_i)}^*(X_i)\right)^\top \left(\widehat{\boldsymbol{\theta}}_{\pi(X_i)}^{(k)}(X_i) - \boldsymbol{\theta}_{\pi(X_i)}^*(X_i)\right) \mid X_i \right],$$

where the last step follows from part (1) of Lemma B.1. By Assumption 3.4, for any $\beta \in (0, 1)$, there exists $N_3 \in \mathbb{N}_+$ such that when $n \geq N_3$,

$$\mathbb{P}\left( \max_{a \in [M]} \|\widehat{\boldsymbol{\theta}}_a^{(k)} - \boldsymbol{\theta}_a^*\|_{L_\infty} > \min\left(\xi, \bar{\alpha}, \underline{\alpha}, \bar{\eta}\right)/2 \right) \leq \beta.$$

On the event $\left\{ \max_{a \in [M]} \|\widehat{\boldsymbol{\theta}}_a^{(k)} - \boldsymbol{\theta}_a^*\|_{L_\infty} \leq \min(\xi, \bar{\alpha}, \underline{\alpha}, \bar{\eta})/2 \right\}$, we have

$$\left| \ell\left(X_i, Y_i; \boldsymbol{\theta}_{\pi(X_i)}^{(k)}(X_i)\right) - \ell\left(X_i, Y_i; \boldsymbol{\theta}_{\pi(X_i)}^*(X_i)\right) - \nabla \ell\left(X_i, Y_i; \boldsymbol{\theta}_{\pi(X_i)}^*(X_i)\right)^\top \left(\widehat{\boldsymbol{\theta}}_{\pi(X_i)}^{(k)} - \boldsymbol{\theta}_{\pi(X_i)}^*\right) \right|$$

$$\leq \bar{\ell}(X_i, Y_i) \cdot \sum_{a \in [M]} \left\| \widehat{\boldsymbol{\theta}}_a(X_i) - \boldsymbol{\theta}_a^*(X_i) \right\|_2^2,$$

As a result,

$$\sup_{\pi \in \Pi} \left| \mathbb{E}_k[K_3(\pi) \mid X] \right| \leq \sup_{\pi \in \Pi} \frac{1}{|\mathcal{D}^{(k)}|} \sum_{i \in \mathcal{D}^{(k)}} \mathbb{E}_k \left[ h_3(Z_i, \pi(X_i)) \mid X_i \right]$$

$$\leq \frac{L}{|\mathcal{D}^{(k)}|} \sum_{i \in \mathcal{D}^{(k)}} \sum_{a \in [M]} \left\| \widehat{\boldsymbol{\theta}}_a(X_i) - \boldsymbol{\theta}_a^*(X_i) \right\|_2^2.$$

On the same event,

$$\left| h_3(Z_i, \pi(X_i)) \right| = \left| \frac{\mathbb{1}\{A_i = \pi(X_i)\}}{\pi_0(A_i \mid X_i)} \cdot \left\{ \ell\left(X_i, Y_i(\pi(X_i)); \boldsymbol{\theta}_{\pi(X_i)}^{(k)}(X_i)\right) - \ell\left(X_i, Y_i; \boldsymbol{\theta}_{\pi(X_i)}^*(X_i)\right) \right\} \right|$$

$$\leq \frac{1}{\varepsilon} \left| \nabla \ell\left(X_i, Y_i(\pi(X_i)); \tilde{\boldsymbol{\theta}}_{\pi(X_i)}(X_i)\right)^\top \left(\boldsymbol{\theta}_{\pi(X_i)}^{(k)}(X_i) - \boldsymbol{\theta}_{\pi(X_i)}^*(X_i)\right) \right|$$

$$\leq \frac{1}{\varepsilon} \left\| \nabla \ell\left(X_i, Y_i(\pi(X_i)); \tilde{\boldsymbol{\theta}}_{\pi(X_i)}(X_i)\right) \right\|_2 \left\| \widehat{\boldsymbol{\theta}}_{\pi(X_i)}^{(k)}(X_i) - \boldsymbol{\theta}_{\pi(X_i)}^*(X_i) \right\|_2$$

$$\leq C_2(\bar{\alpha}, \underline{\alpha}, \bar{\eta}, \delta, \varepsilon) \max_{a \in [M]} \left\| \widehat{\boldsymbol{\theta}}_a^{(k)}(X_i) - \boldsymbol{\theta}_a^*(X_i) \right\|_2,$$

where $C_2(\bar{\alpha}, \underline{\alpha}, \bar{\eta}, \delta, \varepsilon) = (1 + (\bar{y} + \bar{\eta})/\underline{\alpha})e^{(\bar{y}+\bar{\eta})/\underline{\alpha}-1} + \delta + 1$ is a constant. Let $\bar{h}_3(Z_i, \pi(X_i)) = h_3(Z_i, \pi(X_i)) - \mathbb{E}_k[h_3(Z_i, \pi(X_i)) \mid X_i]$, and we have that

$$|\bar{h}_3(Z_i, \pi(X_i))| \leq 2C_2(\bar{\alpha}, \underline{\alpha}, \bar{\eta}, \delta, \varepsilon) \max_{a \in [M]} \left\| \widehat{\boldsymbol{\theta}}_a^{(k)}(X_i) - \boldsymbol{\theta}_a^*(X_i) \right\|_2.$$

Next, we apply the bounded difference theorem conditional on $X_i$'s:

$$\mathbb{P}_k \left( \sup_{\pi \in \Pi} \left| \frac{1}{|\mathcal{D}^{(k)}|} \sum_{i \in \mathcal{D}^{(k)}} \bar{h}_3(Z_i, \pi(X_i)) \right| - \mathbb{E}_k \left[ \sup_{\pi \in \Pi} \left| \frac{1}{|\mathcal{D}^{(k)}|} \sum_{i \in \mathcal{D}^{(k)}} \bar{h}_3(Z_i, \pi(X_i)) \right| \, \bigg| \, X \right] \geq t \, \bigg| \, X \right)$$

$$\leq \exp\left( -\frac{|\mathcal{D}^{(k)}|^2 t^2}{2C_2(\bar{\alpha}, \underline{\alpha}, \bar{\eta}, \delta, \varepsilon)^2 \sum_{i \in \mathcal{D}^{(k)}} \max_{a \in [M]} \left\| \widehat{\boldsymbol{\theta}}_a^{(k)}(X_i) - \boldsymbol{\theta}_a^*(X_i) \right\|_2^2} \right),$$

for any $t > 0$. Taking $t = C_2(\bar{\alpha}, \underline{\alpha}, \bar{\eta}, \delta, \varepsilon)\sqrt{2 \sum_{i \in \mathcal{D}^{(k)}} \max_{a \in [M]} \left\| \widehat{\boldsymbol{\theta}}_a^{(k)}(X_i) - \boldsymbol{\theta}_a^*(X_i) \right\|_2^2 / |\mathcal{D}^{(k)}|}$, we have with probability at least $1 - \beta$ that

$$\sup_{\pi \in \Pi} \left| \frac{1}{|\mathcal{D}^{(k)}|} \sum_{i \in \mathcal{D}^{(k)}} \bar{h}_3(Z_i, \pi(X_i)) \right| \leq \mathbb{E}_k \left[ \sup_{\pi \in \Pi} \left| \frac{1}{|\mathcal{D}^{(k)}|} \sum_{i \in \mathcal{D}^{(k)}} \bar{h}_3(Z_i, \pi(X_i)) \right| \, \bigg| \, X \right]$$

$$+ \frac{C_2(\bar{\alpha}, \underline{\alpha}, \bar{\eta}, \delta, \varepsilon)}{|\mathcal{D}^{(k)}|} \sqrt{2 \sum_{i \in \mathcal{D}^{(k)}} \sum_{a \in [M]} \left\| \widehat{\boldsymbol{\theta}}_a^{(k)}(X_i) - \boldsymbol{\theta}_a^*(X_i) \right\|_2^2}.$$

For the expectation term, the same symmetrization argument as in the proof for $K_1(\pi)$ leads to

$$\mathbb{E}_k \left[ \sup_{\pi \in \Pi} \left| \frac{1}{|\mathcal{D}^{(k)}|} \sum_{i \in \mathcal{D}^{(k)}} \bar{h}_3(Z_i, \pi(X_i)) \right| \, \bigg| \, X \right] \leq 2\mathbb{E} \left[ \sup_{\pi \in \Pi} \left| \frac{1}{|\mathcal{D}^{(k)}|} \sum_{i \in \mathcal{D}^{(k)}} \left| \epsilon_i \bar{h}_3(Z_i, \pi(X_i)) \right| \, \bigg| \, X \right].$$

Then by Lemma B.2, we have

$$\mathbb{E}_\epsilon \left[ \sup_{\pi \in \Pi} \left| \frac{1}{|\mathcal{D}^{(k)}|} \sum_{i \in \mathcal{D}^{(k)}} \left| \epsilon_i \bar{h}_3(Z_i, \pi(X_i)) \right| \right]$$

$$\leq \frac{2C_2(\bar{\alpha}, \underline{\alpha}, \bar{\eta}, \delta, \varepsilon)}{|\mathcal{D}^{(k)}|} (32 + \kappa(\Pi)) \sqrt{\sum_{i \in \mathcal{D}^{(k)}} \sum_{a \in [M]} \|\widehat{\boldsymbol{\theta}}_a^{(k)}(X_i) - \boldsymbol{\theta}_a^*(X_i)\|_2^2}.$$

By Hoeffding's inequality, we have that

$$\mathbb{P}_k \left( \frac{1}{|\mathcal{D}^{(k)}|} \sum_{i \in \mathcal{D}^{(k)}} \|\widehat{\boldsymbol{\theta}}_a^{(k)}(X_i) - \boldsymbol{\theta}_a^*(X_i)\|_2^2 - \|\boldsymbol{\theta}_a^{(k)} - \boldsymbol{\theta}_a\|_{L_2(P_X)}^2 \geq \|\boldsymbol{\theta}_a^{(k)} - \boldsymbol{\theta}_a\|_{L_\infty}^2 \sqrt{\frac{1}{2n} \log\left(\frac{1}{\beta}\right)} \right) \leq \beta.$$

Taking a union bound, with probability at least $1 - 3\beta$, we have that

$$\sup_{\pi \in \Pi} \left| K_3(\pi) \right|$$

$$\leq \frac{C_2(\bar{\alpha}, \underline{\alpha}, \bar{\eta}, \delta, \varepsilon)(130 + 4\kappa(\Pi))}{\sqrt{|\mathcal{D}^{(k)}|}} \left( \sum_{a \in [M]} \|\widehat{\boldsymbol{\theta}}_a^{(k)} - \boldsymbol{\theta}_a\|_{L_2(P_X)} + \sqrt{M(\bar{\alpha} + \bar{\eta})} \left( \frac{1}{|\mathcal{D}^{(k)}|} \log\left(\frac{M}{\beta}\right) \right)^{1/4} \right)$$

$$+ L \left( \sum_{a \in [M]} \|\widehat{\boldsymbol{\theta}}_a^{(k)} - \boldsymbol{\theta}_a\|_{L_2(P_X)}^2 + \frac{M\|\widehat{\boldsymbol{\theta}}_a - \boldsymbol{\theta}_a\|_{L_\infty} \sqrt{\log(M/\beta)}}{\sqrt{|\mathcal{D}^{(k)}|}} \right).$$

By Assumption 3.4, $\|\widehat{\boldsymbol{\theta}}_a^{(k)} - \boldsymbol{\theta}_a\|_{L_2(P_X)} = o_P(n^{-1/4})$ and $\|\widehat{\boldsymbol{\theta}}_a^{(k)} - \boldsymbol{\theta}_a\|_{L_\infty} = o_P(1)$, so there exists $N_3' \geq N_3$ such that when $n \geq N_3'$, with probability at least $1 - \beta/(4K)$,

$$\sup_{\pi \in \Pi} \left| K_3(\pi) \right| \leq \frac{C(\bar{\alpha}, \underline{\alpha}, \bar{\eta}, \underline{\eta})}{4\sqrt{n}}. \tag{20}$$

**Bounding $K_4(\pi)$.** For $K_4(\pi)$, we take

$$h_4(Z_i, \pi(X_i)) = -\frac{\mathbb{1}\{A_i = \pi(X_i)\}}{\pi_0(A_i \mid X_i)}\big(\widehat{g}^{(k)}_{\pi(X_i)}(X_i) - g_{\pi(X_i)}(X_i)\big) + \big(\widehat{g}^{(k)}_{\pi(X_i)}(X_i) - g_{\pi(X_i)}(X_i)\big).$$

and therefore $K_4(\pi) = \frac{1}{|\mathcal{D}|}\sum_{i \in \mathcal{D}^{(k)}} h_4(Z_i, \pi(X_i))$. Again by the unconfoundedness assumption,

$$\mathbb{E}_k\big[h_4(Z_i, \pi(X_i))\big] = 0.$$

Due to the overlap condition, we further have that

$$|h_4(Z_i, \pi(X_i))| \leq \frac{2}{\varepsilon}\big|\widehat{g}^{(k)}_{\pi(X_i)}(X_i) - g_{\pi(X_i)}(X_i)\big| \leq \frac{2}{\varepsilon}\max_{a \in [M]}\big|\widehat{g}^{(k)}_a(X_i) - g_a(X_i)\big|.$$

As before, we apply the bounded difference theorem conditional on $X_i$'s and the symmetrization argument to obtain

$$\mathbb{P}\bigg(\sup_{\pi \in \Pi}\big|K_4(\pi)\big| - 2\mathbb{E}_k\Big[\sup_{\pi \in \Pi}\Big|\frac{1}{|\mathcal{D}^{(k)}|}\sum_{i \in \mathcal{D}^{(k)}} \epsilon_i h_4(Z_i, \pi(X_i))\Big|\,\Big|\,X\Big] \geq t\,\Big|\,X\bigg)$$

$$\leq \mathbb{P}\bigg(\sup_{\pi \in \Pi}\big|K_4(\pi)\big| - \mathbb{E}_k\Big[\sup_{\pi \in \Pi}\big|K_4(\pi)\big|\,\big|\,X\Big] \geq t \mid X\bigg)$$

$$\leq \exp\left(-\frac{\varepsilon^2|\mathcal{D}^{(k)}|^2 t^2}{2\sum_{i \in \mathcal{D}^{(k)}}\max_{a \in [M]}(\widehat{g}^{(k)}_a(X_i) - g_a(X_i))^2}\right).$$

We now apply Lemma B.2:

$$\mathbb{E}_\epsilon\left[\sup_{\pi \in \Pi}\Big|\frac{1}{|\mathcal{D}^{(k)}|}\sum_{i \in \mathcal{D}^{(k)}}\epsilon_i h_4(Z_i, \pi(X_i))\Big|\,\Big|\,X\right] \leq \frac{\sqrt{\sum_{i \in \mathcal{D}^{(k)}}\max_{a \in [M]}\big(\widehat{g}_a(X_i) - g_a(X_i)\big)^2}}{|\mathcal{D}^{(k)}|\varepsilon}(64 + 8\kappa(\Pi)).$$

By Assumption 3.4, there exists $N_4 \in \mathbb{N}_+$, such that when $n \geq N_4$, with probability at least $1 - \beta$,

$$\max_{a \in [M]}\|\widehat{\boldsymbol{\theta}}^{(k)}_a - \boldsymbol{\theta}^*_a\|_{L_\infty} \leq \max(\xi, \bar{\alpha}, \underline{\alpha}, \bar{\eta})/2.$$

On the event $\{\max_{a \in [M]}\|\widehat{\boldsymbol{\theta}}^{(k)}_a - \boldsymbol{\theta}^*_a\|_{L_\infty} \leq \max(\xi, \bar{\alpha}, \underline{\alpha}, \bar{\eta})/2\}$, we have for any $a \in [M]$ that

$$\big|\ell(x, y; \widehat{\boldsymbol{\theta}}_a(x))\big| \leq 2C_1(\bar{\alpha}, \underline{\alpha}, \bar{\eta}, \delta, \varepsilon).$$

On the same event, by Hoeffding's inequality, we have that

$$\mathbb{P}_k\left(\frac{1}{|\mathcal{D}^{(k)}|}\sum_{i \in \mathcal{D}^{(k)}}(\widehat{g}_a(X_i) - g_a(X_i))^2 - \|\widehat{g}^{(k)}_a - g_a\|^2_{L_2(P_X)} \geq t\right) \leq \exp\left(-\frac{t^2|\mathcal{D}^{(k)}|}{8C_1(\bar{\alpha}, \underline{\alpha}, \bar{\eta}, \delta, \varepsilon)^2}\right).$$

Taking a union bound, we have with probability at least $1 - 2\beta$ that

$$\max_{\pi \in \Pi}\big|K_4(\pi)\big| \leq \frac{1}{\varepsilon\sqrt{|\mathcal{D}^{(k)}|}}\Big(128 + 16\kappa(\Pi) + \sqrt{2\log(1/\beta)}\Big)$$

$$\times\Big(\sum_{a \in [M]}\|\widehat{g}^{(k)}_a - g^*_a\|_{L_2(P_X)} + 2M\sqrt{C_1(\bar{\alpha}, \underline{\alpha}, \bar{\eta}, \delta, \varepsilon)}(\log(M/\beta)/n)^{1/4}\Big).$$

By Assumption 3.4, $\sum_{a \in [M]}\|\widehat{g}^{(k)}_a - g^*_a\|_{L_2(P_X)} = o_P(1)$, so there exists $N'_4 \geq N_4$ such that when $n \geq N'_4$, with probability at least $1 - \beta/(4K)$,

$$\sup_{\pi \in \Pi}\big|K_4(\pi)\big| \leq \frac{C(\bar{\alpha}, \underline{\alpha}, \bar{\eta}, \eta)}{4\sqrt{n}}. \tag{21}$$

Combining (17)-(21) and taking a union bound over $k \in [K]$, when $n \geq \max(N_1, N_2, N_3, N_4)$ we have that with probability at least $1 - \beta$,

$$\sup_{\pi \in \Pi}\big|\widehat{\mathcal{V}}^{\mathsf{LN}}_\delta(\pi) - \tilde{\mathcal{V}}_\delta(\pi)\big| \leq \frac{C(\bar{\alpha}, \underline{\alpha}, \bar{\eta}, \eta)}{\sqrt{n}}.$$

We have thus completed the proof of Theorem 4.2.

## B.4 PROOF OF THEOREM 4.5

We first state somes results from Si et al. (2023) that will be used in the proof. For any $p, q \in [0, 1]$, define

$$D(p \,\|\, q) = p \log\left(\frac{p}{q}\right) + (1 - p) \log\left(\frac{1 - p}{1 - q}\right), \text{ and } g_\delta(q) = \inf_{p:D_{\mathrm{KL}}(p \,\|\, q) \leq \delta} p,$$

**Lemma B.3** (Adapted from Lemma A17 of Si et al. (2023))**.** *For $\delta \leq 0.2$, $g_\delta(q)$ is differentiable and $g'_\delta(q) \geq 1/2$ for $q \in [0.4, 0.6]$.*

Note that our definition of $g_\delta(q)$ is slightly different from that in Si et al. (2023), so we include the proof of Lemma B.3 in Appendix C.4 for completeness.

For notational simplicity, we use $d$ to denote the Natarajan dimension of the policy class $\Pi$. By the definition of Natarajan dimension, there exists a set of $d$ data points $\{x_1, \ldots, x_d\} \subseteq \mathcal{X}$ shattered by $\Pi$: there exist two functions $f_{-1}, f_1 : \{x_1, \ldots, x_d\} \mapsto [M]$ such that $f_{-1}(x_j) \neq f_1(x_j)$ for any $j \in [d]$ and for any $\sigma \in \{-1.1\}^d$, there exists $\pi \in \Pi$, such that $\pi(x_j) = f_{\sigma_j}(x_j)$ for all $j \in [d]$.

Next, we construct a class of distributions indexed by $\sigma \in \{\pm 1\}^d$ that are "hard instances" for the learning problem. Fix any $\sigma \in \{\pm 1\}^d$, we construct distribution $P_\sigma$ as follows. First, the covariate are drawn uniformly from $\{x_1, \ldots, x_d\}$, i.e.,

$$X_i \overset{\text{i.i.d.}}{\sim} \mathrm{Unif}(\{x_1, \ldots, x_d\}).$$

Given $X_i$, the action $A_i$ is chosen according to the behavior policy $\pi_0$, where for any $j \in [d]$,

$$\pi_0(f_1(x_j) \,|\, x_j) = \pi_0(f_{-1}(x_j) \,|\, x_j) = \frac{\varepsilon}{2}, \text{ and } \pi_0(a \,|\, x_j) = \frac{1 - \varepsilon}{K - 2} \text{ for all } a \neq f_1(x_j), f_{-1}(x_j).$$

The potential outcomes are generated as follows:

$$Y_i(f_1(x_j)) \,|\, X_i = x_j \sim \bar{y} \cdot \mathrm{Bern}\left(\frac{1 + \sigma_j \Delta}{2}\right), \; Y_i(f_{-1}(x_j)) \,|\, X_i = x_j \sim \bar{y} \cdot \mathrm{Bern}\left(\frac{1 - \sigma_j \Delta}{2}\right),$$

$$\text{and } Y_i(a) = \bar{y} \cdot \mathrm{Bern}(1/4) \text{ for all } a \neq f_1(x_j), f_{-1}(x_j),$$

where $\Delta \in (0, 0.1)$ is some constant to be determined later. Note that the distribution of $(X_i, A_i)$ does not depend on $\sigma$. By construction, it is clear that the data-generating process satisfies Assumption 2.1. For any $p \in \{(1 + \Delta)/2, (1 - \Delta)/2, 1/4\}$, $\log(1/(1 - p)) > \delta$. Therefore, the data-generating process also satisfies Assumption 2.4. As for Assumption 3.3, it suffices to check the Bernoulli distributions with parameters $(1 + \Delta)/2, (1 - \Delta)/2, 1/4$, and can be verified. Since $n \geq d^2$, we can obtain $\widehat{\theta}, \widehat{g}$, and $\widehat{\pi}_0$ that converges at rate $O_P(n^{-1/4})$ (by stratifying on $X$), thereby satisfying Assumption 3.4.

We now proceed to establish the lower bound. For any policy learning algorithm that returns $\widehat{\pi}$, the worst-case regret is lower bounded by the average regret over the class of hard instances we have constructed above:

$$\sup_{P \in \mathcal{P}} \mathbb{E}_{P^n}[\mathcal{R}(\widehat{\pi})] \geq \frac{1}{2^d} \sum_{\sigma \in \{\pm 1\}^d} \mathbb{E}_{P_\sigma^n}[\mathcal{R}_\delta(\widehat{\pi})].$$

We now focus on the right-hand side above. Fix $\sigma \in \{\pm 1\}^d$. Recall that $\mathcal{R}_\delta(\widehat{\pi}) = \mathcal{V}_\delta(\pi^*) - \mathcal{V}_\delta(\widehat{\pi})$. For the optimal policy value, there is

$$\mathcal{V}_\delta(\pi^*) = \max_{\pi \in \Pi} \mathbb{E}_{P_{\sigma, X}}\left[\inf_{Q_{Y \,|\, X} \in \mathcal{P}(P_{\sigma, Y \,|\, X}, \delta)} \mathbb{E}_{Q_{Y \,|\, X}}\left[Y(\pi(X)) \,|\, X\right]\right]$$

$$\overset{(i)}{=} \max_{\pi \in \Pi} \max_{\boldsymbol{\alpha}, \boldsymbol{\eta}} \mathbb{E}_{P_\sigma}\left[-\boldsymbol{\alpha}(X) \exp\left(-\frac{Y(\pi(X)) + \boldsymbol{\eta}(X)}{\boldsymbol{\alpha}(X)} - 1\right) - \boldsymbol{\eta}(X) - \boldsymbol{\alpha}(X)\delta\right]$$

$$= \max_{\boldsymbol{\alpha}, \boldsymbol{\eta}} \max_{\pi \in \Pi} \mathbb{E}_{P_\sigma}\left[-\boldsymbol{\alpha}(X) \exp\left(-\frac{Y(\pi(X)) + \boldsymbol{\eta}(X)}{\boldsymbol{\alpha}(X)} - 1\right) - \boldsymbol{\eta}(X) - \boldsymbol{\alpha}(X)\delta\right], \quad (22)$$

where the step (i) follows from the duality result in Proposition 2.3. We now take a closer look at the expectation above: by the construction of $P_\sigma$,

$$\mathbb{E}_{P_\sigma}\left[-\boldsymbol{\alpha}(X)\exp\left(-\frac{Y(\pi(X))+\boldsymbol{\eta}(X)}{\boldsymbol{\alpha}(X)}-1\right)-\boldsymbol{\eta}(X)-\boldsymbol{\alpha}(X)\delta\right]$$

$$=\frac{1}{d}\sum_{j=1}^{d}\mathbb{E}_{P_\sigma}\left[-\boldsymbol{\alpha}(x_j)\exp\left(-\frac{Y(\pi(x_j))+\boldsymbol{\eta}(x_j)}{\boldsymbol{\alpha}(x_j)}-1\right)-\boldsymbol{\eta}(x_j)-\boldsymbol{\alpha}(x_j)\delta\,\Big|\,X=x_j\right]$$

$$=\frac{1}{d}\sum_{j=1}^{d}-\boldsymbol{\alpha}(x_j)\exp\left(-\frac{\boldsymbol{\eta}(x_j)}{\boldsymbol{\alpha}(x_j)}-1\right)\cdot\mathbb{E}_P\left[\exp\left(-\frac{Y(\pi(x_j))}{\boldsymbol{\alpha}(x_j)}\right)\,\Big|\,X=x_j\right]-\boldsymbol{\eta}(x_j)-\boldsymbol{\alpha}(x_j)\delta.$$

Letting $p_j=\mathbb{P}(Y(\pi(x_j))=1\,|\,X=x_j)$, we have

$$\mathbb{E}_P\left[\exp\left(-\frac{Y(\pi(x_j))}{\boldsymbol{\alpha}(x_j)}\right)\,\Big|\,X=x_j\right]=p_j\exp(-1/\boldsymbol{\alpha}(x_j))+1-p_j,$$

which is decreasing in $p_j$ and is minimized when $\pi(x_j)=f_{\sigma_j}(x_j)$. By construction, such a policy $\pi$ is in $\Pi$. As a result,

$$(22)=\max_{\boldsymbol{\alpha},\boldsymbol{\eta}}\frac{1}{d}\sum_{j=1}^{d}\mathbb{E}_{P_\sigma}\left[-\boldsymbol{\alpha}(x_j)\exp\left(-\frac{Y(f_{\sigma_j}(x_j))+\boldsymbol{\eta}(x_j)}{\boldsymbol{\alpha}(x_j)}-1\right)-\boldsymbol{\eta}(x_j)-\boldsymbol{\alpha}(x_j)\delta\,\Big|\,X=x_j\right]$$

$$=\frac{1}{d}\sum_{j=1}^{d}\max_{\alpha,\eta}\mathbb{E}_{P_\sigma}\left[-\alpha\exp\left(-\frac{Y(f_{\sigma_j}(x_j))+\eta}{\alpha}-1\right)-\eta-\alpha\delta\,\Big|\,X=x_j\right]$$

$$=\frac{1}{d}\sum_{j=1}^{d}\inf_{Q_{Y\,|\,X}\in\mathcal{P}(P_{\sigma,Y\,|\,X=x_j},\delta)}\mathbb{E}_{Q_{Y\,|\,X}}\left[Y(f_{\sigma_j}(x_j))\,|\,X=x_j\right]=g\left(\frac{1+\Delta}{2}\right).$$

The last step is because $Y(f_{\sigma_j}(x_j))\,|\,X=x_j\sim\text{Bern}((1+\Delta)/2)$. Similarly, for $\mathcal{V}(\widehat{\pi})$, we have

$$\mathcal{V}_\delta(\widehat{\pi})=\mathbb{E}_{P_{\sigma,X}}\left[\inf_{Q_{Y\,|\,X}\in\mathcal{P}(P_{\sigma,Y\,|\,X},\delta)}\mathbb{E}_{Q_{Y\,|\,X}}\left[Y(\widehat{\pi}(X))\,|\,X\right]\right]$$

$$=\frac{1}{d}\sum_{j=1}^{d}\inf_{Q_{Y\,|\,X}\in\mathcal{P}(P_{\sigma,Y\,|\,X=x_j},\delta)}\mathbb{E}_{Q_{Y\,|\,X}}\left[Y(\widehat{\pi}(x_j))\,|\,X=x_j\right]$$

$$=\frac{1}{d}\sum_{j=1}^{d}\mathbb{1}\{\widehat{\pi}(x_j)=f_{\sigma_j}(x_j)\}g\left(\frac{1+\Delta}{2}\right)+\mathbb{1}\{\widehat{\pi}(x_j)=f_{-\sigma_j}(x_j)\}g\left(\frac{1-\Delta}{2}\right)$$

$$+\mathbb{1}\{\widehat{\pi}(x_j)\neq f_{\sigma_j}(x_j),f_{-\sigma_j}(x_j)\}g(1/4).$$

Combining the calculation above, we have

$$\mathcal{R}(\widehat{\pi})=\frac{1}{d}\sum_{j=1}^{d}\mathbb{1}\{\widehat{\pi}(x_j)=f_{-\sigma_j}(x_j)\}\cdot\left\{g\left(\frac{1+\Delta}{2}\right)-g\left(\frac{1-\Delta}{2}\right)\right\}$$

$$+\mathbb{1}\{\widehat{\pi}(x_j)\neq f_{\sigma_j}(x_j),f_{-\sigma_j}(x_j)\}\cdot\left\{g\left(\frac{1+\Delta}{2}\right)-g(1/4)\right\}$$

$$\overset{(i)}{\geq}\frac{1}{d}\sum_{j=1}^{d}+\mathbb{1}\{\widehat{\pi}(x_j)\neq f_{\sigma_j}(x_j)\}\cdot\left\{g\left(\frac{1+\Delta}{2}\right)-g(1/4)\right\}$$

$$\overset{(ii)}{\geq}\frac{1}{d}\sum_{j=1}^{d}\mathbb{1}\{\widehat{\pi}(x_j)\neq f_{\sigma_j}(x_j)\}\cdot g'(\xi)\Delta\overset{(iii)}{\geq}\frac{\Delta}{2d}\sum_{j=1}^{d}\mathbb{1}\{\widehat{\pi}(x_j)\neq f_{\sigma_j}(x_j)\},$$

where step (i) uses that $g$ is non-decreasing (c.f. Cauchois et al. (2024, Proposition 1)); in step (ii), $\xi\in((1-\Delta)/2,(1+\Delta)/2)$, and step (iii) follows from Lemma B.3.

Next, we denote $\sigma[j]$ to be the vector $\sigma$ with the $j$-th element flipped. Then, we have

$$
\frac{1}{2^d} \sum_{\sigma \in \{\pm 1\}^d} \mathbb{E}_{P_\sigma^n}[\mathcal{R}(\widehat{\pi})] \geq \frac{1}{2^d} \sum_{\sigma \in \{\pm 1\}^d} \mathbb{E}_{P_\sigma^n}\left[ \frac{\Delta}{2d} \sum_{j=1}^d \mathbb{1}\{\widehat{\pi}(x_j) \neq f_{\sigma_j}(x_j)\} \right]
$$

$$
= \frac{\Delta}{d2^{d+1}} \sum_{j=1}^d \sum_{\sigma:\sigma_j=1} \left\{ \mathbb{P}_{P_\sigma^n}\big(\widehat{\pi}(x_j) \neq f_1(x_j)\big) + \mathbb{P}_{P_{\sigma[j]}^n}\big(\widehat{\pi}(x_j) \neq f_{-1}(x_j)\big) \right\}
$$

$$
\geq \frac{\Delta}{d2^{d+1}} \sum_{j=1}^d \sum_{\sigma:\sigma_j=1} \mathbb{P}_{P_\sigma^n}\big(\widehat{\pi}(x_j) \neq f_1(x_j)\big) + \mathbb{P}_{P_{\sigma[j]}^n}\big(\widehat{\pi}(x_j) = f_1(x_j)\big)
$$

$$
\geq \frac{\Delta}{d2^{d+1}} \sum_{j=1}^d \sum_{\sigma:\sigma_j=1} \big(1 - D_{\mathrm{TV}}(P_\sigma^n, P_{\sigma[j]}^n)\big), \tag{23}
$$

where the last step follows from the definition of the TV distance. By Pinsker's inequality, there is

$$
D_{\mathrm{TV}}^2\big(P_\sigma^n, P_{\sigma[j]}^n\big) \leq \frac{1}{2} D_{\mathrm{KL}}\big(P_\sigma^n \,\|\, P_{\sigma[j]}^n\big)
$$

$$
= \frac{1}{2} \sum_{i=1}^n \mathbb{E}_{P_\sigma}\left[ \log\left( \frac{\mathrm{d}P_\sigma}{\mathrm{d}P_{\sigma[j]}}(X_i, A_i, Y_i) \right) \right]
$$

$$
= \frac{1}{2} \sum_{i=1}^n \mathbb{E}_{P_\sigma}\left[ \mathbb{1}\big\{X_i = x_j, A_i = f_{\pm 1}(x_j)\big\} \cdot \Delta \log\left( \frac{1+\Delta}{1-\Delta} \right) \right]
$$

$$
\leq \frac{3n\varepsilon}{2d} \Delta^2,
$$

where the last step follows from $x \log(\frac{1+x}{1-x}) \leq 3x^2$, for $x \in (0, 1/3)$. Take $\Delta = \frac{1}{15}\sqrt{\frac{d}{n\varepsilon}}$ — this is possible since $n \geq d^2$ and $d \geq 4/(9\varepsilon)$ and then

$$
(23) \geq \sqrt{\frac{d}{n}} \frac{d2^{d-1}}{15d2^{d+2}} = \frac{1}{120} \times \sqrt{\frac{d}{n\varepsilon}}.
$$

## C  PROOF OF TECHNICAL LEMMAS

### C.1  PROOF OF LEMMA B.1

**Proof of (1).**  Given $\theta$, recall that our loss function is

$$
\ell(x, y; \theta) = \alpha \exp\left( -\frac{y+\eta}{\alpha} - 1 \right) + \eta + \alpha\delta.
$$

By the strong duality, $\mathbb{E}[\ell(X, Y(\pi(X)); \theta) \,|\, X]$ is convex in $\theta$; by Proposition 2.5, the first-order condition of convex optimization problem implies

$$
\nabla_\theta \mathbb{E}\left[ \ell\big(x, Y(\pi(x)); \boldsymbol{\theta}_\pi^*(x)\big) \,\big|\, X = x \right] = 0.
$$

Meanwhile, we can compute the gradient of $\ell(x, y; \theta)$ as

$$
\frac{\partial}{\partial \alpha} \ell(x, y; \theta) = \left( 1 + \frac{y+\eta}{\alpha} \right) \cdot \exp\left( -\frac{y+\eta}{\alpha} - 1 \right) + \delta,
$$

$$
\frac{\partial}{\partial \eta} \ell(x, y; \theta) = 1 - \exp\left( -\frac{y+\eta}{\alpha} - 1 \right). \tag{24}
$$

For any $a$ such that $|a - \boldsymbol{\alpha}_\pi^*(x)| \leq \boldsymbol{\alpha}_\pi^*(x)$, we have

$$
\left| \frac{\partial}{\partial \alpha} \ell(x, y; (a, \boldsymbol{\eta}_\pi^*(x))) \right| \leq \left( 1 + \frac{2(\bar{y}+\bar{\eta})}{\underline{\alpha}} \right) \cdot \exp\left( \frac{2(\bar{y}+\bar{\eta})}{\underline{\alpha}} - 1 \right) + \delta < \infty.
$$

By the mean value theorem and the dominated convergence theorem, we can change the order of expectation and taking limits and therefore

$$\mathbb{E}\left[\frac{\partial}{\partial \alpha}\ell\big(x, Y(\pi(x)); \boldsymbol{\theta}_\pi^*(x)\big) \,\big|\, X = x\right] = \frac{\partial}{\partial \alpha}\mathbb{E}\left[\ell\big(x, Y(\pi(x)); \boldsymbol{\theta}_\pi^*(x)\big) \,\big|\, X = x\right] = 0.$$

Similarly, since $\frac{\partial}{\partial \eta}\ell(x, y; (\boldsymbol{\alpha}_\pi^*(x), \eta))$ is non-decreasing in $\eta$, for $|\eta - \boldsymbol{\eta}_\pi^*(x)| \leq 1$,

$$\left|\frac{\partial}{\partial \eta}\ell(x, y; (\boldsymbol{\alpha}_\pi^*(x), \eta))\right| \leq \max\left\{\left|\frac{\partial}{\partial \eta}\ell(x, y; (\boldsymbol{\alpha}_\pi^*(x), \boldsymbol{\eta}_\pi^*(x) + 1))\right|, \left|\frac{\partial}{\partial \eta}\ell(x, y; (\boldsymbol{\alpha}_\pi^*(x), \boldsymbol{\eta}_\pi^*(x) - 1))\right|\right\},$$

with the right-hand side being integrable under $P_{Y\,|\,X}$. Agian by the mean-value theorem and the dominated convergence theorem,

$$\mathbb{E}\left[\frac{\partial}{\partial \eta}\ell\big(x, Y(\pi(x)); \boldsymbol{\theta}_\pi^*(x)\big) \,\big|\, X = x\right] = \frac{\partial}{\partial \eta}\mathbb{E}\left[\ell\big(x, Y(\pi(x)); \boldsymbol{\theta}_\pi^*(x)\big) \,\big|\, X = x\right] = 0.$$

We have thus completed the proof part (1) of Lemma B.1.

**Proof of (2).** We now compute the Hessian of $\ell(x, y; \theta)$:

$$\frac{\partial^2}{\partial \alpha^2}\ell(x, y; \theta) = \frac{(y + \eta)^2}{\alpha^3}\exp\left(-\frac{y + \eta}{\alpha} - 1\right),$$

$$\frac{\partial^2}{\partial \alpha \partial \eta}\ell(x, y; \theta) = -\frac{y + \eta}{\alpha^2}\exp\left(-\frac{y + \eta}{\alpha} - 1\right),$$

$$\frac{\partial^2}{\partial \eta^2}\ell(x, y; \theta) = \frac{1}{\alpha}\exp\left(-\frac{y + \eta}{\alpha} - 1\right).$$

By the Taylor expansion,

$$\ell(x, y; \theta) - \ell(x, y; \boldsymbol{\theta}_\pi^*(x)) = \nabla\ell(x, y; \boldsymbol{\theta}_\pi^*(x))^\top (\theta - \theta_\pi^*(x)) + \frac{1}{2}(\theta - \boldsymbol{\theta}_\pi^*(x))^\top \nabla^2\ell(x, y; \tilde{\theta})(\theta - \boldsymbol{\theta}_\pi^*(x)),$$

$$\Rightarrow \left|\ell(x, y; \theta) - \ell(x, y; \boldsymbol{\theta}_\pi^*(x)) - \nabla\ell(x, y; \boldsymbol{\theta}_\pi^*(x))^\top (\theta - \boldsymbol{\theta}_\pi^*(x))\right|$$

$$\leq \frac{1}{2}\left(\frac{(y + \tilde{\eta})^2}{\tilde{\alpha}^3} + \frac{1}{\tilde{\alpha}}\right)\exp\left(-\frac{y + \tilde{\eta}}{\tilde{\alpha}} - 1\right)\|\theta - \boldsymbol{\theta}_\pi^*(x)\|_2^2,$$

where $\tilde{\theta} = t\theta + (1 - t)\boldsymbol{\theta}_\pi^*(x)$ for some $t \in [0, 1]$ and the last step is because

$$\left\|\nabla^2\ell(x, y; \tilde{\theta})\right\|_{\mathrm{op}} \leq \left(\frac{(y + \tilde{\eta})^2}{\tilde{\alpha}^3} + \frac{1}{\tilde{\alpha}}\right)\exp\left(-\frac{y + \tilde{\eta}}{\tilde{\alpha}} - 1\right)$$

Let $\xi = \min(\underline{\alpha}, \bar{\eta})/2$. For any $\theta$ such that $\|\theta - \boldsymbol{\theta}_\pi^*(x)\|_2 \leq \xi$, we also have $|\tilde{\alpha} - \boldsymbol{\alpha}_\pi^*(x)| \leq \xi$ and $|\tilde{\eta} - \boldsymbol{\eta}_\pi^*(x)| \leq \xi$. Then

$$\frac{1}{2}\left(\frac{(y + \tilde{\eta})^2}{\tilde{\alpha}^3} + \frac{1}{\tilde{\alpha}}\right)\exp\left(-\frac{y + \tilde{\eta}}{\tilde{\alpha}} - 1\right) \leq \left(\frac{8\bar{y}^2 + 8\bar{\eta}^2}{\underline{\alpha}^3} + \frac{2}{\underline{\alpha}}\right) \cdot \exp\left(\frac{2\bar{y} + 4\bar{\eta}}{\underline{\alpha}} - 1\right).$$

Letting the right-hand side be $\bar{\ell}(x, y)$, we have thus completed the proof of (2).

**Proof of (3).** By the Taylor expansion,

$$\ell(x, y; \theta) - \ell(x, y; \boldsymbol{\theta}_\pi^*(x)) = \nabla\ell(x, y; \tilde{\boldsymbol{\theta}})^\top \big(\boldsymbol{\theta}(x) - \boldsymbol{\theta}_\pi^*(x)\big),$$

where $\tilde{\boldsymbol{\theta}} = t\boldsymbol{\theta}(x) + (1 - t)\boldsymbol{\theta}_\pi^*$ for some $t \in [0, 1]$. Let $\xi_1 = \min(\underline{\alpha}, \bar{\eta})/2$. When $\|\boldsymbol{\theta} - \boldsymbol{\theta}_\pi^*\|_{L_\infty} \leq \xi_1$, we have $|\tilde{\alpha} - \boldsymbol{\alpha}_\pi^*(x)| \leq \xi_1$ and $|\tilde{\eta} - \boldsymbol{\eta}_\pi^*(x)| \leq \xi_1$. Plugging the expressions of the gradient in Equation (24), we have

$$\big[\ell(x, y; \boldsymbol{\theta}(x)) - \ell(x, y; \boldsymbol{\theta}_\pi^*(x))\big]^2 = \big[\nabla\ell(x, y; \tilde{\boldsymbol{\theta}}(x))^\top \big(\boldsymbol{\theta}(x) - \boldsymbol{\theta}_\pi^*(x)\big)\big]^2$$

$$\leq \left\{\left[\left(1 + \frac{y + \tilde{\boldsymbol{\eta}}(x)}{\tilde{\boldsymbol{\alpha}}(x)}\right)\exp\left(-\frac{y + \tilde{\boldsymbol{\eta}}(x)}{\tilde{\boldsymbol{\alpha}}(x)} - 1\right) + \delta\right]^2 + \left[1 - \exp\left(-\frac{y + \tilde{\boldsymbol{\eta}}(x)}{\tilde{\boldsymbol{\alpha}}(x)} - 1\right)\right]^2\right\} \cdot \big\|\boldsymbol{\theta}(x) - \boldsymbol{\theta}_\pi^*(x)\big\|_2^2$$

$$\leq C(\bar{y}, \bar{\alpha}, \underline{\alpha}, \bar{\eta}, \delta) \cdot \big\|\boldsymbol{\theta}(x) - \boldsymbol{\theta}_\pi^*(x)\big\|_2^2,$$

where $C(\bar{y}, \bar{\alpha}, \underline{\alpha}, \bar{\eta}, \delta)$ is a function of $(\bar{y}, \bar{\alpha}, \underline{\alpha}, \bar{\eta}, \delta)$. Taking the expectation over $P_{X, Y\,|\,A = \pi(X)}$, we have

$$\big\|\ell(X, Y; \boldsymbol{\theta}(X)) - \ell(X, Y; \boldsymbol{\theta}_\pi^*(X))\big\|_{L_2(P_{X, Y\,|\,A = \pi(X)})} \leq C(\bar{y}, \bar{\alpha}, \underline{\alpha}, \bar{\eta}, \delta) \cdot \big\|\boldsymbol{\theta} - \boldsymbol{\theta}^*\big\|_{L_2(P_{X\,|\,A = \pi(X)})},$$

completing the proof of (3).

### C.2 PROOF OF LEMMA B.2

We first introduce the $\ell_2$ distance on the policy space $\Pi$, as well as the corresponding covering number.

**Definition C.1.** Given a function $h$ and a set of realized data $z_1, \ldots, z_n$,

(1) the $\ell_2$ distance between two policies $\pi_1, \pi_2 \in \Pi$ with respect to $\{z_1, \ldots, z_n\}$ is defined as

$$\ell_2\big(\pi_1, \pi_2; \{z_1, \ldots, z_n\}\big) = \sqrt{\frac{\sum_{i=1}^n \big(h(z_i, \pi_1(x_i)) - h(z_i; \pi_2(x_i))\big)^2}{4\sum_{i=1}^n c_i(z_i)^2}}.$$

(2) $N_2(\gamma, \Pi; \{z_1, \ldots, z_n\})$ is the minimum number of policies needed to $\gamma$-cover $\Pi$ under $\ell_2$ with respect $\{z_1, \ldots, z_n\}$.

Under the $\ell_2$ distance, we define a sequence of approximation operators $A_j : \Pi \mapsto \Pi$ for $j \in [J]$, where $J = \lceil \log_2 n \rceil$. Specifically, for any $j = 0, 1, \ldots, J$, let $S_j$ be the set of policies that $2^{-j}$-covers $\Pi$ and satisfies $|S_j| = N_2(2^{-j}, \Pi; \{Z_1, \ldots, Z_n\})$. Specially, $S_0 = \{\bar{\pi}\}$, with $\bar{\pi}$ is an arbitrary policy in $\Pi$ — this is a valid choice since for any $\pi \in \Pi$,

$$\ell_2(\pi, \bar{\pi}; \{z_1, \ldots, z_n\}) = \sqrt{\frac{\sum_{i=1}^n \big(h(z_i, \pi(x_i)) - h(z_i, \bar{\pi}(x_i))\big)^2}{4\sum_{i=1}^n c_i(z_i)^2}} \leq 1.$$

We shall let $\Lambda = 2\sqrt{\sum_{i=1}^n c_i(z_i)^2}$ to denote the normalization factor. The approximation operators are defined in a backward manner: for any $\pi \in \Pi$,

(1) define $A_J[\pi] = \underset{\pi' \in S_J}{\operatorname{argmin}} \, \ell_2\big(\pi, \pi'; \{z_1, \ldots, z_n\}\big)$;

(2) for $j = J - 1, \ldots, 0$, define
$$A_j[\pi] = \underset{\pi' \in S_j}{\operatorname{argmin}} \, \ell_2\big(A_{j+1}[\pi], \pi'; \{z_1, \ldots, z_n\}\big).$$

Using the sequential approximation operators, we decompose the inner expectation term in (16) (Rademacher complexity) as

$$\mathbb{E}_\epsilon \left[ \sup_{\pi \in \Pi} \left| \frac{1}{n} \sum_{i \in [n]} \epsilon_i h(Z_i, \pi(X_i)) \right| \right]$$

$$\leq \mathbb{E}_\epsilon \left[ \sup_{\pi \in \Pi} \left| \frac{1}{n} \sum_{i \in [n]} \epsilon_i \big[ h(Z_i, \pi(X_i)) - h(Z_i, A_J[\pi](X_i)) \big] \right| \right]$$

$$+ \mathbb{E}_\epsilon \left[ \sup_{\pi \in \Pi} \left| \sum_{j=1}^J \frac{1}{n} \sum_{i \in [n]} \epsilon_i \big[ h(Z_i, A_j[\pi](X_i)) - h(Z_i, A_{j-1}[\pi](X_i)) \big] \right| \right]$$

$$+ \mathbb{E}_\epsilon \left[ \sup_{\pi \in \Pi} \left| \frac{1}{n} \sum_{i \in [n]} \epsilon_i h(Z_i, A_0[\pi](X_i)) \right| \right]$$

$$=: \Xi_1 + \Xi_2 + \Xi_3.$$

For any $\pi \in \Pi$, by the Cauchy-Schwarz inequality,

$$\sup_{\pi \in \Pi} \left| \frac{1}{n} \sum_{i \in [n]} \epsilon_i \big[ h(z_i, \pi(x_i)) - h(z_i, A_J[\pi](x_i)) \big] \right|$$

$$\leq \frac{1}{n} \sqrt{n \sum_{i \in [n]} \big( h(z_i, \pi(x_i)) - h(z_i, A_J[\pi](x_i)) \big)^2}$$

$$= \frac{\Lambda}{\sqrt{n}} \cdot \ell_2(\pi, A_J(\pi); \{z_1, \ldots, z_n\})$$

$$\leq \frac{\Lambda}{\sqrt{n}} 2^{-J} \leq \frac{\Lambda}{n^{3/2}},$$

where the second-to-last step is because $A_J(\pi)$ is $2^{-J}$-close to $\pi$ and the last step is by the choice of $J$. As a result the above derivation, $\Xi_1 \leq \Lambda/n^{3/2}$.

Next, for any $j = 1, \ldots, J$ we use $P_j$ to denote the projection of projecting a policy to $S_j$, i.e., $A_{j-1}[\pi] = P_{j-1}[A_j[\pi]]$. Once $A_j(\pi)$ is determined, $A_{j-1}(\pi)$ is also determined. For any $s > 0$,

$$
\mathbb{P}_\epsilon \left( \sup_{\pi \in \Pi} \left| \frac{1}{n} \sum_{i \in [n]} \epsilon_i \big[ h(z_i, A_j[\pi](x_i)) - h(z_i; A_{j-1}[\pi](x_i)) \big] \right| \geq s \right)
$$

$$
\leq \sum_{\pi' \in S_j} \mathbb{P}_\epsilon \left( \left| \frac{1}{n} \sum_{i \in [n]} \epsilon_i \Big[ h(z_i, \pi'(x_i)) - h(z_i, P_{j-1}[\pi'](x_i)) \Big] \right| \geq s \right)
$$

$$
\leq \sum_{\pi' \in S_j} 2 \cdot \exp \left( - \frac{2n^2 s^2}{\sum_{i=1}^n \big[ h(z_i, \pi'(x_i)) - h(z_i, P_{j-1}[\pi'](x_i)) \big]^2} \right)
$$

$$
= \sum_{\pi' \in S_j} 2 \cdot \exp \left( - \frac{2n^2 s^2}{\Lambda^2 \ell_2(\pi', P_{j-1}(\pi'); z)^2} \right)
$$

$$
\leq 2 N_2(2^{-j}, \Pi; Z) \cdot \exp \left( - \frac{n^2 s^2}{\Lambda^2 2^{-2j+1}} \right),
$$

we $z$ is a shorthand for $\{z_1, \ldots, z_n\}$. For any $j = 1, \ldots, J$ and $m \in \mathbb{N}$, take

$$
s_{j,m} = \frac{\Lambda}{n 2^{j-1/2}} \sqrt{\log \big( N_2(2^{-j}, \Pi; Z) \cdot 2^{m+1} j^2 \big)}.
$$

For a fixed $m$, with a union bound over $j = 1, \ldots, J$ we have that

$$
P_\epsilon \left( \sup_{\pi \in \Pi} \left| \sum_{j=1}^J \frac{1}{n} \sum_{i \in [n]} \epsilon_i \big[ h(z_i, A_j[\pi](x_i)) - h(z_i, A_{j-1}[\pi](x_i)) \big] \right| \geq \sum_{j=1}^J s_{j,m} \right)
$$

$$
\leq \sum_{j=1}^J P_\epsilon \left( \sup_{\pi \in \Pi} \left| \frac{1}{n} \sum_{i \in [n]} \epsilon_i \big[ h(z_i, A_j[\pi](x_i)) - h(z_i, A_{j-1}[\pi](x_i)) \big] \right| \geq s_{j,m} \right) \leq \sum_{j=1}^J \frac{1}{j^2 2^m} \leq \frac{1}{2^{m-1}}.
$$

To proceed, we shall use the following lemma, whose proof is deferred to Appendix C.3.

**Lemma C.2.** *For any realization $z_1, \ldots, z_n$ and $\gamma > 0$, there is $N_2(\gamma, \Pi; z_1, \ldots, z_n) \leq N_H(\gamma^2, \Pi)$.*

By Lemma C.2, for any $m \in \mathbb{N}_+$,

$$
\sum_{j=1}^J s_{j,m} = \sum_{j=1}^J \frac{\Lambda}{2^{j-1/2} n} \sqrt{\log \big( N_2(2^{-j}, \Pi; Z) \cdot 2^{m+1} j^2 \big)}
$$

$$
\leq \sum_{j=1}^J \frac{\Lambda}{2^{j-1/2} n} \sqrt{\log(N_H(2^{-2j}, \Pi)) + (m+1)\log 2 + 2\log(j)}
$$

$$
\overset{(i)}{\leq} \frac{2\Lambda}{n} \sum_{j=1}^J 2^{-j} \cdot \left( \sqrt{\log(N_H(2^{-2j}, \Pi))} + \sqrt{m+1} + \sqrt{2\log(j)} \right)
$$

$$
\overset{(ii)}{\leq} \frac{4\Lambda}{n} \big( \kappa(\Pi) + \sqrt{m+1} + 1 \big) =: u_m,
$$

where step (i) uses $\sqrt{a+b+c} \leq \sqrt{a} + \sqrt{b} + \sqrt{c}$ for $a, b, c \geq 0$; step (ii) uses the definition of $\kappa(\Pi)$. Then

$$\Xi_2 = \mathbb{E}_\epsilon\left[\sup_{\pi \in \Pi}\left|\sum_{j=1}^J \frac{1}{n}\sum_{i\in[n]}\epsilon_i\Big[h(z_i, A_j[\pi](x_i)) - h(z_i, A_{j-1}[\pi](x_i))\Big]\right|\right]$$

$$= \int_0^\infty \mathbb{P}_\epsilon\left(\sup_{\pi \in \Pi}\left|\sum_{j=1}^J \frac{1}{n}\sum_{i\in[n]}\epsilon_i\Big[h(z_i, A_j[\pi](x_i)) - h(z_i, A_{j-1}[\pi](x_i))\Big]\right| > s\right)\,\mathrm{d}s$$

$$\leq u_1 + \sum_{k=1}^\infty (u_{k+1} - u_k)\cdot 2^{-k+1}$$

$$= \frac{4\Lambda}{n}\cdot\left(\kappa(\Pi) + \sqrt{2} + 1 + \sum_{k=1}^\infty(\sqrt{k+2} - \sqrt{k+1})\cdot 2^{-k+1}\right) \leq \frac{4\Lambda}{n}\cdot\big(\kappa(\Pi) + 7\big).$$

Finally, we consider $\Xi_3$. Recall that $S_0 = \{\bar{\pi}\}$, and therefore

$$\Xi_3 = \mathbb{E}_\epsilon\left[\left|\frac{1}{n}\sum_{i\in[n]}\epsilon_i h(z_i, \bar{\pi}(x_i))\right|\right] = \int_0^\infty \mathbb{P}_\epsilon\left(\left|\frac{1}{n}\sum_{i\in[n]}\epsilon_i h(z_i, \bar{\pi}(x_i))\right| > s\right)\,\mathrm{d}s$$

$$\leq \int_0^\infty 2\exp\left(-\frac{n^2 s^2}{\Lambda^2}\right)\,\mathrm{d}s = \frac{3\Lambda}{n}.$$

Putting everything together,

$$\mathbb{E}_\epsilon\left[\left|\frac{1}{n}\sum_{i=1}^n \epsilon_i h\big(x_i, a_i, y_i, \pi(x_i)\big)\right|\right] \leq \frac{\Lambda}{n}\cdot(4\kappa(\Pi) + 32)$$

$$= \frac{2\sqrt{\sum_{i=1}^n c_i(z_i)^2}}{n}(4\kappa(\Pi) + 32).$$

### C.3 PROOF OF LEMMA C.2

Fix $\gamma > 0$. If $N_H(\gamma^2, \Pi) = \infty$, the lemma is trivially true. Otherwise, let $N_0 = N_H(\gamma^2; \Pi)$. For any realization $z_1, \ldots, z_n$, define

$$(\pi_{i,1}^*, \pi_{i,2}^*) = \operatorname*{argmax}_{\pi_1, \pi_2}\big\{|h(z_i, \pi_1(x_i)) - h(z_i, \pi_2(x_i))|\big\}.$$

Implicitly, $(\pi_{i,1}^*, \pi_{i,2}^*)$ depends on $z_i$. For an arbitrary positive integer $m$ and $i \in [n]$, we define

$$n_i = \left\lceil \frac{m}{\Lambda^2 n}\big\{h(z_i, \pi_{i,1}^*(x_i)) - h(z_i, \pi_{i,2}^*(x_i))\big\}^2 \right\rceil,$$

where we recall that $\Lambda^2 = 4\sum_{i=1}^n c_i(z_i)^2$. We then construct a new set of data

$$\{\tilde{z}_1, \ldots, \tilde{z}_N\} = \{z_1, \ldots, z_1, z_2, \ldots, z_2, \ldots, z_n, \ldots, z_n\},$$

where $z_i$ appears $n_i$ times and

$$N = \sum_{i=1}^n n_i = \sum_{i=1}^n \left\lceil \frac{m}{\Lambda^2}\big\{h(z_i, \pi_{i,1}^*(x_i)) - h(z_i, \pi_{i,2}^*(x_i))\big\}^2 \right\rceil \leq m + n.$$

By definition, there exists a policy set $S_0$ to be a $\gamma^2$-cover of $\Pi$ the Hamming distance with respect to $\tilde{x} := \{\tilde{x}_1, \ldots, \tilde{x}_N\}$ such that $|S_0| = N_0$. As a result, for any $\pi \in \Pi$, there exists $\pi' \in S_0$ such that

$H(\pi, \pi'; \tilde{x}) \leq \gamma^2$. On the other hand,

$$H(\pi, \pi'; \tilde{x}) = \frac{1}{N} \sum_{i=1}^{N} \mathbb{1}\{\pi(\tilde{x}_i) \neq \pi'(\tilde{x}_i)\}$$

$$\overset{(i)}{=} \frac{1}{N} \sum_{i=1}^{n} n_i \mathbb{1}\{\pi(x_i) \neq \pi'(x_i)\}$$

$$\geq \frac{1}{N} \sum_{i=1}^{n} \frac{m}{\Lambda^2} \{h(z_i, \pi_{i,1}^*(x_i)) - h(z_i, \pi_{i,2}^*(x_i))\}^2 \cdot \mathbb{1}\{\pi(x_i) \neq \pi'(x_i)\}$$

$$\overset{(ii)}{\geq} \frac{1}{N} \sum_{i=1}^{n} \frac{m}{\Lambda^2} \{h(z_i, \pi(x_i)) - h(z_i, \pi'(x_i))\}^2 \cdot \mathbb{1}\{\pi(x_i) \neq \pi'(x_i)\}$$

$$\overset{(iii)}{=} \frac{1}{N} \sum_{i=1}^{n} \frac{m}{\Lambda^2} \{h(z_i, \pi(x_i)) - h(z_i, \pi'(x_i))\}^2.$$

Above, step (i) and (ii) follow from the choice of $\tilde{z}$ and $(\pi_{i,1}^*, \pi_{i,2}^*)$, respectively; step (iii) is because when $\pi(x_i) = \pi'(x_i)$, $h(z_i, \pi(x_i)) = h(z_i, \pi(x_i'))$. By the definition of the $\ell_2$ distance and that $N \leq m + n$, we further have

$$\gamma^2 \geq H(\pi, \pi'; \tilde{x}) \geq \frac{m}{(m+n)} \ell^2(\pi, \pi'; z).$$

Since $m$ is arbitrary, we take $m$ to infinity and have $\ell_2(\pi, \pi'; z) \leq \gamma$. By definition, $S_0$ is a $\gamma$-cover of $\Pi$ under $\ell_2$ with respect to $z_1, \ldots, z_n$, and therefore $N_2(\gamma, \Pi; z_1, \ldots, z_n) \leq N_H(\gamma^2, \Pi)$.

## C.4 Proof of Lemma B.3

By Yang et al. (2022, Lemma B12), $g_\delta(q)$ is differentiable in $q$, and

$$g_\delta'(q) = -\frac{\partial_q D_{\mathrm{KL}}(g(q) \,\|\, q)}{\partial_p D_{\mathrm{KL}}(g(q) \,\|\, q)} = \frac{g(q)/q - (1 - g(q))/(1-q)}{\log\big(g(q)/(1 - g(q))\big) - \log\big(q/(1-q)\big)}.$$

Also by Yang et al. (2022, Lemma B12), $g_\delta(q)$ is convex in $q$, so $g_\delta'(q)$ is increasing in $q$. Since $q \in [0.4, 0.6]$, $g_\delta'(q) \geq g_\delta'(0.4)$. From the dual form, we can check that $g(0.4) \geq 0.1$. Plugging in $q = 0.4$, we have

$$g_\delta'(0.4) = \frac{\frac{g(0.4)}{0.4} + \frac{g(0.4)}{0.6} - 5/3}{\log\big(g(0.4)/(1 - g(0.4))\big) - \log(2/3)} = \frac{g(0.4)/0.24 - 5/3}{\log(g(0.4)/(1 - g(0.4))) - \log(2/3)}.$$

Since the function $f(x) = \frac{x/0.24 - 5/3}{\log(x/(1-x)) - \log(2/3)}$ is increasing in $x$ for $x \in (0, 0.4)$, we conclude that

$$g_\delta'(0.4) \geq \frac{1/2.4 - 5/3}{\log(1/9) - \log(2/3)} \geq 1/2,$$

completing the proof.

