# OpenReview forum: "Distributionally Robust Policy Learning under Concept Drifts"
_ICLR.cc/2025/Conference — Submitted to ICLR 2025_

### Official Review · Reviewer_Bw8b · 2024-11-01

**Soundness:** 2
**Presentation:** 3
**Contribution:** 2
**Rating:** 5
**Confidence:** 3

**Summary:**

This paper addresses distributionally robust policy learning under concept drift, focusing on cases where only the conditional relationship between covariates and outcomes changes. Using a KL-divergence-based framework, the authors define an uncertainty set that constrains shifts in the conditional distribution and develop a computational approach supported by duality to evaluate and optimize policies efficiently. They provide theoretical guarantees, including asymptotic normality and convergence rates, adding robustness to the methodology. Initial numerical results suggest the approach is less conservative than traditional joint distribution shift methods.

**Strengths:**

This paper discusses distributionally robust policy learning under concept shift. It relies on standard assumptions and considers a KL-divergence-based evaluation framework. It presents a duality result and provides an efficient estimator based on this result, with guarantees for asymptotic estimation rates and asymptotic normality.

The result is interesting and proposed concept shift makes sense because in practice, considering the worst-case joint distribution of the covariate and the outcome can be too conservative.

**Weaknesses:**

- The duality and theoretical results appear fairly standard, with proofs similar to existing literature.
- The numerical results are very preliminary and lack in-depth discussion.  For example, $𝑥$ is assumed to follow a Gaussian distribution.
- Additionally, the authors should expand on the results in Table 1. While they claim their method is better, the table mainly shows it is less conservative (which is expected, given the smaller uncertainty set they consider compared to the joint distribution worst-case uncertainty set).
- They should provide more justification for why this choice makes sense in practice and discuss how to empirically determine the appropriate uncertainty type for policy learning (joint distribution shift vs. concept shift), which is not covered in the experiments.

**Questions:**

The authors could also expand on how to choose the uncertainty set parameter, as the results depend heavily on this choice. Additionally, they could discuss why they opted for KL-divergence over alternatives like Wasserstein distance, which might offer different insights or benefits in the context of distributional robustness.

---

> ### Author Response · Authors · 2024-11-19
>
> We would like to thank the reviewer for dedicating the time to review our paper and for providing  the insightful comments and helpful suggestions. Please refer to our latest manuscript for the revised version, in which we have made the following edits according to the reviewers' suggestions: (i) we revised the introduction section to explain our contribution of providing distributional robust inference and policy learning under concept shift more clearly; (ii) we proved a lower bound guarantee to show the optimality of our policy learning regret bounds; (iii) we added a new experiment in a more realistic setting that mimics real-world concept shifts.
>
> The following details our responses to the reviewer’s comments and questions.
>
> Weakness #1, Reply: Thank you for your comment. We apologize for the ambiguity in our presentation. We would like to note that we do not attempt to claim Lemma 2.3 as our major contribution,
> and we have modified our narratives accordingly with sufficient credit given to the
> classical results in our revised manuscript.
>
> To clarify, the main contribution of our work is to provide distributional robust inference and policy learning schemes under concept shift.
> For this problem, previous work [2] can **only be applied to discrete distributions
> on finite support with lower bounded probability mass function**.
> Our work provides an algorithm allowing for general types of covariates,
> substantially generalizing the applicability of the algorithm.
> In our revision, we have also included a matching regret lower bound
> that characterizes the fundamental difficulty of the learning problem we consider.
>
> We also note that technical development is not trivial, as the duality form in Lemma 2.3 poses challenge due to the high dimensional nuisance parameters space. The nuisance parameter $\alpha$ in conventional distributional robust offline policy learning literature with joint distribution shift, is a scalar; while in our work, both nuisance parameters $\alpha_\pi(x),\eta_\pi(x)$ are functions of the covariate $x$ and policy $\pi$. For policy estimation, the major challenge is the slow estimation rate of $\alpha_\pi,\eta_\pi$, which is overcome by incorporating a de-biasing technique introduced in Section 3.1. For policy learning, the critical challenge is that it is difficult to estimate $\alpha_\pi,\eta_\pi$ as a function of $\pi$ using the offline data. We overcome this by decoupling the dependence of $\alpha_\pi,\eta_\pi$ on policy to the dependence on actions, which helps us to construct an learning algorithm based on policy trees. We discuss this technique in details in Section 4. Proof-wise, we developed uniform concentration techniques for an estimator with a large nuisance parameter space, utilizing chaining (Appendix B.3).
>
> We have revised our introduction section to better explain the research question and our contribution. Please refer to our latest version.
>
> Weakness #2, Reply: We would like to first note that $X$ follows a uniform distribution as stated in Section 5. We have added a new experiment to include a more realistic simulation setting that mimics real world concept shifts. Please refer to Table 3 in the latest version.
>
> Weakness #3, Reply: Thank you for your comment, and this is
> exactly the motivation of our paper --- that knowing the source of the distribution
> shift effectively shrinks the uncertainty set, thereby yielding less conservative results.
>
> The way we see our work is that, instead of saying our algorithm improves upon the
> existing ones based on joint modeling, we would rather say that it makes it **possible**
> to make use of the one extra bit of information (that the source of shift is in the
> reward distribution) to obtain less conservative results both theoretically
> and empirically. The result of Table 1 demonstrates that our learning algorithm succeed in doing so.
> Meanwhile, our attempt to make this learning paradigm feasible is technically nontrivial (as discussed in our response to Weakness 1), and has a wide range of applications, including new product launches in a familiar market and campaign strategy in face of changes in the value system of the population.

---

> > ### Author Response · Authors · 2024-11-19
> >
> > Weakness #4, Reply: Thank you for your helpful suggestion.
> > We have provided several examples where the concept shift is a more reasonable modeling
> > choice. More generally, we argue that the covariate shift is often identifiable
> > (in causal problems or situations with access to unlabeled data), whereas concept shift is not
> > identifiable --- it is therefore a "waste" to consider the worst-case covariate shift than directly correcting form them.
> > (In fact, our algorithm can be generalized to the setting with an identifiable covariate shift and a bounded conditional distribution shift. We leave it for future work.) Under cases where the decision maker observes no covariate shift and would like to hedge against the risk of concept shift, applying the existing method designed for joint distributional shift [4] is too conservative. It would be reasonable to apply our method, as it performs better under concept shift (Section 5, Table 2 and 3 in the revised version). We have added this discussion to Section 5 in our manuscript.
> >
> > Question #1, Reply: Thank you for your insightful comments. The choice of the uncertainty set parameter $\delta$ is the common challenge in distributional robust policy learning literature. The parameter $\delta$ controls the size of the uncertainty set considered and thus controls the degree of robustness in our model --- the larger $\delta$, the more robust the output.
> > The empirical performance of the algorithm substantially depends on the selection of $\delta$. A small $\delta$ leads to negligible robustification effect and the algorithm would learn an over-aggressive policy; a large $\delta$, on the other hand, tends to yield more conservative results. In practice, choosing the appropriate $\delta$ is problem dependent as it depends on the decision-makers’ own risk-aversion level and their own perception of the new environments. Selecting an appropriate $\delta$ is a common challenge in the distributionally robust optimization
> > literature, and a more detailed discussion can be found in [4].
> >
> > Many works has been dedicated to distributionally robust policy learning under joint distributional shifts described by KL divergence [3,4]. Our works aims to extend this line of work to robust learning under only outcome shifts described by KL divergence. It is straightforward to extend this work to robust policy learning under other $f$ divergence. However, it is not in the scope of this manuscript, and we leave it for future works.
> >
> > 1. Luenberger, David G. Optimization by vector space methods. John Wiley \& Sons, 1997.
> >
> > 2. Mu, Tong, et al. Factored DRO: Factored distributionally robust policies for contextual bandits. Advances in Neural Information Processing Systems 35 (2022): 8318-8331.
> >
> > 3. Kallus, N., Mao, X., Wang, K., and Zhou, Z. Doubly robust distributionally robust off-policy evaluation and learning. In International Conference on Machine Learning, pp. 10598–10632. PMLR, 2022.
> >
> > 4. Si, N., Zhang, F., Zhou, Z., and Blanchet, J. Distributionally robust batch contextual bandits. Management Science, 2023.

---

> > ### Comment · Reviewer_Bw8b · 2024-11-26
> >
> > Thank you for the reply. Now I have a better understanding of the paper. However, after reviewing the paper again, I decided to keep my score unchanged.

---

> ### Author Response · Authors · 2024-11-24
>
> The authors would like to kindly remind the reviewer to read the responses before the discussion deadline. We have revised the manuscript according to the helpful suggestions. Any further comments would be deeply appreciated!

---

> ### Author Response · Authors · 2024-11-30
> **Remaining Concerns?**
>
> Thank you so much for your response! We would like to ask if you have remaining concerns that have not yet been addressed?

---

### Official Review · Reviewer_i8S9 · 2024-11-03

**Soundness:** 2
**Presentation:** 1
**Contribution:** 1
**Rating:** 3
**Confidence:** 4

**Summary:**

This paper studies the problem of distributionally robust contextual bandit, where the uncertainty only lies in the conditional reward distribution P(Y|X).

Some major comments are listed below:
1. First, Lemma 2.3 cannot be claimed as the contribution of this paper, as it is a standard result in the literature of KL constrained DRO. The authors shall provide credit to existing literature properly.
Kullback-Leibler Divergence Constrained Distributionally Robust Optimization, Hu and Hong
Moreover, min over \eta in (3) has a closed form solution. Please check theorem 1 in the above paper. Therefore, there is no need for an iterative method to obtain \eta^*.

2. The technical presentation is hard to follow. Notations are not defined or not clearly introduced before use. For example g_\pi is not defined. What does propensity score function \pi_0 represent is not clear. Does the propensity score function mean the same thing as the behavior policy?

3. Assumption 3.3. holds if $\theta^*_{\pi}(x)$ is continuous in x. Can the authors provide examples for this to hold?

4. The problem in this paper is in an offline setting, however, assumption 2.1 guarantees that any action can be visited in the training dataset with high probability, and therefore, the most challenging part of offline RL, e..g, partial coverage, is not addressed in this paper. How the performance depends on the concentrability coefficient is not clear from this paper. This limits the significance of this work.

5. This paper investigates only distributional shift in the conditional reward, however, this does not seems to require development of new techniques comparing to distribution shift in the entire P(x,y).

**Strengths:**

see the summary

**Weaknesses:**

see the summary

**Questions:**

see the summary

---

> ### Author Response · Authors · 2024-11-19
>
> We would like to thank the reviewer for dedicating the time to review our paper and for providing  the insightful comments and helpful suggestions. Please refer to our latest manuscript for the revised version, in which we have made the following edits according to the reviewers' suggestions: (i) we revised the introduction section to explain our contribution of providing distributional robust inference and policy learning under concept shift more clearly; (ii) we proved a lower bound guarantee to show the optimality of our policy learning regret bounds; (iii) we added a new experiment in a more realistic setting that mimics real-world concept shifts.
>
> The following details our responses to the reviewer’s comments and questions.
>
> #1, Reply: Thank you for the comments!
>
> **Contributions.**
> We apologize for the ambiguity in stating our contribution. In fact, we do not attempt to claim Lemma 2.3 as the contribution of our work --- we have modified our narratives accordingly
> in the revised manuscript, giving credit to the standard results.
>
> To clarify, the main contribution of our work is to provide distributional robust policy evaluation and policy learning schemes under conditional reward shifts --- that is, the inference problems.
> The type of distribution shift model we consider leads to an objective that
> contains nuisance parameters that are high-dimensional (they are functions
> of the covariates); substantial works have been devoted to dealing with
> these nuisance parameters to achieve an efficient policy evaluation/learning algorithm.
> We have also included a matching regret lower bound in the revision,
> proving that our proposed learning algorithm is minimax-optimal in terms of the
> sample size and the policy class complexity. In the revised manuscript, we have included
> a table detailing the comparison with our work and preious ones.
>
> **Solving for** $\eta^*$.
> Indeed, for the DRO problem, $\eta^*$ can be solved analytically and
> previous work [1] indeed adopts Theorem 1 in [2] to establish
> strong duality; such a formulation, however,
> leads to an objective function
> \begin{align*}
> \max_{\mathbf{\alpha}} -\mathbb{E}\bigg[\mathbf{\alpha}(X) \log
> \Big(\mathbb{E}\Big[\exp\Big(-\frac{Y(\pi(X))}{\mathbf{\alpha}(X)}\Big) \mid X\Big]\Big) +
> \mathbf{\alpha}(X)\delta\bigg].
> \end{align*}
> In turns out that it is hard to work with the objective function above
> to derive an efficient estimator (and correspondingly the learning algorithm)
> because of the conditional expectation in the logrithmic function. We then choose to
> work with the objective function in our work. We have revised our introduction section to better explain the research question and our contribution. Please refer to the latest version.
>
> #2, Reply: We apologize for the ambiguity in our text. We have edited the manuscript to make the definition clearer. We define $g_\pi(x):=\mathbb{E}[G_\pi(X,Y(\pi(X)))|X=x]$. The behavior policy $\pi_0$ is the action-assignment during data collection, and the propensity score function $\pi_0(a\mid x)$ is the probability of $\pi_0$ assigning action $a$ given the covariate $x$ (Line 166 in the revised version).
>
> #3, Reply: Thank you the question. As we have mentioned in Line 303 of our submission,
> a previous paper [3] has provided detailed examples where $\theta_\pi^*(x)$
> is continuous in $x$ (in their
> Appendix B.2). For instance, when the conditional distribution of $Y(\pi(X)) \mid X = x$
> is continous in $x$, then the resulting $\mathbf{\theta}_\pi(x)$ is continuous in $x$
> (e.g., $Y(\pi(X)) \mid X = x\sim \mathcal{N}(\mu(x),\sigma^2)$ for some smooth function
> $\mu(x)$).
>
> #4, Reply: Thank you for the comment.
> We would like to first note that we do not assume knowing the behavior policy
> $\pi_0$ and use the data to learn it (this is typically not the case in offline RL
> unless there are additional assumptions on the reward distribution) --- without the overlap
> condition, learning the behavior policy $\pi_0$ becomes a challenging problem itself.
>
> Meanwhile, we note that Assumption 2.1 is standard in offline policy learning literature (see e.g. [1,4-7]), in which the overlap assumption ensures sufficient exploration when collecting the training dataset. The regret bound in Theorem 4.2 shows the dependence of the performance on the concentrability coefficient $\varepsilon$: if given the covariate $x$, some action $a$ has little probability of being explored by the behavior policy $\pi_0$, then it is hard to learn the outcome of the action covariate pair $(x,a)$ and thus incurring a larger regret.
>
> We agree  that partial coverage is an important and challenging problem in offline policy learning. In the setting when we have access to $\pi_0$ (potentially without overlap),
> we conjecture that we could leverage the pessimistic framework in a fashion similar to [8]
> to achieve a similar bound in the absence of overlap conditions. We leave the investigation to future work since it is outside the scope of the current one.

---

> > ### Author Response · Authors · 2024-11-19
> >
> > #5 Reply: Thank you for your comment. As discussed in our response to #1,
> > our optimization problem admits an ``per-$x$'' optimizer, which poses a significant
> > challenge to the subsequent inference problems since now the nuisance parameters are
> > of high dimensions. To be precise, the nuisance parameter $\alpha$ in conventional distributional robust offline policy learning literature with joint distribution shift, is a scalar;
> > whiles in our work, both nuisance parameters $\alpha_\pi(x),\eta_\pi(x)$ are functions of the covariate $x$. For policy estimation, the major challenge is the slow estimation rate of $\alpha_\pi,\eta_\pi$, and we overcome this by incorporating a novel de-biasing technique introduced in Section 3.1.
> > For policy learning, the critical challenge is that it is difficult to estimate $\alpha_\pi,\eta_\pi$ as a function of $\pi$ using the offline data. We overcome this by decoupling the dependence of $\alpha_\pi,\eta_\pi$ on policy to the dependence on actions, which helps us to construct an learning algorithm based on policy trees. We discuss this technique in details in Section 4. Proof-wise, we developed uniform concentration results
> > for an estimator with a large nuisance parameter space, which is again,
> > requires a different treatment than that in joint distributional shift literature (Appendix B.3).
> >
> > 1. Mu, Tong, et al. Factored DRO: Factored distributionally robust policies for contextual bandits. Advances in Neural Information Processing Systems 35 (2022): 8318-8331.
> >
> > 2. Hu, Zhaolin, and L. Jeff Hong. "Kullback-Leibler divergence constrained distributionally robust optimization." Available at Optimization Online 1.2 (2013): 9.
> >
> > 3. Jin, Ying, Zhimei Ren, and Zhengyuan Zhou. "Sensitivity analysis under the $ f $-sensitivity models: a distributional robustness perspective." arXiv preprint arXiv:2203.04373 (2022).
> >
> > 4. Athey, Susan, and Stefan Wager. "Policy learning with observational data." Econometrica 89.1 (2021): 133-161.
> >
> > 5. Kallus, N., Mao, X., Wang, K., and Zhou, Z. Doubly robust distributionally robust off-policy evaluation and learning. In International Conference on Machine Learning, pp. 10598–10632. PMLR, 2022.
> >
> > 6. Si, N., Zhang, F., Zhou, Z., and Blanchet, J. Distributionally robust batch contextual bandits. Management Science, 2023.
> >
> > 7. Zhou, Zhengyuan, Susan Athey, and Stefan Wager. "Offline multi-action policy learning: Generalization and optimization." Operations Research 71.1 (2023): 148-183.
> >
> > 8. Jin, Ying, et al. "Policy learning" without''overlap: Pessimism and generalized empirical Bernstein's inequality." arXiv preprint arXiv:2212.09900 (2022).

---

> ### Author Response · Authors · 2024-11-24
>
> The authors would like to kindly remind the reviewer to read the responses before the discussion deadline. We have revised the manuscript according to the helpful suggestions. Any further comments would be deeply appreciated!

---

> > ### Comment · Reviewer_i8S9 · 2024-11-24
> > **Thank you**
> >
> > I would like to thank the authors for the response. However my major concern about the partial coverage of offline rl still remains. I do not see a straightforward extension of this result to the partial covered setting. For the third question can you provide an exact example where this assumption is satisfied and explain how widely this assumption generalize?
> >  For now I will keep my score.

---

> > > ### Author Response · Authors · 2024-11-25
> > >
> > > Thank you for the reply.
> > >
> > > 1. **Partial coverage/overlap**.
> > > We want to reiterate that (1) in the setting we consider --- where **we do not know
> > > the form of behavior policy and we do not put parametric assumptions on the
> > > reward distribution** --- the overlap assumption is standard in the literature
> > > (please see the list of references
> > > [1,4-7] we provided in our previous reply).
> > > To the best of our knowledge, there is **no** off-policy learning
> > > literature that addresses the partial coverage issue without knowing
> > > the form of the behavior policy and not putting parametric assumptions on the outcome model.
> > > Would you kindly provide the references if we missed anything?
> > > (2) More importantly, partial coverage/overlap (itself being a challenging issue)
> > > is not the focus
> > > of this work.
> > > Instead, we consider the problem of **off-line policy learning under concept drift**.
> > > Addressing the relaxation of overlap is beyond the scope of our work.
> > > (3) The major applications we consider in this paper are observational studies
> > > (as opposed to data adaptively collected by bandit or RL algorithms).
> > > In observational studies, the overlap assumption is also reasonable
> > > (again see the references provided in
> > > our previous reply).
> > >
> > > 2. **Concrete examples satisfying Assumption 3.3.**
> > > In reply to the reviewer's second concern, Assumption 3.3 is discussed in details in [3] as mentioned before. Intuitively, Assumption 3.3 is satisfied when the potential outcome has similar conditional distributions for similar covariates.
> > > As mentioned in our previous reply, concrete examples can be
> > > when the outcome distribution is in the form of
> > > $Y(\pi(X)) \mid X = x\sim \mathcal{N}(\mu(x),\sigma^2)$, e.g.,
> > > $Y(\pi(x)) = x^\top \beta + \epsilon$,
> > > $Y(\pi(x)) = (x^\top \beta)^2 + \epsilon$, where $\epsilon\sim \mathcal{N}(0,\sigma^2)$. In theory, this assumption is satisfied if the expected outcome given some covariate $x$ is a continuous function in $x$, which is a reasonable assumption in a wide range of real world applications. For example, in healthcare, conducting the same treatment on patients with similar covariates would yield similar outcomes.

---

> > > > ### Author Response · Authors · 2024-11-30
> > > > **Remaining Concerns?**
> > > >
> > > > As the discussion period deadline approaches, we would like to ask if you have remaining concerns that have not yet been addressed? Any further feedback would be greatly appreciated.

---

### Official Review · Reviewer_QE3j · 2024-11-04

**Soundness:** 3
**Presentation:** 2
**Contribution:** 2
**Rating:** 5
**Confidence:** 3

**Summary:**

This paper studies robust policy learning under the concept drift, where the distributional shift occurs only in the conditional reward distribution. The authors first develop a doubly-robust estimator for the worst-case policy value under concept drift, which is proved to be asymptotic normal, and then design a robust policy learning algorithm which achieves $n^{-1/2}$ regret.

**Strengths:**

Separating concept shift from covariate shift offers a fresh and insightful perspective in this field. The methods developed in this paper come with strong theoretical guarantees: the policy value estimator exhibits asymptotic normality, and the policy learning algorithm has $n^{-1/2}$ sub-optimality gap, improving previous results. Additionally, experiments conducted in a synthetic environment demonstrate that the proposed algorithm consistently outperforms baseline methods, further supporting the effectiveness of the approach.

**Weaknesses:**

While the paper is motivated by the idea that existing methods may be suboptimal under concept shift alone, it does not clearly demonstrate how its proposed rates improve upon existing ones. The improvement noted in Remark 4.3 seems more a result of advanced theorem-proving techniques (e.g., chaining) rather than a genuine improvement in performance metrics. Additionally, while the introduction raises the question of "optimal worst-case average performance" (lines 75-77), the paper lacks an optimality analysis of the derived bounds. Consequently, the theoretical contributions do not align strongly with the paper's motivation and feel somewhat insufficient in addressing the posed questions.

Furthermore, the experiments are conducted solely in a synthetic environment, with no practical examples illustrating the benefits of the proposed algorithm.

**Questions:**

How are the rates here compared with existing rates?

Are there lower bounds of value estimation and regret?

---

> ### Author Response · Authors · 2024-11-19
>
> We would like to thank the reviewer for dedicating the time to review our paper and for providing  the insightful comments and helpful suggestions. Please refer to our latest manuscript for the revised version, in which we have made the following edits according to the reviewers' suggestions: (i) we revised the introduction section to explain our contribution of providing distributional robust inference and policy learning under concept shift more clearly; (ii) we proved a lower bound guarantee to show the optimality of our policy learning regret bounds; (iii) we added a new experiment in a more realistic setting that mimics real-world concept shifts.
>
> The following details our responses to the reviewer’s comments and questions.
>
> Weakness #1: "While the paper is motivated by the idea that existing methods may be suboptimal under concept shift alone, it does not clearly demonstrate how its proposed rates improve upon existing ones. The improvement noted in Remark 4.3 seems more a result of advanced theorem-proving techniques (e.g., chaining) rather than a genuine improvement in performance metrics."
>
> Reply: Thank you for the question.
> We would like to first clarify that our work improves the existing literature [1] not only in the regret bounds but also in the applicability of the algorithm. To be specific, [1] also investigates policy learning under conditional reward shift; **their methodology and theory, however, require the covariate $X$ to have a finite support and a
> lower-bounded probability mass function**, which restricts its applicability to real-world problems with continues covariates.
> Our work can handle general types of $X$ (with a finite or infinite support). This extension substantially improves the applicability of the
> algorithm, and is technically highly nontrivial --- in terms of algorithmic design
> and theoretical treatment. (At a high level, the challenge here is that we
> cannot enumerate all the possible values in the support of $X$ as in [1].) In the meantime, we also improve the regret bound $O(n^{-\frac{1}{2}}\log n)$ in [1] by a $\log n$ factor through the chaining proof technique.
>
> Weakness #2: "Additionally, while the introduction raises the question of "optimal worst-case average performance" (lines 75-77), the paper lacks an optimality analysis of the derived bounds. Consequently, the theoretical contributions do not align strongly with the paper's motivation and feel somewhat insufficient in addressing the posed questions."
>
> Reply: Thank you so much for your helpful suggestion.
> We have added a lower bound on the regret that matches our upper bound in terms of
> sample size and the policy class complexity, establishing the optimality of our algorithm. Please refer to our latest version for details.
>
> Weakness #3: "Furthermore, the experiments are conducted solely in a synthetic environment, with no practical examples illustrating the benefits of the proposed algorithm."
>
> Reply: Thank you for your helpful suggestion! In the revision, we have added a new experiment with a more realistic setting
> that mimics real-world concept shifts (we evaluate the learned policy under
> target environment instead of directing evaluating the theoretical worst-case policy
> value). Please refer to our latest version.
> We leave the experiments using real-world dataset for future works, once we have access to such datasets.

---

> > ### Author Response · Authors · 2024-11-19
> >
> > Question #1, Reply: The most relevant paper to our work is [1], which also investigate offline policy learning under the worst-case concept drift. **Note that [1] only $X$ finitely supported covariate $X$ with lower-bounded probability mass function
> > while our algorithm could handle general types of $X$ (on finite or infinite support).**
> > In terms of rates:
> >
> > $\bullet$ When the covariate assumptions are met, [1] achieves a regret bound
> > $O(\sqrt{\frac{\log(n) \log(|\mathcal{X}||\mathcal{A}|/\delta)}{n}})$ with probability at least $1-\delta$. Here $|\mathcal{X}|$ and $|\mathcal{A}|$ refers to the cardinality of
> > the support and the action set, respectively.
> >
> > $\bullet$ Our proposed algorithm achieves a regret bound of
> > $O(\frac{\kappa(\Pi) + \sqrt{\log(1/\delta)}}{\sqrt{n}})$,
> > with probability at least $1-\delta$. Here $\kappa(\Pi)$ is
> > the entropy integral of $\Pi$ under the Hamming distance, which reduces
> > to $O(\sqrt{|\mathcal{X}||\mathcal{A}|})$ when the covariate support is
> > finite. A detailed discussion of this comparison is in Section 1.1.
> >
> > Question #2, Reply: Yes, we have added a matching lower bound
> > on the learning regret.
> > Please refer to our latest manuscript.
> >
> > 1. Mu, Tong, et al. Factored DRO: Factored distributionally robust policies for contextual bandits. Advances in Neural Information Processing Systems 35 (2022): 8318-8331.

---

> ### Author Response · Authors · 2024-11-24
>
> The authors would like to kindly remind the reviewer to read the responses before the discussion deadline. We have revised the manuscript according to the helpful suggestions. Any further comments would be deeply appreciated!

---

> > ### Comment · Reviewer_QE3j · 2024-11-26
> >
> > Thank you very much for your responses! I will keep my rating.

---

> ### Author Response · Authors · 2024-11-30
> **Remaining Concerns?**
>
> Thank you so much for your response! We would like to ask if you have remaining concerns that have not yet been addressed?

---

### Meta-Review · Area_Chair_49ta · 2024-12-21

**Metareview:**

This paper investigates the distributionally robust bandit problem, focusing on distributional shifts in the conditional distribution of rewards, referred to as concept shift. The authors propose a doubly-robust estimator for the worst-case policy value under concept drift and design a robust policy learning algorithm. Theoretical analysis is provided for both the estimator and the algorithm. While the novelty of addressing concept shift is acknowledged, the reviewers remain unconvinced about the motivation and technical contributions of this work. Specifically, the paper appears to generalize an immediately relevant prior work to a slightly broader setting. Moreover, the duality and proof techniques employed are standard in the literature on distributionally robust bandits. The writing does not clearly justify why separate techniques are necessary for addressing the concept shift problem. The paper could be improved by providing greater clarity on the problem’s motivation and technical contributions or by conducting more extensive experiments.

**Additional Comments On Reviewer Discussion:**

The rebuttal effectively clarifies technical implementation details; however, it does not sufficiently address fundamental concerns regarding the research motivation and technical innovation. A more thorough discussion of these aspects would strengthen the contribution of this work.

---

### Decision · Program_Chairs · 2025-01-22

Reject